# Truncating Trajectories in Monte Carlo Policy Evaluation: an Adaptive Approach

**Riccardo Poiani**
DEIB, Politecnico di Milano
riccardo.poiani@polimi.it

**Nicole Nobili**
DEIB, Politecnico di Milano
nicole.nobili@mail.polimi.it

**Alberto Maria Metelli**
DEIB, Politecnico di Milano
albertomaria.metelli@polimi.it

**Marcello Restelli**
DEIB, Politecnico di Milano
marcello.restelli@polimi.it

## Abstract

Policy evaluation via Monte Carlo (MC) simulation is at the core of many MC Reinforcement Learning (RL) algorithms (e.g., policy gradient methods). In this context, the designer of the learning system specifies an interaction budget that the agent usually spends by collecting trajectories of *fixed length* within a simulator. However, is this data collection strategy the best option? To answer this question, in this paper, we consider as quality index the variance of an unbiased policy return estimator that uses trajectories of different lengths, i.e., *truncated*. We first derive a closed-form expression of this variance that clearly shows the sub-optimality of the fixed-length trajectory schedule. Furthermore, it suggests that adaptive data collection strategies that spend the available budget sequentially might be able to allocate a larger portion of transitions in timesteps in which more accurate sampling is required to reduce the variance of the final estimate. Building on these findings, we present an *adaptive* algorithm called **R**obust and **I**terative **D**ata collection strategy **O**ptimization (RIDO). The main intuition behind RIDO is to split the available interaction budget into mini-batches. At each round, the agent determines the most convenient schedule of trajectories that minimizes an empirical and robust estimate of the estimator's variance. After discussing the theoretical properties of our method, we conclude by assessing its performance across multiple domains. Our results show that RIDO can adapt its trajectory schedule toward timesteps where more sampling is required to increase the quality of the final estimation.

## 1 Introduction

In Reinforcement Learning [RL, Sutton and Barto, 2018], an agent acts in an unknown, or partially known, environment to maximize/estimate the infinite expected discounted sum of an external reward signal, i.e., the expected return. Monte Carlo evaluation [MC, Owen, 2013] is at the core of many successful RL algorithms. Whenever a simulator with reset possibility is available to the learning systems designer, a large family of approaches [Williams, 1992, Baxter and Bartlett, 2001, Schulman et al., 2015, 2017, Cobbe et al., 2021] that can be used to solve the RL problem relies on MC simulations for estimating performance or gradient estimates on the task to be solved. In this scenario, since the goal is to estimate the expected infinite sum of rewards, the designer usually specifies a sufficiently large estimation horizon $T$, along with a transition budget $\Lambda = QT$, so that the agent

37th Conference on Neural Information Processing Systems (NeurIPS 2023).

interacts with the simulator, via MC simulation, collecting a batch of $Q$ episodes of length $T$.[1] In this sense, the agent spends its available budget $\Lambda$ *uniformly along the estimation horizon*.

In the context of MC policy evaluation, where the goal lies in estimating the performance of a given policy via MC simulations, Poiani et al. [2023] have recently shown that, given the discounted nature of the RL objective, this uniform-in-the-horizon budget allocation strategy may not be the best option. The core intuition behind their work is that, since rewards are exponentially discounted through time, early interactions weigh exponentially more than late ones, and, consequently, a larger portion of the available budget $\Lambda$ should be dedicated to estimating the initial rewards. To theoretically validate this point, the authors designed a *non-adaptive* budget allocation strategy which, by exploiting the reset possibility of the simulator, leads to the collection of trajectories of different lengths, i.e., *truncated*. They show that this approach provably minimizes Höeffding-like confidence intervals [Boucheron et al., 2003] around the empirical estimates of the expected return. Remarkably, this implies a robustness w.r.t. the uniform strategy that holds for any pair of environment and policy to be evaluated, thus, clearly establishing the theoretical benefits of the proposed method.

Nevertheless, it has to be noticed that although minimizing confidence intervals around the expected return estimator comes with desirable theoretical guarantees (e.g., PAC-bound improvements [Even-Dar et al., 2002]), the resulting schedule of trajectories is computed *before* the interaction with the environment (being determined by the discount factor). Consequently, as the usual uniform-in-the-horizon scheme, it fails to adapt to the peculiarities of the problem at hand, and, ultimately, might not produce a low error estimate. For the sake of clarity, we illustrate this sub-optimality of pre-determined schedule of trajectories with the following extreme examples.

**Example 1.** *Consider an environment where the reward is gathered only at the end of the horizon $T$ (e.g., a goal-based). In this scenario, any strategy that truncates trajectories is intuitively sub-optimal, and we expect that an intelligent agent will spend all its budget according to the uniform schedule.*

**Example 2.** *Conversely, consider a problem where the reward is different from $0$ in the first interaction step only (e.g., in the case of a highly sub-optimal policy that immediately reaches the "zero reward region" of an environment); the uniform schedule wastes a significant portion of its budget collecting samples without variability, and, to reduce the estimation error, we would like the agent to spend all of its interaction budget estimating the reward of the first action.*

Abstracting away from the previous examples, we realize that the main issue of existing approaches arises from the fact that determining a schedule of trajectories *before* interacting with the environment does not allow the agent to adapt it to the environment peculiarities, allocating more samples where this is required to obtain a high-quality estimate. For this reason, in this work, we focus on designing *adaptive* data collection strategies that aim directly at minimizing the *error* of the final estimate. Our main intuition lies in splitting the available budget $\Lambda$ into mini-batches and adapting *online* the data collection strategy of the agent based on the previously collected information.

**Original Contributions and Outline** After introducing the necessary notation and backgrounds (Section 2), we consider the problem of maximizing the estimation quality of a policy expected return estimator using trajectories of different lengths collected via MC simulation with a finite budget $\Lambda$ of transitions (Section 3). More specifically, since we use an *unbiased* return estimator, we consider its variance as a quality index, of which we derive a closed-form expression and analyze it for every possible schedule of trajectories. Then, we define the optimal trajectories schedule as the one that attains the minimum variance subject to the available budget constraint. As expected, computing this optimal data collection strategy requires knowledge of the underlying environment (e.g., the variance of the rewards at each timestep), which is not available to the agent prior to the interaction. Nevertheless, as we shall see, all the quantities that define the optimal strategy can be estimated from the data. These facts confirm our intuition about the weakness of non-adaptive schedules of trajectories and suggest that algorithms that spend the available budget $\Lambda$ iteratively might be able to dynamically allocate their budget to minimize the variance of the final estimate. Building on these findings, in Section 4, we present a novel algorithm, **R**obust and **I**terative **D**ata collection strategy **O**ptimization (RIDO), which splits its available budget $\Lambda$ into mini-batches of interactions that are allocated sequentially to minimize an empirical and robust estimate of the

---

[1]While another large class of RL algorithms is based on Temporal Difference [TD, Sutton and Barto, 2018] learning, which do not require the finite horizon nor the reset possibility, Monte Carlo simulation approaches continue to be extensively adopted. Indeed, unlike TD methods, they can be applied effortlessly to non-Markovian environments, which is a common occurrence in real-world problems.

objective function of interest, i.e., the variance of the estimator. Furthermore, we perform a statistical analysis on the behavior of RIDO, and we derive theoretical guarantees expressed as upper bounds on the variance of the policy return estimator. Our result shows that, under favorable conditions, the variance of the return estimator computed by RIDO is of the same order as the one of the oracle's baseline. To conclude, in Section 5, we conduct an experimental comparison between RIDO and non-adaptive schedules. As we verify, our method achieves the most competitive performance across different domains, discount factor values, and budget, thus clearly highlighting the benefits of adaptive strategies over pre-determined ones.

## 2   Backgrounds and Notation

This section provides the notation and necessary backgrounds used in the rest of this document.

**Markov Decision Processes**  A discrete-time Markov Decision Process [MDP, Puterman, 2014] is defined as a tuple $\mathcal{M} := (\mathcal{S}, \mathcal{A}, R, P, \gamma, \nu)$, where $\mathcal{S}$ is the set of states, $\mathcal{A}$ is the set of actions, $R : \mathcal{S} \times \mathcal{A} \to [0, 1]$ is the reward function the specifies the reward $R(s, a)$ received by the agent upon taking action $a$ in state $s$, $P : \mathcal{S} \times \mathcal{A} \to \Delta(\mathcal{S})^2$ is the transition kernel that specifies the probability distribution over the next states $P(\cdot|s, a)$, when taking action $a$ in state $s$, $\gamma \in (0, 1)$ is the discount factor, and $\nu \in \Delta(\mathcal{S})$ is the distribution over initial states. The agent's behavior is modeled by a policy $\pi : \mathcal{S} \to \Delta(\mathcal{A})$, which for each state $s$, prescribes a distribution over actions $\pi(\cdot|s)$. A trajectory $\boldsymbol{\tau}_h$ of length $h$ is a sequence of states and actions $(s_0, a_0, s_1, \ldots s_{h-1}, a_{h-1}, s_h)$ observed by following $\pi$ for $h$ steps, where $s_0 \sim \nu$, $a_t \sim \pi(\cdot|s_t)$, and $s_{t+1} \sim P(\cdot|s_t, a_t)$ for $t < h$. The return of a trajectory is defined as $G(\boldsymbol{\tau}_h) = \sum_{t=0}^{h-1} \gamma^t R_t$, where $R_t$ is shortcut for $R(s_t, a_t)$. The agent that is following policy $\pi$ is evaluated according to expected cumulative discounted sum of rewards over an estimation horizon $T$,[3] namely $J(\pi) = \mathbb{E}_\pi \left[ \sum_{t=0}^{T-1} \gamma^t R_t \right]$, where the expectation is taken w.r.t. the stochasticity of the policy, the transition kernel, and the initial state distribution.

**Data Collection Strategy**  Poiani et al. [2023] formalized the concept of Data Collection Strategy (DCS) to model how the agent collects data within an environment. More specifically, given an interaction budget $\Lambda \in \mathbb{N}$ such that $\Lambda \mod T = 0$, a DCS is defined as a $T$-dimensional vector $\boldsymbol{m} := (m_1, \ldots, m_T)$ where $m_h \in \mathbb{N}$ and $\sum_{h=1}^T m_h h = \Lambda$. Each element $m_h$ specifies the number of trajectories of length $h$ that the agent collects in the environment while following a policy $\pi$. Given a DCS $\boldsymbol{m}$, it is possible to compute the total number of steps $\boldsymbol{n} := (n_0, \ldots, n_{T-1})$ that will be gathered by the agent at any step $t$; more specifically, the following relationship holds: $n_{T-1} = m_T$, and $n_t = n_{t+1} + m_{t+1}$ for $t < T - 1$. For this reason, in the rest of the paper we will adopt the most convenient symbol depending on the context. For any DCS $\boldsymbol{m}$ such that $m_T \geq 1$ holds, it is possible to build the following unbiased estimator of $J(\pi)$:

$$\hat{J}_{\boldsymbol{m}}(\pi) = \sum_{h=1}^T \sum_{i=1}^{m_h} \sum_{t=0}^{h-1} \frac{\gamma^t}{n_t} R_t^{(i)}. \tag{1}$$

The two external summations in Equation (1) sum over the collected trajectories of different lengths a rescaled empirical trajectory return, where the reward at step $t$ is divided by the number of samples collected at step $t$.[4] In the case the budget $\Lambda$ is spent uniformly, i.e., $\boldsymbol{m} = \left(0, \ldots, 0, \frac{\Lambda}{T}\right)$, Equation (1) reduces to the usual Monte Carlo estimator of $J(\pi)$, namely $\frac{T}{\Lambda} \sum_{i=1}^{\Lambda/T} \sum_{t=0}^{T-1} \gamma^t R_t^{(i)}$.

**Robust Data Collection Strategy Optimization**  Leveraging the estimator of Equation (2), Poiani et al. [2023] investigated alternatives to the usual uniform-in-the-horizon DCS from the worst-case perspective of confidence intervals [Boucheron et al., 2003]. More specifically, given $\boldsymbol{m}$ such that $m_T \geq 1$, the estimator of Equation (1) enjoys the following generalization of the Höeffding

---

[2]Given a set $\mathcal{X}$, we denote with $\Delta(\mathcal{X})$ the set of probability distributions over $\mathcal{X}$.

[3]As common in Monte-Carlo simulation [see e.g., Papini et al., 2022] we approximate the infinite horizon MDP model with a finite estimation horizon $T$. Indeed, if $T$ is sufficiently large, i.e., $T = \mathcal{O}\left(\frac{1}{1-\gamma} \log \frac{1}{\epsilon}\right)$, the expected return computed with horizon $T$ is $\epsilon$ close to the infinite-horizon one [Kakade, 2003].

[4]Rescaling by $n_t$ prevents the estimate from being biased towards time steps for which more samples are available.

confidence intervals holding with probability at least $1 - \delta$:

$$|J(\pi) - \hat{J}_{\boldsymbol{m}}(\pi)| \leq \sqrt{\frac{1}{2} \log\left(\frac{2}{\delta}\right) \sum_{t=0}^{T-1} \frac{d_t}{n_t}}, \tag{2}$$

where $d_t = \frac{\gamma^t(\gamma^t + \gamma^{t+1} - 2\gamma^T)}{1-\gamma}$ controls the relative importance of samples gathered at step $t$. Poiani et al. [2023] designed a closed-form DCS that provably minimizes the bound of Equation (2). Since $d_t$ is a decreasing function of time whose decay speed is governed by the discount factor $\gamma$, the aforementioned DCS gives priority to the collection of experience at earlier time steps, i.e., it truncates the trajectories. Note that the smaller $\gamma$, the higher the number of samples reserved for earlier time steps. We refer the reader for Theorem 3.3 and Theorem B.10 of their work for the exact expressions of the resulting robust DCS. However, we remark that the resulting schedule is non-adaptive (i.e., it is computed before the interaction with the environment takes place) and its shape depends exclusively on $\Lambda$, $\gamma$, and $T$.

## 3    Toward Adaptive Data Collection Strategies

In this section, we lay down the theoretical groundings behind optimizing data collection strategies that directly aim at minimizing the final estimation error. We stick to methods that adopt the estimator of Equation (1), which has a simple interpretation and desirable theoretical properties. More specifically, since the estimator is unbiased, the Mean Squared Error (MSE) simply reduces to the variance. For this reason, to set a proper baseline for DCS optimization (i.e., an optimal strategy according to the MSE), we first analyze the variance of the estimator of Equation (1) for an arbitrary DCS $\boldsymbol{m}$. The following Theorem (proof in Appendix A) summarizes our result.

**Theorem 3.1.** *Consider a generic DCS $\boldsymbol{m}$ such that $m_T \geq 1$, then:*

$$\mathbb{Var}_{\boldsymbol{m}}\left[\hat{J}_{\boldsymbol{m}}(\pi)\right] = \sum_{t=0}^{T-1} \frac{1}{n_t}\left(\gamma^{2t}\mathbb{Var}(R_t) + 2\sum_{t'=t+1}^{T-1} \gamma^{t+t'}\mathbb{Cov}(R_t, R_{t'})\right) =: \sum_{t=0}^{T-1} \frac{f_t}{n_t}. \tag{3}$$

Theorem 3.1 expresses, in closed form, the variance of the estimator of Equation (1) when adopted with an arbitrary DCS that guarantees the estimation to be unbiased (i.e., $m_T \geq 1$). From Equation (3), we can see that this variance results in a summation, over the different time steps, of $\frac{1}{n_t}$, i.e., the reciprocal of the number of samples collected under $\boldsymbol{m}$ at step $t$, multiplied by the variance of the reward at step $t$ plus the covariances between $R_t$ and the rewards gathered at future steps. For brevity, we shortcut this term with $f_t$. Furthermore, Theorem 3.1 leads to a direct formulation of an optimal DCS baseline for our setting. More specifically, given a budget $\Lambda$, we define the optimal DCS $\boldsymbol{n}^*$ for the estimator in Equation (1) as the solution of the following optimization problem:

$$
\begin{aligned}
\min_{\boldsymbol{n}} \quad & \sum_{t=0}^{T-1} \frac{1}{n_t}\left(\gamma^{2t}\mathbb{Var}(R_t) + 2\sum_{t'=t+1}^{T-1} \gamma^{t+t'}\mathbb{Cov}(R_t, R_{t'})\right) \\
\text{s.t.} \quad & \sum_{t=0}^{T-1} n_t \leq \Lambda \\
& n_t \geq n_{t+1}, \quad \forall t \in \{0, \dots, T-2\} \\
& n_t \in \mathbb{N}_+, \quad \forall t \in \{0, \dots, T-1\},
\end{aligned}
\tag{4}
$$

where the constraints $n_t \geq n_{t+1}$ directly encode the sequential nature of the interaction with the environment. At this point, some comments are in order. First of all, we notice that the above optimization problem nicely captures the intuitive examples of Section 1.

**Example 1** (*cont.*). *When the reward is different from 0 in the last interaction step only, the objective function reduces to $\frac{\gamma^{2(T-1)}}{n_{T-1}}\mathbb{Var}\left[R_{T-1}\right]$, which is clearly minimized for the uniform strategy.*

**Example 2** (*cont.*). *Conversely, when the reward is different from 0 in the first step only, we obtain $\frac{\mathbb{Var}[R_0]}{n_0}$, meaning that the entire interaction budget should be dedicated to estimate $R_0$.*

---
**Algorithm 1** Robust and Iterative DCS Optimization (RIDO).
---
**Require:** Interaction budget $\Lambda$, batch size $b$, robustness level $\beta$, policy $\pi$

1: Collect $\mathcal{D}$ using policy $\pi$ and $\hat{\boldsymbol{n}}_0 = \left( \frac{b}{T}, \ldots, \frac{b}{T} \right)$

2: Set $K = \frac{\Lambda}{b}$ and initialize empirical estimates $\sqrt{\widehat{\mathbb{V}}\mathrm{ar}_1 [R_t]}$ and $\widehat{\mathbb{C}}\mathrm{ov}_1 [R_t, R'_t]$

3: **for** $i = 1, \ldots, K - 1$ **do**

4:     Collect $\mathcal{D}_i$ using policy $\pi$ and $\hat{\boldsymbol{n}}_i$, where $\hat{\boldsymbol{n}}_i$ is computed solving problem (5)

5:     Update empirical estimates $\sqrt{\widehat{\mathbb{V}}\mathrm{ar}_i [R_t]}$ and $\widehat{\mathbb{C}}\mathrm{ov}_i [R_t, R'_t]$ using $\mathcal{D}_i$ and set $\mathcal{D} \leftarrow \mathcal{D} \cup \mathcal{D}_i$

6: **end for**
---

Furthermore, the formulation of problem (4) highlights the pitfalls of a non-adaptive data collection strategies. Indeed, consider, for the sake of clarity, the two examples mentioned above. Before executing policy $\pi$, the agent has no way of distinguishing between the two different objective functions, i.e., $\frac{\gamma^{2(T-1)}}{n_{T-1}} \mathbb{V}\mathrm{ar} [R_{T-1}]$ and $\frac{\mathbb{V}\mathrm{ar}[R_0]}{n_0}$, and, consequently, any pre-determined schedule fails to adapt to the actual objective function. More generally, this is because the optimal strategy resulting from the optimization problem (4) can be computed prior to the interaction with the environment only by an oracle that knows in advance the underlying reward process induced by the agent's policy $\pi$ in the MDP. Nevertheless, we note that all the terms appearing in the objective function, i.e., the only unknowns in optimization problem (4), can be estimated if some interactions with the environment are available to the agent. This suggests that strategies that sequentially allocate the available budget $\Lambda$ might successfully adapt their DCS to minimize Equation (3), i.e., the variance of the return estimator.

## 4   Robust and Iterative DCS Optimization

Given the findings of Section 3, we now present our algorithmic solution that aims at avoiding the highlighted pitfalls of pre-determined DCSs. Our approach is called **R**obust and **I**terative **D**ata collection strategy **O**ptimization (RIDO), and its pseudocode is available in Algorithm 1. The central intuition behind RIDO lies in splitting the available budget $\Lambda$ into mini-batches of interactions that the agent will allocate *sequentially*. At each iteration, the agent will compute the most convenient schedule of trajectories that optimizes an *empirical and robust* version of the objective function presented in (4), whose quality improves as the agent gathers more data.

We now describe in-depth the behavior of the algorithm. For simplicity of exposition and analysis, we suppose that the size of the mini-batch $b$ is such that $b \mod T = 0$ and $b \geq 2T$. At the beginning (Lines 1-2 in Algorithm 1), the agent spends the first mini-batch $\hat{\boldsymbol{n}}_0$ at collecting $\frac{b}{T}$ trajectories of length $T$ (i.e., the uniform approach). This preliminary collection phase is a starting round in which some initial experience is gathered to properly initialize estimates of relevant quantities used throughout the algorithm. More specifically, at each iteration $i$, the agent maintains empirical estimates of the unknown quantities that define the variance of the estimate, i.e., the standard deviation of the reward at step $t$, namely $\sqrt{\widehat{\mathbb{V}}\mathrm{ar}_i [R_t]}$, and the covariances between rewards at different steps, namely $\widehat{\mathbb{C}}\mathrm{ov}_i (R_t, R_{t'})$. Then, at each round (Lines 4-5 in Algorithm 1), the DCS of the current mini-batch $\hat{\boldsymbol{n}}_i$ is computed solving the optimization problem (5) whose objective function is a robust estimate of the objective function of the original optimization problem (4). More specifically, at each round $i$, the agent aims at solving the following problem:

$$
\min_{\boldsymbol{n}} \quad \sum_{t=0}^{T-1} \frac{1}{n_t} \left( \gamma^{2t} \left( \sqrt{\widehat{\mathbb{V}}\mathrm{ar}_i(R_t)} + \mathbf{C}_{i,t}^\sigma \right)^2 + 2 \sum_{t'=t+1}^{T-1} \gamma^{t+t'} \left( \widehat{\mathbb{C}}\mathrm{ov}_i(R_t, R_{t'}) + \mathbf{C}_{i,t,t'}^c \right) \right)
$$

$$
\text{s.t.} \quad \sum_{t=0}^{T-1} n_t \leq b \tag{5}
$$

$$
n_t \geq n_{t+1}, \quad \forall t \in \{0, \ldots, T-2\}
$$

$$
n_t \in \mathbb{N}_+, \quad \forall t \in \{0, \ldots, T-1\},
$$

where $C_{i,t}^{\sigma}$ and $C_{i,t,t'}^{c}$ are exploration bonuses for variances and covariances respectively, defined as:

$$C_{i,t}^{\sigma} := \sqrt{\frac{2 \log (\beta)}{\sum_{j=1}^{i-1} \hat{n}_{j,t}}}, \qquad C_{i,t,t'}^{c} := 3 \sqrt{\frac{2 \log (\beta)}{\sum_{j=1}^{i-1} \hat{n}_{j,t'}}}, \qquad (6)$$

where $\beta \geq 1$ is a hyper-parameter that specifies the amount of exploration used to solve the optimization problem, and $\hat{n}_{j,t}$ is the number of samples collected by RIDO during phase $j$ at time step $t$. We now provide further explanations on the optimization problem (5) and Equation (6). First of all, we notice how each term in the original objective function, namely $f_t$, is replaced with its relative empirical estimation plus exploration bonuses, each of which is directly related to components within $f_t$, e.g., $\mathbb{V}\mathrm{ar}\,(R_t)$ is replaced with $\left( \sqrt{\widehat{\mathbb{V}}\mathrm{ar}_i(R_t)} + C_{i,t}^{\sigma} \right)^2$ and $\mathbb{C}\mathrm{ov}\,(R_t, R_{t'})$ is replaced with $\widehat{\mathbb{C}}\mathrm{ov}_i\,(R_t, R_{t'}) + C_{i,t,t'}^{c}$. Intuitively, the purpose of the exploration bonus is to consider the uncertainty that arises from replacing exact quantities with their empirical estimation. This introduces in RIDO a source of robustness w.r.t. the noise that is intrinsically present in the underlying estimation process. At this point, concerning the shape of Equation (6), focus for the sake of exposition on $C_{i,t}^{\sigma}$. First of all, we notice that the hyper-parameter $\beta$ governs the robustness which is taken into account while replacing $\mathbb{V}\mathrm{ar}\,(R_t)$ with its empirical estimate. Larger values of $\beta$, correspond, indeed, to larger $C_{i,t}^{\sigma}$, and, consequently, a higher level of robustness w.r.t. the uncertainty. Furthermore, as we can notice, $C_{i,t}^{\sigma}$ decreases with the number of samples collected in the previous iterations at step $t$, i.e., $\sum_{j=1}^{i-1} \hat{n}_{j,t}$. This quantity coincides with the number of samples that are used to estimate $\sqrt{\widehat{\mathbb{V}}\mathrm{ar}_i [R_t]}$.[5] This formulation captures the following aspect: more data is available to the agent to estimate $\mathbb{V}\mathrm{ar}\,(R_t)$, the more accurate its estimate will be, and, consequently, its exploration bonus will shrink to 0. As one can expect, with this approach, the quality of the objective function used in RIDO increases with the number of iterations. Consequently, the agent will progressively adapt the mini-batch DCS toward time steps where more data is required to minimize Equation (3), i.e., the variance of the return estimator. We conclude with two remarks. First, we notice that RIDO can be applied with $\gamma = 1$, as it does not deeply rely on the property of discounted sums. Secondly, the optimization problem (5) is a complex integer and non-linear optimization problem. Before diving into the statistical analysis of RIDO, we discuss how to solve (5) in the next section.

## 4.1 Solving the Empirical Optimization Problem

As noticed above, directly solving problem (5) requires significant effort since it is an integer, non-linear optimization problem. In this section, we discuss how to overcome these challenges.

We first perform a *continuous relaxation*, replacing the integer constraint $n_t \in \mathbb{N}_+$ with $n_t \geq 1$. Once a solution $\bar{n}^*$ to the relaxed optimization problem is found, it is possible to obtain a proper (i.e., integer) DCS by flooring each $\bar{n}_t^*$ and allocating the remaining budget uniformly. As we shall see, this approximation introduces constant terms in the theoretical guarantees of RIDO only. At this point, the resulting optimization problem is a non-linear problem that, unfortunately, is generally non-convex. This issue occurs when the following condition is verified for some time step $t$:

$$f_t = \gamma^{2t} \left( \sqrt{\widehat{\mathbb{V}}\mathrm{ar}_i(R_t)} + C_{i,t}^{\sigma} \right)^2 + 2 \sum_{t'=t+1}^{T-1} \gamma^{t+t'} \left( \widehat{\mathbb{C}}\mathrm{ov}_i(R_t, R_{t'}) + C_{i,t,t'}^{c} \right) < 0. \qquad (7)$$

To solve this challenge and make RIDO computationally efficient, we develop an approach based on a hidden property of the original optimization problem (4). More specifically, we start by noticing that even the continuous relaxation of (4) is non-convex since $f_{\bar{t}} < 0$ might occur, for some $\bar{t} \in \{0, \ldots, T-2\}$, in the presence of negative covariances with future steps. In this case, however, since $\sum_{t=\bar{t}}^{T-1} f_t$ represents a proper variance, which is always non-negative, there always exists $t' > \bar{t}$ such that $\sum_{t=\bar{t}}^{t'} f_t \geq 0$. Furthermore, it is possible to show that the optimal solution of the relaxed optimization problem is uniform in the interval $\{\bar{t}, \ldots, t'\}$, namely $n_{\bar{t}}^* = n_{\bar{t}+1}^* = \cdots = n_{t'}^*$ (proof in

---

[5]Similar comments apply to $C_{i,t,t'}^{c}$ as well. The only difference stands in the fact that to estimate the empirical covariance between two subsequence steps $t$ and $t'$, samples up to time $t'$ are required. For this reason, the denominator implies the summation of the number of samples gathered at $t'$ over the previous iterations.

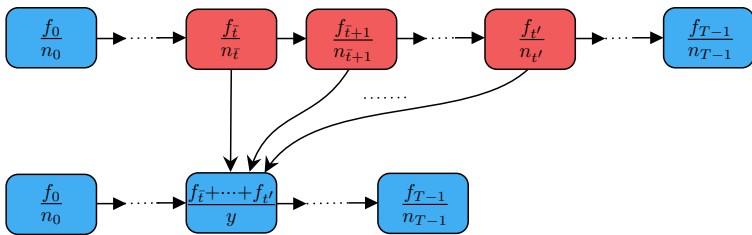

Figure 1: Visualization of the transformation between the optimization problems. The first row shows the objective function of the original optimization problem, while the second one its transformation.

Appendix A). For this reason, it is possible to define a *transformation* of the optimization problem that preserves the optimal solution, in which the variables $n_{\bar{t}}, \ldots, n_{t'}$ are replaced with a single variable $y$. The objective function is modified accordingly, namely $\frac{f_{\bar{t}}}{n_{\bar{t}}} + \cdots + \frac{f_{t'}}{n_{t'}}$ is replaced with $\frac{f_{\bar{t}} + \cdots + f_{t'}}{y}$ in the objective function. A visualization of the transformation is proposed in Figure 1. By repeating the procedure for all the negative $f_t$, we obtain a transformation of the original problem which is now convex. Once the solution to this convex transformed optimization problem is found, one can quickly recover the relaxed DCS in its $T$-dimensional form.

Building on these results, we apply in RIDO a similar procedure that transforms the relaxed version of (5) into a new problem where the negative time steps (i.e., steps in which Equation (7) holds) are "grouped" with future time steps as long as the total summation is positive. In this way, (i) the resulting optimization problem is convex and (ii) as our analysis will reveal, this procedure has no impact on the theoretical properties of RIDO (i.e., the result is the same as assuming access to an oracle that can solve non-linear and non-convex problems). As a concluding remark, we refer the reader to Appendix A for a formal description of the above-mentioned procedure.

### 4.2 Theoretical Analysis

We now present theoretical guarantees on the performance of RIDO. More specifically, we derive high-probability guarantees on the variance of the estimator of Equation (1) when used with the data collected by RIDO. Before diving into the presentation of our results, we highlight some critical challenges behind the result. First, in our analysis, we do not assume access to an oracle that solves (5), but we consider the modifications discussed in Section 4.1 that make the computation tractable. This introduces a first level of challenges in the analysis (e.g., dealing with the roundings that arise from the relaxation and the peculiar strategy that overcomes the non-convexity of the optimization problem). Secondly, we notice that none of the optimization problems (4) and (5), and the ones obtained by relaxing the integer constraints, admit a closed-form solution (further details are provided in Appendix A). This clearly results in an additional challenge in our analysis. At this point, we are ready to state our main theoretical result (proof in Appendix A).

**Theorem 4.1.** *Let $n^*$ be the optimal solution of problem* (4), $f_t$ *as in Equation* (3), $b \geq 2T$ *and* $\beta = \frac{6(T+T^2)\Lambda K}{\delta}$. *Consider the DCS $\hat{n}$ computed by Algorithm 1. Then, with probability at least* $1 - \delta$ *it holds that:*

$$\mathbb{Var}_{\hat{n}}\left[\hat{J}_{\hat{n}}(\pi)\right] \leq 192 \left(\frac{b}{\Lambda}\right)^{\frac{3}{2}} \log(\beta) \left(\sum_{t=0}^{T-1} \gamma^t\right)^2 + 4\mathbb{Var}_{n^*}\left[\hat{J}_{n^*}(\pi)\right] + \frac{2b}{\Lambda} \sum_{t:f_t<0} |f_t|. \quad (8)$$

Equation (8) comprises three terms which we now discuss in detail. The former is directly responsible for taking into account the cumulative error computed during each phase $i$. This term shrinks to zero with rate $\widetilde{\mathcal{O}}\left(\left(\frac{b}{\Lambda}\right)^{\frac{3}{2}} \log \Lambda\right)$. We notice that this gets smaller as we decrease $b$, thus suggesting to use small batch sizes. This should come as no surprise; indeed, using smaller batch sizes intuitively improves the adaptiveness of the algorithm, since a larger portion of the budget $\Lambda$ will be allocated following more precise estimates of the quantities of interest. In this sense, there exists a trade-off between theoretical guarantees and computational requirements, since the number of iterations (and, thus, the number of optimization problems to be solved) grows linearly as the batch size decreases. The second term, instead, is the variance of the optimal DCS computed as in (4), and shrinks with a

rate that is at most $\frac{4}{\Lambda} \sum_{t=0}^{T-1} f_t$. This term, as we shall show below, represents a particularly desirable property. Finally, the last component of Equation (8) is related to the negative terms possibly present in the objective function, and, among the three terms, it is the one with the worst dependence on $\Lambda$. Currently, we are unsure whether this term is an artifact of the analysis, a sub-optimality of the algorithm, or a key challenge of the setting. We leave closing this gap to future work. At this point, we highlight a particular relevant property of Equation (8). Suppose that $f_t \geq 0$ holds for all time steps $t$ (so that the last component is not present). In this case, under the mild assumption that $\sum_{t=0}^{T-1} f_t > 0$ (i.e., the variance is different from 0), for sufficiently large budget of $\Lambda$, we have that (formal statement and proof in Appendix A):

$$\mathbb{V}\mathrm{ar}_{\hat{\boldsymbol{n}}}\left[\hat{J}_{\hat{\boldsymbol{n}}}(\pi)\right] \leq 5\mathbb{V}\mathrm{ar}_{\boldsymbol{n}^*}\left[\hat{J}_{\boldsymbol{n}^*}(\pi)\right]. \tag{9}$$

Thus, in this scenario, the variance of the returned DCS computed by RIDO is proportional to the optimal one. Note that this sort of result is not possible for the uniform strategy, nor for the robust one of Poiani et al. [2023]. Further details on this point are provided in Appendix A.

## 5   Numerical Validation

In this section, we propose numerical validations that aim at assessing the empirical performance of RIDO. More specifically, we focus on the comparison between our approach, the classical uniform-in-the-horizon strategy, and the robust DCS by Poiani et al. [2023]. We report the results across multiple domains, values of budget $\Lambda$, and discount factor $\gamma$. As a performance index, all experiments measure the empirical variance of the estimator in Equation (1) at the end of the data collection process. Before discussing our results in detail, we describe our experimental settings in depth.

**Experimental Setting**  In our experiments, we consider the following four domains. We start with the Inverted Pendulum [Brockman et al., 2016], a classic continuous control benchmark, where the agents' goal is to swing up a suspended body and keep it in the vertical direction. We, then, continue with the Linear Quadratic Gaussian Regulator [LQG, Curtain, 1997], where the agent controls a linear dynamical system with the objective of reducing a total cost that is expressed as a quadratic function. Then, we consider a 2D continuous navigation problem, where an agent starts at the bottom left corner of a room and needs to reach a goal region in the upper right corner. The agent receives reward 0 everywhere except inside the goal area, where the reward is positive and sampled from a Gaussian distribution. Finally, we consider the Ant environment from the MuJoCo [Todorov et al., 2012] suite, where the agent controls a four-legged 3D robot with the goal of moving it forward. Further domain details are provided in Appendix B. Concerning the policy that we evaluate for the Inverted Pendulum and the Ant, we rely on pre-trained deep RL agents made publicly available by Raffin [2020]. For the LQG, instead, we evaluate the optimal policy that is available in closed form by solving the Riccati equations, and, finally, for the 2D navigation task, we roll out a hand-designed policy that minimizes the distance of the agent's position w.r.t. to the center of the goal region. Regarding the performance index, as already anticipated, we report the variance of the empirical policy return at the end of the data collection process. Given a budget and a DCS, for a single run, we estimate this empirical variance using 100 simulations. We then average the results over 100 runs and report the empirical mean together with 95% confidence intervals. We notice that for each considered value of $\Lambda$, the experiment is repeated (i.e., we do not use data collected with smaller $\Lambda$'s). To conclude, we refer the reader to Appendix B for further details on the experiments (e.g., ablations, additional results, experiments with $\gamma = 1$, hyper-parameters, visualizations of the resulting DCSs).

**Results**  Figure 2 reports the results varying the discount factor and the available budget. The second row is obtained under the same experimental setting as the first one, but with lower values of $\gamma$. Let us first focus on the sub-optimality of the non-adaptive DCSs (i.e., the uniform strategy and the robust one of Poiani et al. [2023]). Indeed, as suggested by Theorem 3.1, being computed prior to the interaction with the environment, these algorithms cannot adapt the collection of samples to minimize the variance of the return estimator. This is clear by looking, for instance, at the results of Continuous Navigation and the LQG. Indeed, in the Continuous Navigation domain, the reward is sparse and received close to the end of the estimation horizon $T$. In this scenario, the robust DCS blindly truncates trajectories, thus, avoiding the collection of experience in the most relevant timesteps. Conversely, in the LQG experiments, the optimal policy that arises from the Riccati equation pays a

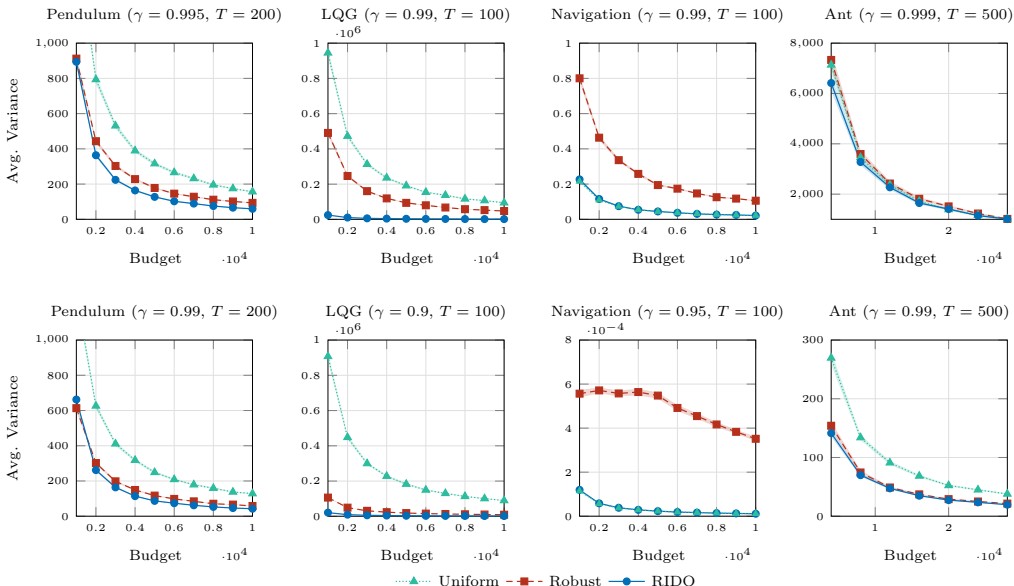

Figure 2: Empirical variance (mean and 95% confidence intervals over 100 runs) on the considered domains and baselines. The first row considers higher values of $\gamma$ w.r.t. the second one.

stochastic control cost[6] at the beginning of the estimation horizon to bring the state of the system close to stability, after which the reward will remain almost constant. In this case, the uniform DCS results in a highly sub-optimal behavior as most of the estimation uncertainty is related to the initial interaction steps. RIDO, on the other hand, thanks to its adaptivity, is able to obtain the best results in both domains. Indeed, in the Continuous Navigation problem, it achieves the same performance level as the uniform strategy, while in the LQG it even outperforms the robust DCS of Poiani et al. [2023]. The reason is that Poiani et al. [2023] truncates trajectories solely depending on the value of $\gamma$, and, therefore, it might waste a portion of its budget in trajectories of sub-optimal length, while RIDO, since it aims at minimizing the variance of the final estimation, is able to focus the collection of data in the most convenient way. Similar comments to those made for the LQG hold for the Pendulum domain as well. Concerning the Ant environment, instead, we notice that for $\gamma = 0.999$ there is no significant difference between any of the presented schedules. Interestingly, however, as soon as we decrease $\gamma$ to $0.99$, we can appreciate the sub-optimality of the uniform strategy, which wastes a portion of its budget in gathering samples that are significantly discounted, and, therefore, their weight in the estimator's variance shrinks to $0$. On the other hand, the robust strategy and RIDO avoid this pitfall thanks to the exploitation of the discount factor, thus obtaining reduced variance estimates. Finally, we remark that RIDO has achieved the most competitive performance across various domains, values of the discount factor, and budget, thus clearly highlighting the benefits of adaptive strategies w.r.t. pre-determined ones.

## 6 Related Works

We now present a comprehensive analysis and discussion of previous works that are closely connected to our own research. First of all, our work focuses on estimating a policy's performance in a given MDP [Sutton and Barto, 2018]. Considering the significance of this task, reducing the variance, or more generally, the error of the return estimator, is a problem that has received significant attention in the literature. A vast family of approaches that can be used to solve this problem deeply exploits the Markovian properties of the environment by relying on Temporal Difference [TD, see, e.g., Singh and Sutton, 1996, Sutton, 1988, Lee and He, 2019, Riquelme et al., 2019, Qu et al., 2019] learning. On the other hand, our work focuses purely on Monte Carlo simulation, which can be transparently applied to non-Markovian environments. Another relevant line of work deals with optimizing the

---

[6]The uncertainty, in this case, arises both from the noise of the system together with the stochasticity of the initial state distribution.

agent's policy to collect data within an environment (i.e., *behavioral* policy) to reduce the variance of an unbiased estimator for the return of a different *target* policy [Hanna et al., 2017, Zhong et al., 2022, Mukherjee et al., 2022]. These techniques are referred to as off-policy evaluation methods and usually rely on Importance Sampling [e.g, Hesterberg, 1988, Owen, 2013] techniques to guarantee the unbiasedness of the resulting estimate. However, these studies significantly differ from ours in that, instead of aiming for a behavior policy that reduces the estimator variance, our goal is to directly exploit the properties of Monte Carlo data collection to reduce the on-policy estimator variance.

In the context of RL, exploration bonuses are widely adopted in control (where the goal is learning an optimal policy) to tackle the exploration-exploitation dilemma [e.g., Brafman and Tennenholtz, 2002, Auer et al., 2008, Tang et al., 2017, Jin et al., 2018, O'Donoghue et al., 2018, Zanette and Brunskill, 2019]. Initially, when the agent has limited knowledge about the environment, the exploration bonuses drive it to explore widely. As the agent's knowledge improves, the exploration bonuses decrease, and the agent can shift towards exploiting its learned policy more. In our work, instead, we use exploration bonuses to introduce a source of robustness w.r.t. the objective function that we are interested in, i.e., the variance of the return estimator for a given data collection strategy.

Finally, the work that is most related to ours is Poiani et al. [2023], where, the concept of truncating trajectories has been analyzed in the context of Monte Carlo RL. More specifically, the authors derived a *non-adaptive* schedule of trajectories that provably minimizes confidence intervals around the return estimator. In this work, on the other hand, we have shown the sub-optimality of pre-determined schedules, and we designed an *adaptive* algorithm that aims at minimizing the variance of the final estimate. The concept of truncating trajectories has also received some attention in other fields of research such as model-based policy optimization [Nguyen et al., 2018, Janner et al., 2019, Bhatia et al., 2022, Zhang et al., 2023], multi-task RL [Farahmand et al., 2016] and imitation learning [Sun et al., 2018]. However, in all these works, the motivation, the method, and the analysis completely differ w.r.t. what has been considered here. Finally, the concept of truncating trajectories in Monte Carlo RL drew inspiration from a recent work in the field of multi-fidelity bandit [Poiani et al., 2022], where the authors considered the idea of cutting trajectories while interacting with the environment to obtain a biased estimate of the return of a policy in planning algorithms such as depth-first search.

## 7 Conclusions and Future Works

In this work, we studied the problem of allocating a budget $\Lambda$ of transitions in the context of Monte Carlo policy evaluation to reduce the error of the policy expected return estimate. Leveraging the formalism of Data Collection Strategy (DCS) to model how an agent spends its interaction budget, we started by analyzing, in closed form, the variance of an unbiased return estimator for any possible DCS. Our result reveals that DCSs determined prior to the interaction with the environment (e.g., the usual uniform-in-the-horizon one and the robust one of Poiani et al. [2023]) fail to satisfy the ultimate goal of policy evaluation, i.e., produce a low error estimate. Furthermore, it also suggests that algorithms that spend the available budget $\Lambda$ iteratively might successfully adapt their strategy to minimize the variance of the return estimator. Inspired by these findings, we propose an *adaptive* method, RIDO, that, by exploiting information that has already been collected, can dynamically adapt its DCS to allocate a larger portion of transitions in time steps in which more accurate sampling is required to reduce the variance of the final estimate. After conducting a theoretical analysis on the properties of the proposed method, we present empirical studies that confirm its adaptivity across a different number of domains, values of budget $\Lambda$, and discount factors $\gamma$.

Our study offers exciting possibilities for future research. For example, it would be interesting to extend our ideas to policy search algorithms (e.g., Williams [1992]), with the goal of finding DCSs that minimize the variance of the empirical gradient that is adopted in the update rule. Furthermore, we notice that, since our approach is purely based on MC simulation, it does not fully leverage the Markovian properties of the underlying MDP. Combining TD techniques [Sutton and Barto, 2018] with mechanisms that truncate trajectories is a challenging and open research question that could lead to further improvements in the efficiency of RL algorithms.

## Acknowledgements

This paper is supported by PNRR-PE-AI FAIR project funded by the NextGeneration EU program.

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
