# A    Proofs and Derivations

In this section, we provide complete proofs of our theoretical results. More specifically, Section A.1 contains the proof of Theorem 3.1; Section A.2 the proof of Theorem 4.1, and Section A.3 proofs and details of additional statements that have been made in the main text (i.e., formal description of the transformation between optimization problems and how we applied this technique in RIDO, difficulties in deriving closed-form solutions for the optimization problems of interest, formal statement and proof of Equation (9), sub-optimality examples of non-adaptive methods whose variance cannot scale with the variance of the optimal DCS, as RIDO, instead, does).

## A.1    Proof of Theorem 3.1

**Theorem 3.1.** *Consider a generic DCS $\boldsymbol{m}$ such that $m_T \geq 1$, then:*

$$\mathbb{V}\mathrm{ar}_{\boldsymbol{m}}\left[\hat{J}_{\boldsymbol{m}}(\pi)\right] = \sum_{t=0}^{T-1}\frac{1}{n_t}\left(\gamma^{2t}\mathbb{V}\mathrm{ar}(R_t) + 2\sum_{t'=t+1}^{T-1}\gamma^{t+t'}\mathbb{C}\mathrm{ov}(R_t, R_{t'})\right) =: \sum_{t=0}^{T-1}\frac{f_t}{n_t}. \quad (3)$$

*Proof.* Given that the different trajectories are independent, we have that:

$$
\begin{aligned}
\mathbb{V}\mathrm{ar}_{\boldsymbol{m}}\left[\hat{J}_{\boldsymbol{m}}(\pi)\right] &= \sum_{h=1}^{T} m_h \mathbb{V}\mathrm{ar}\left[\sum_{t=0}^{h-1}\frac{\gamma^t R_t}{n_t}\right] \\
&= \sum_{h=1}^{T} m_h \sum_{t=0}^{h-1}\mathbb{V}\mathrm{ar}\left[\frac{\gamma^t R_t}{n_t}\right] + \sum_{h=1}^{T} m_h \sum_{t=0}^{h-2}\sum_{t'=t+1}^{h-1} 2\mathbb{C}\mathrm{ov}\left(\frac{\gamma^t R_t}{n_t}, \frac{\gamma^{t'} R_{t'}}{n_{t'}}\right) \\
&= \sum_{h=1}^{T} m_h \sum_{t=0}^{h-1}\frac{\gamma^{2t}}{n_t^2}\mathbb{V}\mathrm{ar}[R_t] + \sum_{h=1}^{T} m_h \sum_{t=0}^{h-2}\sum_{t'=t+1}^{h-1} 2\frac{\gamma^{t+t'}}{n_t n_{t'}}\mathbb{C}\mathrm{ov}\left(R_t, R_{t'}\right),
\end{aligned}
$$

where the first step follows from the fact that different trajectories are independent, the second one from the variance of the sum of dependent random variable, namely $\mathbb{V}\mathrm{ar}\left[\sum_{i=1}^{n} X_i\right] = \sum_{i=1}^{n}\mathbb{V}\mathrm{ar}[X_i] + \sum_{i\neq j}\mathbb{C}\mathrm{ov}(X_i, X_j)$, and the third one by the fact that $\mathbb{V}\mathrm{ar}[aX] = a^2\mathbb{V}\mathrm{ar}[X]$ for some scalar $a \in \mathbb{R}$ and $\mathbb{C}\mathrm{ov}(aX, bY) = ab\mathbb{C}\mathrm{ov}(X, Y)$ for scalars $a, b \in \mathbb{R}$.

At this point, focus on:

$$\sum_{h=1}^{T} m_h \sum_{t=0}^{h-1}\frac{\gamma^{2t}}{n_t^2}\mathbb{V}\mathrm{ar}[R_t],$$

and fix $\bar{t} \in \{0, \ldots, T-1\}$. By unrolling the summation, we notice that its contribution appears only in all $h$ such that $h > \bar{t}$, thus leading to:

$$\sum_{h=1}^{T} m_h \sum_{t=0}^{h-1}\frac{\gamma^{2t}}{n_t^2}\mathbb{V}\mathrm{ar}[R_t], = \sum_{t=0}^{T-1}\frac{\gamma^{2t}}{n_t^2}\mathbb{V}\mathrm{ar}[R_t]\sum_{h=t+1}^{T} m_h.$$

However, given the relationship between $\boldsymbol{n}$ and $\boldsymbol{m}$, we have that:

$$\sum_{h=t+1}^{T} m_h = n_t - n_{t+1} + n_{t+1} - n_{t-2} + \cdots + n_{T-2} - n_{T-1} + n_{T-1} = n_t. \quad (10)$$

Therefore:

$$\sum_{h=1}^{T} m_h \sum_{t=0}^{h-1}\frac{\gamma^{2t}}{n_t^2}\mathbb{V}\mathrm{ar}[R_t] = \sum_{t=0}^{T-1}\frac{\gamma^{2t}}{n_t}\mathbb{V}\mathrm{ar}[R_t]. \quad (11)$$

Now, let us focus on:

$$\sum_{h=1}^{T} m_h \sum_{t=0}^{h-2}\sum_{t'=t+1}^{h-1} 2\frac{\gamma^{t+t'}}{n_t n_{t'}}\mathbb{C}\mathrm{ov}\left(R_t, R_{t'}\right). \quad (12)$$

Fix the the index of the outer summation over time by considering $\bar{t} \in \{0, \ldots, T-2\}$. By unrolling the summation, we notice that its contribution appears only in all $h$ such that $h > \bar{t} + 1$. For this reason, Equation (12) can be rewritten as:

$$\sum_{h=1}^{T} m_h \sum_{t=0}^{h-2} \sum_{t'=t+1}^{h-1} 2\frac{\gamma^{t+t'}}{n_t n_{t'}} \mathbb{C}\text{ov}\left(R_t, R_{t'}\right) = \sum_{t=0}^{T-2} \sum_{h=t+2}^{T} m_h \sum_{t'=t+1}^{h-1} \frac{2\gamma^{t+t'}}{n_t n_{t'}} \mathbb{C}\text{ov}\left(R_t, R_{t'}\right). \tag{13}$$

At this point, fix again $\bar{t} \in \{0, \ldots, T-2\}$ as the index of the outer summation, and consider $t' \geq \bar{t}+1$. By unrolling the summation, we notice that $t'$ appears only for $h > t'$. For this reason, we can rewrite Equation (13) as:

$$\sum_{t=0}^{T-2} \sum_{h=t+2}^{T} m_h \sum_{t'=t+1}^{h-1} \frac{2\gamma^{t+t'}}{n_t n_{t'}} \mathbb{C}\text{ov}\left(R_t, R_{t'}\right) = \sum_{t=0}^{T-2} \sum_{t'=t+1}^{T-1} \frac{2\gamma^{t+t'}}{n_t n_{t'}} \mathbb{C}\text{ov}\left(R_t, R_{t'}\right) \left(\sum_{h=t'+1}^{T} m_h\right). \tag{14}$$

However, by Equation (10), we obtain that:

$$\sum_{h=t'+1}^{T} m_h = n_{t'},$$

thus leading to:

$$\sum_{t=0}^{T-2} \sum_{t'=t+1}^{T-1} \frac{2\gamma^{t+t'}}{n_t n_{t'}} \mathbb{C}\text{ov}\left(R_t, R_{t'}\right) \left(\sum_{h=t'+1}^{T} m_h\right) = \sum_{t=0}^{T-2} \sum_{t'=t+1}^{T-1} \frac{2\gamma^{t+t'}}{n_t} \mathbb{C}\text{ov}\left(R_t, R_{t'}\right). \tag{15}$$

Combining Equation (11) and (15) concludes the proof. $\qquad \square$

## A.2 Proof of Theorem 4.1

To prove Theorem 4.1, we first provide some preliminaries lemmas on the properties of the optimization problems that we are considering, togheter with some technical results that will be used in our proofs. Then, we will move towards the analysis of RIDO.

### A.2.1 Preliminaries for the proof of Theorem 4.1

We begin by proving the fact that for any timestep $t$ in which $f_t$ is negative, that there exists some future timestep $t'$ such that $\sum_{i=t}^{t'} f_i \geq 0$.

**Lemma A.1** (Variance function property). *Consider $f_t = \gamma^{2t}\mathbb{V}\text{ar}R_t + 2\sum_{t'=t+1}^{T-1} \gamma^{t+t'}\mathbb{C}\text{ov}(R_t, R_{t'})$. For any $t \in \{0, \ldots, T-2\}$ such that $f_t < 0$, there exists $\bar{t} > t$ such that $\sum_{i=t}^{\bar{t}} f_t \geq 0$.*

*Proof.* We proceed by contradiction. Suppose the claim to be false, then we would have:

$$\sum_{i=t}^{T-1} f_i < 0.$$

However, by manipulating $\sum_{i=t}^{T-1} f_i$, we obtain:

$$\sum_{i=t}^{T-1} f_i = \sum_{i=t}^{T-1} \gamma^{2i}\mathbb{V}\text{ar}R_i + 2\sum_{t'=i+1}^{T-2} \gamma^{i+t'}\mathbb{C}\text{ov}(R_i, R_{t'}) = \mathbb{V}\text{ar}\left[\sum_{i=t}^{T-1} \gamma^i R_i\right],$$

which is always greater or equal than 0, thus concluding the proof. $\qquad \square$

We then continue by proving the result of Section 4.1 that justifies the transformation between optimization problems. However, rather than considering directly the optimization problem we are interested in (i.e., the one defined with $f_t$), we focus on a generalization that consider arbitrary vectors that satisfy the same properties as the one of Lemma A.1.

**Lemma A.2** (Optimization of Variance-like functions). *Let $c = (c_1, \ldots, c_k)$ with $c_i \in \mathbb{R}$, such that $c_1 < 0$, $\sum_{i=1}^{\bar{k}} c_i \leq 0$ for all $\bar{k} < k$, and $\sum_{i=1}^{k} c_i \geq 0$. Let $\Lambda \geq k$ and consider the following optimization problem:*

$$
\begin{aligned}
\min_{\boldsymbol{x}} \quad & \sum_{i=1}^{K} \frac{c_t}{x_t} \\
\text{s.t.} \quad & \sum_{t=0}^{T-1} x_t = \Lambda \\
& x_i \geq x_{i+1}, \quad \forall i \in \{1, \ldots, K-1\} \\
& x_i \geq 1, \quad \forall i \in \{1, \ldots, K\}.
\end{aligned}
\tag{16}
$$

*Then, $\bar{\boldsymbol{x}} = \left(\frac{\Lambda}{k}, \ldots, \frac{\Lambda}{k}\right)$ is an optimal solution of (16).*

*Proof.* If $\bar{\boldsymbol{x}}$ is an optimal solution of (16), for all $\boldsymbol{x} = (x_1, \ldots, x_k)$ that belongs to the feasible region it holds that:

$$
\sum_{i=1}^{k} \frac{c_i}{\Lambda/k} \leq \sum_{i=1}^{k} \frac{c_i}{x_i},
$$

which can be rewritten as:

$$
\sum_{i=1}^{k-1} \frac{c_i}{\Lambda/k} \leq c_k \left(\frac{1}{x_k} - \frac{1}{\Lambda/k}\right) + \sum_{i=1}^{k-1} \frac{c_i}{x_i} = c_k \frac{\Lambda - kx_k}{x_k \Lambda} + \sum_{i=1}^{k-1} \frac{c_i}{x_i}.
\tag{17}
$$

At this point, we notice that $\Lambda \geq kx_k$ for any $\boldsymbol{x}$ that belongs to the feasible region. Furthermore, $\sum_{i=1}^{k} c_i \geq 0$, implies that $c_k \geq -\sum_{i=1}^{k-1} c_i \geq 0$. Therefore, a sufficient for Equation (17) to hold is that:

$$
\sum_{i=1}^{k-1} \frac{c_i}{\Lambda/k} \leq \sum_{i=1}^{k-1} \frac{c_i}{x_i} - \sum_{i=1}^{k-1} c_i \left(\frac{1}{x_k} - \frac{1}{\Lambda/k}\right),
$$

that can be rewritten as:

$$
\sum_{i=1}^{k-1} c_i \left(\frac{1}{x_i} - \frac{1}{x_k}\right) \geq 0,
$$

or equivalently:

$$
\sum_{i=1}^{k-1} c_i \leq \sum_{i=1}^{k-1} c_i \frac{x_k}{x_i}.
\tag{18}
$$

However, as we shall show, Equation (18) is always satisfied. Indeed, since $\sum_{i=1}^{k-1} c_i \leq 0$ and $x_k \leq x_{k-1}$ we have that:

$$
\sum_{i=1}^{k-1} c_i \leq \sum_{i=1}^{k-1} c_i \frac{x_k}{x_{k-1}} = c_{k-1} \frac{x_k}{x_{k-1}} + \sum_{i=1}^{k-2} c_i \frac{x_k}{x_{k-1}}.
$$

Moreover, since $\sum_{i=1}^{k-1} c_i \leq 0$ and since $x_{k-1} \leq x_{k-2}$,

$$
c_{k-1} \frac{x_k}{x_{k-1}} + \sum_{i=1}^{k-2} c_i \frac{x_k}{x_{k-1}} \leq c_{k-1} \frac{x_k}{x_{k-1}} + c_{k-2} \frac{x_k}{x_{k-2}} + \sum_{i=1}^{k-2} c_i \frac{x_k}{x_{k-2}}.
$$

The properties that $x_i \geq x_{i+1}$ together with the fact that $\sum_{i=1}^{\bar{k}} c_i \leq 0$ for any $\bar{k} < k$ allows to iterate the process, thus concluding the proof. $\qquad \square$

As one can see, applying multiple times Lemma A.2, to the problem we are considering, we obtain a transformed problem that is convex, since the objective function will be composed of summation of convex functions. We will provide additional details on this point later on. We now continue by studying the properties of optimization problems whose objective function satisfies the condition of Lemma A.2. More specifically, the following Lemma allows us to quantify the difference in the optimal solution when changing the budget constraint.

**Lemma A.3** (Budget sensitivity analysis). *Let $c_t \in \mathbb{R}$ for each $t \in \{0, \ldots, T-1\}$. Define $\mathcal{Y} = \{i \in \{0, \ldots, T-1\} : c_i < 0\}$. Let $y \in \mathcal{Y}$, and define $q(y)$ as the smallest integer in $\{y+1, \ldots, T-1\}$ such that $\sum_{i=y}^{q(y)} c_i \geq 0$. Suppose that $q(y)$ is well-defined for any $y \in \mathcal{Y}$.*

*Consider the following optimization problems:*

$$
\begin{aligned}
\min_{\boldsymbol{x}} \quad & \sum_{t=0}^{T-1} \frac{c_t}{x_t} \\
s.t. \quad & \sum_{t=0}^{T-1} x_t = \Lambda \\
& x_t \geq x_{t+1}, \quad \forall t \in \{0, \ldots, T-2\} \\
& x_t \geq 0, \quad \forall t \in \{0, \ldots, T-1\} \\
& x_y = x_{y+1} = \cdots = x_{q(y)}, \quad \forall y \in \mathcal{Y},
\end{aligned}
\tag{19}
$$

*and,*

$$
\begin{aligned}
\min_{\boldsymbol{x}} \quad & \sum_{t=0}^{T-1} \frac{c_t}{x_t} \\
s.t. \quad & \sum_{t=0}^{T-1} x_t = \Lambda' \\
& x_t \geq x_{t+1}, \quad \forall t \in \{0, \ldots, T-2\} \\
& x_t \geq 0, \quad \forall t \in \{0, \ldots, T-1\} \\
& x_y = x_{y+1} = \cdots = x_{q(y)}, \quad \forall y \in \mathcal{Y},
\end{aligned}
\tag{20}
$$

*where $\Lambda, \Lambda' \in \mathbb{R}$ such that $\Lambda \geq T$ and $\Lambda' \geq T$. Define $\alpha = \frac{\Lambda'}{\Lambda}$ and consider $\boldsymbol{x}^*$ an optimal solution of (19). Then, $\alpha \boldsymbol{x}^*$ is an optimal solution of (20).*

*Proof.* First of all, it is important to notice that both problems takes finite and positive value. This directly follow from the equality constraints, together with the fact that $q(y)$ is well-defined for any $y \in \mathcal{Y}$.

We now continue in proving the claim. Proceed by contradiction and suppose that $\alpha \boldsymbol{x}^*$ is not an optimal solution of (20), and let $\bar{\boldsymbol{x}}$ be an optimal solution of (20).

At this point, first of all, we notice that $\alpha \boldsymbol{x}^*$ is a feasible solution of (20). Indeed, we have that $\alpha x_t^* \geq 0$, $\alpha x_t^* \geq \alpha x_{t+1}^*$, $\alpha \sum_{t=0}^{T-1} x_t^* = \alpha \Lambda = \frac{\Lambda'}{\Lambda} \Lambda = \Lambda'$, and for all $y \in \mathcal{Y}$, $\alpha x_y^* = \alpha x_{y+1}^* = \cdots = \alpha x_{q(y)}^*$.

Therefore, we can write:

$$
\sum_{t=0}^{T-1} \frac{c_t}{\bar{x}_t^*} < \sum_{t=0}^{T-1} \frac{c_t}{\alpha x_t^*} = \frac{1}{\alpha} \sum_{t=0}^{T-1} \frac{c_t}{x_t^*}.
$$

From which it follows that:

$$
\sum_{t=0}^{T-1} \frac{c_t}{x_t^*} > \sum_{t=0}^{T-1} \frac{c_t}{\bar{x}_t^*/\alpha}.
$$

However, for similar reasoning w.r.t. to the ones presented above, $\left(\bar{x}_1^*/\alpha, \ldots, \bar{x}_{T-1}^*/\alpha\right)$ is a feasible solution for (19), from which it follows that $\boldsymbol{x}^*$ would not be optimal, which is impossible.

$\square$

The following result, instead, is a technical Lemma that will be used to analyze the error that RIDO accumulates in each optimization round.

**Lemma A.4** (Technical lemma). *Consider a sequence of $K \in \mathbb{N}$ elements $(a_1, \ldots, a_K)$ such that $a_i \in \mathbb{R}$ and $a_i > 0$ for all $i \in [K]$. Then:*

$$\frac{1}{\sum_{i=1}^{K} a_i} \leq \frac{1}{K^2} \sum_{i=1}^{K} \frac{1}{a_i}. \tag{21}$$

*Proof.* We begin with some notation. Consider $K \in \mathbb{N}$ such that $K > 1$, we denote with $\mathcal{V}_K$ the subset of entry-wise strictly positive vectors of $\mathbb{R}^K$, namely:

$$\mathcal{V}_K = \left\{ (a_1, \ldots, a_K) \in \mathbb{R}^K \mid a_i > 0 \text{ for all } i \in [K] \right\}.$$

We now proceed by induction on $K$.

Consider $K = 1$ and $\boldsymbol{v} = (a_1) \in \mathcal{V}_1$. In this case, Equation (21) holds for all $\boldsymbol{v} \in \mathcal{V}_1$ since it reduces to:

$$\frac{1}{a_1} \leq \frac{1}{a_1}.$$

At this point, suppose that:

$$\frac{1}{\sum_{i=1}^{K} a_i} \leq \frac{1}{K^2} \sum_{i=1}^{K} \frac{1}{a_i},$$

holds for $K$ and for all vectors $\boldsymbol{v}_K \in \mathcal{V}_K$, and consider:

$$\frac{1}{\sum_{i=1}^{K+1} a_i} \leq \frac{1}{(K+1)^2} \sum_{i=1}^{K+1} \frac{1}{a_i},$$

for any vector $\boldsymbol{v}_{K+1} = (a_1, \ldots a_{K+1}) \in \mathcal{V}_{K+1}$. At this point, notice that, for all $\boldsymbol{v}_{K+1} \in \mathcal{V}_{K+1}$ the vector $\boldsymbol{v}_{K,-i}$ that is obtained from $\boldsymbol{v}_{K+1}$ by removing the $i$-th component belongs to $\mathcal{V}_K$. At this point, focus on:

$$\frac{1}{(K+1)^2} \sum_{i=1}^{K+1} \frac{1}{a_i} = \frac{1}{(K+1)^2} \left( \sum_{i=1}^{K} \frac{1}{a_i} + \frac{1}{a_{k+1}} \right).$$

Thanks to the inductive hypothesis and some algebraic manipulations, we have that:

$$\frac{1}{(K+1)^2} \left( \sum_{i=1}^{K} \frac{1}{a_i} + \frac{1}{a_{k+1}} \right) = \frac{K^2}{K^2(K+1)^2} \sum_{i=1}^{K} \frac{1}{a_i} + \frac{1}{(K+1)^2} \frac{1}{a_{k+1}}$$

$$\geq \frac{K^2}{(K+1)^2} \left( \frac{1}{\sum_{i=1}^{K} a_i} \right) + \frac{1}{(K+1)^2 a_{K+1}}$$

$$= \frac{K^2}{(K+1)^2} \left( \frac{1}{\sum_{i=1}^{K} a_i} + \frac{1}{a_{K+1} K^2} \right).$$

At this point, we need to show that:

$$\frac{K^2}{(K+1)^2} \left( \frac{1}{\sum_{i=1}^{K} a_i} + \frac{1}{a_{K+1} K^2} \right) \geq \frac{1}{\sum_{i=1}^{K} a_i + a_{K+1}},$$

holds. Set, for the sake of exposition $c = \sum_{i=1}^{K} a_i$ and $d = a_{K+1}$. Then, we can rewrite the previous inequality as:

$$\frac{K^2}{(K+1)^2} \left( \frac{1}{c} + \frac{1}{K^2 d} \right) \geq \frac{1}{c+d}.$$

Rearranging the terms we obtain:

$$\frac{K^2}{(K+1)^2}\left(\frac{K^2 d + c}{cdK^2}\right) \geq \frac{1}{c+d}.$$

Which, in turns, lead to:

$$K^2(K^2 d + c)(c+d) \geq (K+1)^2 cdK^2.$$

Multiplying each term and dividing by $K^2$ leads to:

$$d^2 K^2 - cdK + c^2 \geq 0,$$

which holds for any value of $K > 0$, and $d, c > 0$, thus concluding the proof. $\qquad\square$

Finally, the following Lemma will be used to take into account the rounding effect that comes from solving a continuous relaxation rather than an integer optimization problem.

**Lemma A.5** (Rounding effect error). *Consider a generic $T$-dimensional vector $\boldsymbol{n} = (n_0, \ldots n_{T-1})$ such that $n_i \geq 1$ for all $i \in \{0, \ldots, T-1\}$. Let $q = \sum_{t=0}^{T-1} n_t$, and define $k = q - \sum_{t=0}^{T-1} \lfloor n_t \rfloor$. Consider the vector $\bar{\boldsymbol{n}} = (\bar{n}_0, \ldots \bar{n}_{T-1})$ such that:*

$$\bar{n}_t = \lfloor n_t \rfloor + \mathbf{1}\left\{t < k\right\}.$$

*Define $g(\boldsymbol{n}) = \sum_{t=0}^{T-1} \frac{c_t}{n_t}$ for some vector $\boldsymbol{c} = (c_0, \ldots, c_{T-1})$ with $c_t \in \mathbb{R}$. Then, the following holds:*

$$\sum_{t:c_t \geq 0} \frac{c_t}{\bar{n}_t} \leq 2 \sum_{t:c_t \geq 0} \frac{c_t}{n_t}, \tag{22}$$

$$\sum_{t:c_t \leq 0} \frac{c_t}{\bar{n}_t} \leq \frac{1}{2} \sum_{t:c_t \leq 0} \frac{c_t}{n_t}. \tag{23}$$

*Proof.* We begin by proving Equation (22). First of all, let us notice that:

$$\sum_{t:c_t \geq 0} \frac{c_t}{n_t} \geq \sum_{t:c_t \geq 0} \frac{c_t}{\bar{n}_t + 1} \geq \sum_{t:c_t \geq 0} \frac{c_t}{2\bar{n}_t}, \tag{24}$$

where in the first inequality we have used $c_t \geq 0$ together with $|n_t - \bar{n}_t| \leq 1$, while in the second one we have used $c_t \geq 0$ together with $\bar{n}_t \geq 1$. Equation (22) directly follows from Equation (24).

We continue by proving Equation (23). Similar to Equation (24), it is possible to obtain:

$$\sum_{t:c_t \leq 0} \frac{c_t}{\bar{n}_t} \leq \sum_{t:c_t \leq 0} \frac{c_t}{n_t + 1} \leq \sum_{t:c_t \leq 0} \frac{c_t}{2n_t} = \frac{1}{2} \sum_{t:c_t \leq 0} \frac{c_t}{n_t}, \tag{25}$$

where in the first step we have used $c_t \leq 0$ together with $c_t \leq 0$, while in the second one we have used $c_t \leq 0$ together with $n_t \geq 1$. $\qquad\square$

### A.2.2 RIDO analysis

We begin with some concentration inequalities. We report for completeness the result (Theorem 10) of Maurer and Pontil [2009] that we use to construct confidence intervals around the standard deviation.

**Lemma A.6** (Standard deviation confidence intervals). *Let $n \geq 2$ and consider $X_1, \ldots, X_n$ be i.i.d. random variables with values in $[0, 1]$. Define:*

$$\hat{\sigma} = \sqrt{\frac{1}{n(n-1)} \sum_{i<j} (X_i - X_j)^2}.$$

*Then, for $\delta \in (0, 1)$, with probability at least $1 - \delta$ we have that:*

$$|\hat{\sigma} - \sigma| \leq \sqrt{\frac{2\ln(1/\delta)}{n-1}},$$

*where $\sigma = \mathbb{E}\hat{\sigma}$.*

We then continue with similar results for the estimation of the covariances between random variables.

**Lemma A.7** (Covariance confidence intervals). *Consider $(X_1, Y_1), \ldots (X_n, Y_n)$ i.i.d. random variables with values in $[0, 1]$ sampled from the joint distribution $f_{X,Y}$. Moreover, let $X_{n+1}, \ldots, X_{n+k}$ be $k$ i.i.d. random variables with values in $[0, 1]$ sampled from distribution $f_X = \mathbb{E}_Y [f_{X,Y}]$. Define, for all $i \in [n]$, $Z_i = X_i Y_i$, and let $\hat{z} = \frac{1}{n} \sum_{i=1}^{n} Z_i$, $\hat{x} = \frac{1}{n+k} \sum_{i=1}^{n+k} X_i$ and $\hat{y} = \frac{1}{n} \sum_{i=1}^{n} Y_i$. Then, for $\delta \in (0, 1)$, we have that:*

$$|\mathbb{E}\hat{z} - \mathbb{E}\hat{x}\mathbb{E}\hat{y} - (\hat{z} - \hat{x}\hat{y})| \leq 3\sqrt{\frac{2 \log(6/\delta)}{n}}.$$

*Proof.* By Hoeffding Inequality Boucheron et al. [2003], we have that, for some confidence level $\delta'$, the following holds with probability at least $1 - \delta'$:

$$|\hat{z} - \mathbb{E}\hat{z}| \leq \sqrt{\frac{2 \log(2/\delta')}{n}},$$

and, similarly for $\hat{x}$ and $\hat{y}$. Therefore, by Boole's inequality, it follows that, with probability at least $1 - \delta$, we have that:

$$|\hat{z} - \mathbb{E}\hat{z}| \leq \sqrt{\frac{2 \log(6/\delta)}{n}}, \tag{26}$$

and, similarly, for $\hat{x}$ and $\hat{y}$. [7]

Therefore, with probability at least $1 - \delta$ we have that:

$$\begin{aligned}
|\mathbb{E}\hat{z} - \mathbb{E}\hat{x}\mathbb{E}\hat{y} - (\hat{z} - \hat{x}\hat{y})| &\leq |\mathbb{E}\hat{z} - \hat{z}| + |\mathbb{E}\hat{x}\mathbb{E}\hat{y} - \hat{x}\hat{y}| \\
&\leq \sqrt{\frac{2 \log(6/\delta)}{n}} + |\mathbb{E}\hat{x}\mathbb{E}\hat{y} - \hat{y}\mathbb{E}\hat{x} + \hat{y}\mathbb{E}x - \hat{x}\hat{y}| \\
&\leq \sqrt{\frac{2 \log(6/\delta)}{n}} + |\mathbb{E}\hat{x}(\mathbb{E}\hat{y} - \hat{y})| + |\hat{y}(\mathbb{E}\hat{x} - \hat{x})| \\
&\leq 2\sqrt{\frac{2 \log(6/\delta)}{n}} + |\hat{y}|\sqrt{\frac{2 \log(6/\delta)}{n}} \\
&\leq 3\sqrt{\frac{2 \log(6/\delta)}{n}}.
\end{aligned}$$

where we combined Equation (26) together with triangular inequalities. $\qquad\square$

At this point, before diving into the presentation of the good event under which we will conduct our analysis, we provide a formal definition of our estimators. Consider a generic dataset of trajectories of different lenght. Define, for each $t \in \{0, \ldots, T - 1\}$:

$$\sqrt{\widehat{\mathbb{V}}\text{ar}(R_t)} = \sqrt{\frac{1}{n_t(n_t - 1)} \sum_{1 \leq i < j \leq n} \left(R_t^{(i)} - R_t^{(j)}\right)^2}, \tag{27}$$

where $R_t^{(i)}$ denotes the reward gathered at step $t$ in some trajectory whose length is at least $t + 1$. Moreover, for $t, t'$ such that $t < t'$, define:

$$\widehat{\mathbb{C}}\text{ov}(R_t, R_t') = \frac{1}{n_{t'}} \sum_{i=1}^{n_{t'}} R_t^{(i)} R_{t'}^{(i)} - \left(\frac{1}{n_t} \sum_{i=1}^{n_t} R_t^{(i)}\right) \left(\frac{1}{n_{t'}} \sum_{i=1}^{n_{t'}} R_{t'}^{(i)}\right). \tag{28}$$

**Lemma A.8** (Good event). *The following conditions holds for all phases of RIDO, with probability at least $1 - \delta$:*

$$\left|\sqrt{\mathbb{V}\text{ar}(R_t)} - \sqrt{\widehat{\mathbb{V}}\text{ar}_i(R_t)}\right| \leq \sqrt{\frac{2 \log\left(\frac{6(T+T^2)\Lambda K}{\delta}\right)}{n_t}} = C_{i,t}^{\sigma}. \tag{29}$$

---

[7]For $\hat{x}$ the confidence intervals holds with $\sqrt{\frac{2 \log(6/\delta)}{n+k}}$, which is possibly smaller since $n \leq n + k$.

*and:*

$$\left| \mathbb{C}\text{ov}\left(R_t, R_{t'}\right) - \widehat{\mathbb{C}}\text{ov}_i\left(R_t, R_{t'}\right) \right| \leq 3\sqrt{\frac{2\log\left(\frac{6(T+T^2)\Lambda K}{\delta}\right)}{n_{t'}}} = C_{i,t,t'}^c. \tag{30}$$

*Proof.* The proof follows by combining Lemma A.7 and Lemma A.6, and by taking the union bound over the different time steps, optimization rounds, and possible ways in which the budget can be spent. ☐

At this point, first we show that, with high probability, the objective function of the empirical optimization problem (5) satisfies the same property of the objective function of the original optimization problem (4), i.e., Lemma A.1. Consequently, it holds that the procedure described in the main text in Section 4.1 leads to a transformed convex optimization problem that preserves the optimal solution. For this reason, in the rest of this section, under the good event of Lemma A.8, we assume that RIDO has actually access to an optimal solution of the continuous relaxation of (5), which can be obtained in a computational efficient way by transforming the optimization problem.

**Lemma A.9** (High probability property of the empirical problem). *Let $\beta = \frac{6(T+T^2)\Lambda K}{\delta}$ and consider a generic phase $i$ of Algorithm 1. Define:*

$$\hat{f}_{t,i} = \gamma^{2t}\left(\sqrt{\widehat{\mathbb{V}}\text{ar}_i\left(R_t\right)} + C_{i,t}^\sigma\right)^2 + 2\sum_{t'=t+1}^{T-1} \gamma^{t+t'}\left(\widehat{\mathbb{C}}\text{ov}_i(R_t, R_{t'}) + C_{i,t,t'}^c\right).$$

*Suppose that $\hat{f}_{t,i} < 0$. Then, with probability at least $1 - \delta$, for any $t \in \{0, \ldots, T-2\}$ there exists $\bar{t} > t$ such that $\sum_{j=t}^{\bar{t}} \hat{f}_{j,i} \geq 0$ holds.*

*Proof.* We proceed by contradiction. Suppose that $\hat{f}_{t,i} < 0$ and $\sum_{j=t}^{\bar{t}} \hat{f}_{j,i} < 0$ for all $\bar{t} > t$, and, thus, also for $\bar{t} = T - 1$. Due to Lemma A.8, we have that:

$$\sum_{j=t}^{T-1} \hat{f}_{j,i} \geq \sum_{j=t}^{T-1} \gamma^{2j}\mathbb{V}\text{ar}\left(R_j\right) + 2\sum_{t'=j+1}^{T-1} \gamma^{i+t'}\mathbb{C}\text{ov}\left(R_j, R_{t'}\right) = \mathbb{V}\text{ar}\left(\sum_{j=t}^{T-1} \gamma^j R_j\right),$$

which, however, is always greater or equal than 0, thus leading to a contradiction and concluding the proof. ☐

To analyze the performance of Algorithm 1, we will study the following quantity:

$$\mathbb{V}\text{ar}_{\hat{\boldsymbol{n}}}\left[\hat{J}_{\hat{\boldsymbol{n}}}(\pi)\right] - \mathbb{V}\text{ar}_{\boldsymbol{n}^*}\left[\hat{J}_{\boldsymbol{n}^*}(\pi)\right] = \sum_{t=0}^{T-1} \frac{f_t}{\sum_{i=0}^{K-1} \hat{n}_{t,i}} - \sum_{t=0}^{T-1} \frac{f_t}{n_t^*}. \tag{31}$$

More specifically, by upper bounding Equation (31) we implicitly upper-bound also the variance of the DCS computed by RIDO. At this point, we proceed by analyzing this quantity. The first step, which is presented in the following Lemma, stands in deriving a first errro decomposition on Equation (31).

**Lemma A.10** (Error decomposition). *Let $f_t = \gamma^{2t}\mathbb{V}\text{ar}\left(R_t\right) + 2\sum_{t'=t+1}^{T-1} \gamma^{t+t'}\mathbb{C}\text{ov}\left(R_t, R_{t'}\right)$. Let $y \in \mathcal{Y}$, and define $q(y)$ as the smallest integer in $\{y+1, \ldots, T-1\}$ such that $\sum_{i=y}^{q(y)} f_i \geq 0$. Equation (31) can be upper bounded by:*

$$\frac{2}{K^2}\sum_{i=1}^{K}\sum_{t=0}^{T-1} f_t\left(\frac{1}{\bar{n}_{t,i}} - \frac{1}{\tilde{x}_t^* + 1}\right) + \frac{2}{K^2}\sum_{i=1}^{K}\sum_{t:f_t<0} \frac{|f_t|}{\bar{n}_{t,i}} - \sum_{t:f_t<0} \frac{|f_t|}{\sum_{i=1}^{k} \hat{n}_{t,i}}$$

$$+ \frac{2}{K}\sum_{t=0}^{T-1} \frac{f_t}{\tilde{x}_t^* + 1} - \sum_{t=0}^{T-1} \frac{f_t}{n_t^*}. \tag{32}$$

*where $\bar{n}_i$ is the optimal solution of the continuous relaxation (5), $\hat{n}_i$ is the rounding DCS obtained from $\bar{n}_i$, and $\tilde{x}^*$ is the optimal solution of the following optimization problem:*

$$
\begin{aligned}
\min_{\boldsymbol{x}} \quad & \sum_{i=1}^{K} \frac{f_t}{x_t} \\
s.t. \quad & \sum_{t=0}^{T-1} x_t = b - T \\
& x_i \geq x_{i+1}, \quad \forall i \in \{1, \ldots, K-1\} \\
& x_i \geq 0, \quad \forall i \in \{1, \ldots, K\} \\
& x_y = x_{y+1} = \cdots = x_{q(y)}, \quad \forall y \in \mathcal{Y}.
\end{aligned} \tag{33}
$$

*Proof.* Let us start by analyzing Equation (31):

$$
\mathcal{R} = \sum_{t=0}^{T-1} \frac{f_t}{\sum_{i=1}^{k} \hat{n}_{t,i}} - \sum_{t=0}^{T-1} \frac{f_t}{n_t^*} = \sum_{t:f_t \geq 0} \frac{f_t}{\sum_{i=1}^{k} \hat{n}_{t,i}} + \sum_{t:f_t < 0} \frac{f_t}{\sum_{i=1}^{k} \hat{n}_{t,i}} - \sum_{t=0}^{T-1} \frac{f_t}{n_t^*}.
$$

Due to Lemma A.4, we can upper the previous Equation obtaining:

$$
\begin{aligned}
\mathcal{R} &\leq \frac{1}{K^2} \sum_{i=1}^{K} \sum_{t:f_t \geq 0} \frac{f_t}{\hat{n}_{t,i}} + \sum_{t:f_t < 0} \frac{f_t}{\sum_{i=1}^{k} \hat{n}_{t,i}} - \sum_{t=0}^{T-1} \frac{f_t}{n_t^*} \\
&\leq \frac{2}{K^2} \sum_{i=1}^{K} \sum_{t:f_t \geq 0} \frac{f_t}{\bar{n}_{t,i}} + \sum_{t:f_t < 0} \frac{f_t}{\sum_{i=1}^{k} \hat{n}_{t,i}} - \sum_{t=0}^{T-1} \frac{f_t}{n_t^*} \\
&= \frac{2}{K^2} \sum_{i=1}^{K} \sum_{t=0}^{T-1} \frac{f_t}{\bar{n}_{t,i}} - \frac{2}{K^2} \sum_{i=1}^{K} \sum_{t:f_t < 0} \frac{f_t}{\bar{n}_{t,i}} + \sum_{t:f_t < 0} \frac{f_t}{\sum_{i=1}^{k} \hat{n}_{t,i}} - \sum_{t=0}^{T-1} \frac{f_t}{n_t^*}.
\end{aligned}
$$

where in the first step, we have used Lemma A.4, in the second one Lemma A.5, and in the third one we have added and subtracted $\frac{2}{K^2} \sum_{i=1}^{K} \sum_{t:f_t < 0} \frac{f_t}{\bar{n}_{t,i}}$. The proof directly follows by adding and subtracting:

$$
\frac{2}{K^2} \sum_{i=1}^{K} \sum_{t=0}^{T-1} \frac{f_t}{\tilde{x}_t^* + 1} = \frac{2}{K} \sum_{t=0}^{T-1} \frac{f_t}{\tilde{x}_t^* + 1}.
$$

$\square$

At this point, the following Lemma provides an upper bound on Equation (32). More specifically, we focus on the first term, that is:

$$
\frac{2}{K^2} \sum_{i=1}^{K} \sum_{t=0}^{T-1} f_t \left( \frac{1}{\bar{n}_{t,i}} - \frac{1}{\tilde{x}_t^* + 1} \right),
$$

which can be interpreted as the error that RIDO cumulates in its rounds.

**Lemma A.11** (Cumulative error). *Let $\beta = \frac{6(T+T^2)\Lambda K}{\delta}$. Let $y \in \mathcal{Y}$, and define $q(y)$ as the smallest integer in $\{y+1, \ldots, T-1\}$ such that $\sum_{i=y}^{q(y)} f_i \geq 0$. Let $\tilde{x}^*$ be the solution of the following optimization problem:*

$$
\begin{aligned}
\min_{\boldsymbol{x}} \quad & \sum_{i=1}^{K} \frac{f_t}{x_t} \\
s.t. \quad & \sum_{t=0}^{T-1} x_t = b - T \\
& x_i \geq x_{i+1}, \quad \forall i \in \{1, \ldots, K-1\} \\
& x_i \geq 0, \quad \forall i \in \{1, \ldots, K\} \\
& x_y = x_{y+1} = \cdots = x_{q(y)}, \quad \forall y \in \mathcal{Y},
\end{aligned} \tag{34}
$$

and let $\bar{\boldsymbol{n}}_i$ be the solution of the continuous relaxation of (5) during phase $i$. Then, with probability at least $1 - \delta$, the following holds:

$$\frac{2}{K^2} \sum_{i=1}^{K} \sum_{t=0}^{T-1} f_t \left( \frac{1}{\bar{n}_{t,i}} - \frac{1}{\tilde{x}_t^* + 1} \right) \leq \frac{192}{K^{\frac{3}{2}}} \log \left( \frac{6(T + T^2)\Lambda K}{\delta} \right) \left( \sum_{t=0}^{T-1} \gamma^t \right)^2 \tag{35}$$

*Proof.* The proof is split into 3 parts. In particular, we will analyze:

$$\frac{2}{K^2} \sum_{i=1}^{K} \sum_{t=0}^{T-1} f_t \left( \frac{1}{\bar{n}_{t,i}} - \frac{1}{\tilde{x}_t^* + 1} \right)$$

first for a generic phase $i > 1$, then for $i = 1$, and finally we will put everything together.

Let us start by considering a generic phase $i > 1$, and focus on:

$$\sum_{t=0}^{T-1} f_t \left( \frac{1}{\bar{n}_{t,i}} - \frac{1}{\tilde{x}_t^* + 1} \right) \tag{36}$$

First of all, focus on $\sum_{t=0}^{T-1} \frac{f_t}{\tilde{x}_t+1}$. Let us define $\tilde{\boldsymbol{g}} = (\tilde{g}_0, \ldots, \tilde{g}_{T-1})$ as the solution to the following optimization problem:

$$\begin{aligned} \min_{\boldsymbol{g}} \quad & \sum_{t=0}^{T-1} \frac{f_t}{g_t} \\ \text{s.t.} \quad & \sum_{t=0}^{T-1} g_t = b \\ & g_t \geq g_{t+1}, \quad \forall t \in \{0, \ldots, T-2\} \\ & g_t \geq 1, \quad \forall t \in \{0, \ldots, T-1\}. \end{aligned} \tag{37}$$

It is easy to see that: [8]

$$\sum_{t=0}^{T-1} \frac{f_t}{\tilde{g}_t} \leq \sum_{t=0}^{T-1} \frac{f_t}{\tilde{x}_t^* + 1}.$$

Plugging this result into Equation (36) leads to:

$$\sum_{t=0}^{T-1} f_t \left( \frac{1}{\bar{n}_{t,i}} - \frac{1}{\tilde{x}_t^* + 1} \right) \leq \sum_{t=0}^{T-1} f_t \left( \frac{1}{\bar{n}_{t,i}} - \frac{1}{\tilde{g}_t^*} \right). \tag{38}$$

Due to Lemma A.8, with probability at least $1 - \delta$, we can further upper bound Equation (38) with:

$$\sum_{t=0}^{T-1} \frac{\gamma^{2t} \left( \sqrt{\widehat{\mathbb{V}}\mathrm{ar}_i(R_t)} + \mathbf{C}_{i,t}^{\sigma} \right)^2}{\bar{n}_{t,i}} + 2 \sum_{t=0}^{T-2} \sum_{t'=t+1}^{T-1} \frac{\gamma^{t+t'} \left( \widehat{\mathbb{C}}\mathrm{ov}(R_t, R_{t'}) + \mathbf{C}_{i,t,t'}^c \right)}{\bar{n}_{t,i}} - \sum_{t=0}^{T-1} \frac{f_t}{\tilde{g}_t},$$

However, since $\tilde{\boldsymbol{g}}$ is a feasible solution of the continuous relaxation of (5), and since $\bar{n}_{t,i}$ is the minimizer of the continuous relaxation of (5) at phase $i$, we can further bound the previous equation with:

$$\sum_{t=0}^{T-1} \frac{\gamma^{2t} \left( \sqrt{\widehat{\mathbb{V}}\mathrm{ar}_i(R_t)} + \mathbf{C}_{i,t}^{\sigma} \right)^2}{\tilde{g}_t} + 2 \sum_{t=0}^{T-2} \sum_{t'=t+1}^{T-1} \frac{\gamma^{t+t'} \left( \widehat{\mathbb{C}}\mathrm{ov}(R_t, R_{t'}) + \mathbf{C}_{i,t,t'}^c \right)}{\tilde{g}_t} - \sum_{t=0}^{T-1} \frac{f_t}{\tilde{g}_t}.$$

---

[8]This step follows by considering the optimization problem that defines $\tilde{\boldsymbol{g}}$. With a change of variable $g_t = x_t + 1$, we can notice that $\tilde{x}_t^* + 1$ is indeed a feasible solution of the same optimization problem. Furthermore, notice that due to Lemma A.2, we can neglect the constraints on $y$.

Moreover, due to Lemma A.8, we can further upper-bound the previous Equation with:

$$\sum_{t=0}^{T-1} \frac{\gamma^{2t}\left(\sqrt{\mathbb{V}\text{ar}\left[R_t\right]}+2\mathbf{C}_{i,t}^{\sigma}\right)^2}{\tilde{g}_t} + 2\sum_{t=0}^{T-2}\sum_{t'=t+1}^{T-1} \frac{\gamma^{t+t'}\left(\mathbb{C}\text{ov}(R_t,R_{t'})+2\mathbf{C}_{i,t,t'}^{c}\right)}{\tilde{g}_t} - \sum_{t=0}^{T-1} \frac{f_t}{\tilde{g}_t},$$

Let us now focus on:

$$\sum_{t=0}^{T-1} \frac{\gamma^{2t}\left(\sqrt{\mathbb{V}\text{ar}\left[R_t\right]}+2\mathbf{C}_{i,t}^{\sigma}\right)^2}{\tilde{g}_t} + 2\sum_{t=0}^{T-2}\sum_{t'=t+1}^{T-1} \frac{\gamma^{t+t'}\left(\mathbb{C}\text{ov}(R_t,R_{t'})+2\mathbf{C}_{i,t,t'}^{c}\right)}{\tilde{g}_t}. \qquad (39)$$

Equation (39) can be decomposed into:

$$\sum_{t=0}^{T-1} \frac{\gamma^{2t}4\sqrt{\mathbb{V}\text{ar}\left[R_t\right]}\mathbf{C}_{i,t}^{\sigma}+4\gamma^{2t}\left(\mathbf{C}_{i,t}^{\sigma}\right)^2}{\tilde{g}_t} + 4\sum_{t=0}^{T-2}\sum_{t'=t+1}^{T-1} \frac{\gamma^{t+t'}\mathbf{C}_{i,t,t'}^{c}}{\tilde{g}_t} \qquad (40)$$

and,

$$\sum_{t=0}^{T-1} \frac{f_t}{\tilde{g}_t}. \qquad (41)$$

Thus leading to:

$$\mathcal{R} \le \sum_{t=0}^{T-1} \frac{\gamma^{2t}4\sqrt{\mathbb{V}\text{ar}\left[R_t\right]}\mathbf{C}_{i,t}^{\sigma}+4\gamma^{2t}\left(\mathbf{C}_{i,t}^{\sigma}\right)^2}{\tilde{g}_t} + 4\sum_{t=0}^{T-2}\sum_{t'=t+1}^{T-1} \frac{\gamma^{t+t'}\mathbf{C}_{i,t,t'}^{c}}{\tilde{g}_t} \qquad (42)$$

We now proceed by bounding each term in Equation (42). Define, for brevity $h_{i,t} = \sum_{j=0}^{i-1} \hat{n}_{t,j}$. Let us first focus on:

$$\sum_{t=0}^{T-1} \frac{4\gamma^{2t}\sqrt{\mathbb{V}\text{ar}\left[R_t\right]}\mathbf{C}_{i,t}^{\sigma}}{\tilde{g}_t} \le \sum_{t=0}^{T-1} \frac{4\gamma^{2t}}{\tilde{g}_t}\sqrt{\frac{2\log\left(\frac{2(T+T^2)\Lambda K}{\delta}\right)}{h_{i-1,t}}}$$

$$\le 8\sqrt{\log\left(\frac{2(T+T^2)\Lambda K}{\delta}\right)}\sum_{t=0}^{T-1} \frac{\gamma^{2t}}{\tilde{g}_t\sqrt{i-1}}$$

$$\le 16\sqrt{\log\left(\frac{2(T+T^2)\Lambda K}{\delta}\right)}\sum_{t=0}^{T-1} \frac{\gamma^{2t}}{\sqrt{i}}.$$

where the first step follows from the definition of the confidence intervals, together with the fact that rewards are bounded in $[0, 1]$, the second one by recalling that $h_{i-1,t} = \sum_{j=1}^{i-1} \hat{n}_{t,j} \ge i-1$, and the third one by noticing that $\sqrt{i} \le 2\sqrt{i-1}$.

Similary, for what concerns:

$$\sum_{t=0}^{T-1} \frac{4\gamma^{2t}\left(\mathbf{C}_{i,t}^{\sigma}\right)^2}{\tilde{g}_t} \le \sum_{t=0}^{T-1} \frac{4\gamma^{2t}}{\tilde{g}_t}\frac{2\log\left(\frac{2(T+T^2)\Lambda K}{\delta}\right)}{h_{i-1,t}}$$

$$\le 8\log\left(\frac{2(T+T^2)\Lambda K}{\delta}\right)\sum_{t=0}^{T-1} \frac{\gamma^{2t}}{\tilde{g}_t(i-1)}$$

$$\le 16\log\left(\frac{2(T+T^2)\Lambda K}{\delta}\right)\sum_{t=0}^{T-1} \frac{\gamma^{2t}}{\sqrt{i}}$$

Finally, what is left is:

$$4\sum_{t=0}^{T-2}\sum_{t'=t+1}^{T-1} \frac{\gamma^{t+t'}\mathbf{C}_{i,t,t'}^{c}}{\tilde{g}_t} \le 24\sqrt{\log\left(\frac{6(T+T^2)\Lambda K}{\delta}\right)}\sum_{t=0}^{T-2}\sum_{t'=t+1}^{T-1} \frac{\gamma^{t+t'}}{\tilde{g}_t\sqrt{i-1}}$$

$$\le 48\sqrt{\log\left(\frac{6(T+T^2)\Lambda K}{\delta}\right)}\sum_{t=0}^{T-2}\sum_{t'=t+1}^{T-1} \frac{\gamma^{t+t'}}{\sqrt{i}}$$

For what concerns phase $i = 1$, instead, the budget is allocated uniformly. Therefore, we have that:

$$\sum_{t=0}^{T-1} f_t \left( \frac{1}{b/T} - \frac{1}{\tilde{g}_t} \right) \leq \sum_{t=0}^{T-1} \frac{f_t}{b/T} \leq \sum_{t=0}^{T-1} f_t \leq \left( \sum_{t=0}^{T-1} \gamma^t \right)^2$$

At this point, plugging these results into Equation (38) leads to:

$$\frac{2}{K^2} \sum_{i=1}^{K} \left( 48 \log \left( \frac{6(T + T^2)\Lambda K}{\delta} \right) \left( \sum_{t=0}^{T-1} \gamma^t \right)^2 \right) \frac{1}{\sqrt{i}} \tag{43}$$

To conclude the proof, we notice that $\sum_{i=1}^{n} \frac{1}{\sqrt{i}} \leq 2\sqrt{n} - 1$, thus leading to:

$$\frac{192}{K^{\frac{3}{2}}} \log \left( \frac{6(T + T^2)\Lambda K}{\delta} \right) \left( \sum_{t=0}^{T-1} \gamma^t \right)^2$$

which is the desired result. $\qquad\qquad\qquad\qquad\qquad\qquad\qquad\qquad\qquad\qquad\qquad\qquad$ □

We now continue by upper bounding another term of Equation (31), that is:

$$\frac{2}{K} \sum_{t=0}^{T-1} \frac{f_t}{\tilde{x}_t^* + 1} - \sum_{t=0}^{T-1} \frac{f_t}{n_t^*}.$$

**Lemma A.12** (Exploration error). *Let $y \in \mathcal{Y}$, and define $q(y)$ as the smallest integer in $\{y + 1, \ldots, T - 1\}$ such that $\sum_{i=y}^{q(y)} f_i \geq 0$. Let $\tilde{x}^*$ be the solution of the following optimization problem:*

$$\min_{x} \quad \sum_{i=1}^{K} \frac{f_t}{x_t}$$

$$\text{s.t.} \quad \sum_{t=0}^{T-1} x_t = b - T \tag{44}$$

$$x_i \geq x_{i+1}, \quad \forall i \in \{1, \ldots, K - 1\}$$

$$x_i \geq 0, \quad \forall i \in \{1, \ldots, K\}$$

$$x_y = x_{y+1} = \cdots = x_{q(y)}, \quad \forall y \in \mathcal{Y}.$$

*Then,*

$$\frac{2}{K} \sum_{t=0}^{T-1} \frac{f_t}{\tilde{x}_t^* + 1} - \sum_{t=0}^{T-1} \frac{f_t}{n_t^*} \leq \frac{c+1}{c-1} \sum_{t=0}^{T-1} \frac{f_t}{x_t^*},$$

*where $c$ is such that $cT = b$, and $x^*$ is the solution of the following optimization problem:*

$$\min_{x} \quad \sum_{i=1}^{K} \frac{f_t}{x_t}$$

$$\text{s.t.} \quad \sum_{t=0}^{T-1} x_t = \Lambda \tag{45}$$

$$x_i \geq x_{i+1}, \quad \forall i \in \{1, \ldots, K - 1\}$$

$$x_i \geq 0, \quad \forall i \in \{1, \ldots, K\}$$

$$x_y = x_{y+1} = \cdots = x_{q(y)}, \quad \forall y \in \mathcal{Y}.$$

*Proof.* Consider the following optimization problem:

$$\min_{\boldsymbol{x}} \quad \sum_{i=1}^{K} \frac{f_t}{x_t}$$

$$\text{s.t.} \quad \sum_{t=0}^{T-1} x_t = K(b - T) \tag{46}$$

$$x_i \geq x_{i+1}, \quad \forall i \in \{1, \ldots, K-1\}$$
$$x_i \geq 0, \quad \forall i \in \{1, \ldots, K\},$$
$$x_y = x_{y+1} = \cdots = x_{q(y)}, \quad \forall y \in \mathcal{Y}$$

and let $\bar{\boldsymbol{x}}^*$ be its optimal solution. Then, due to Lemma A.3, $K\tilde{\boldsymbol{x}}^* = \bar{\boldsymbol{x}}^*$. Therefore, we have that:

$$\frac{2}{K} \sum_{t=0}^{T-1} \frac{f_t}{\tilde{x}_t^* + 1} - \sum_{t=0}^{T-1} \frac{f_t}{n_t^*} = 2 \sum_{t=0}^{T-1} \frac{f_t}{\bar{x}_t^* + K} - \sum_{t=0}^{T-1} \frac{f_t}{n_t^*}$$

Furthermore, due to the fact that $\bar{x}_y^* = \bar{x}_{y+1}^* = \cdots = \bar{x}_{q(y)}^*$ for all $y \in \mathcal{Y}$, we have that:

$$2 \sum_{t=0}^{T-1} \frac{f_t}{\bar{x}_t^* + K} - \sum_{t=0}^{T-1} \frac{f_t}{n_t^*} \leq 2 \sum_{t=0}^{T-1} \frac{f_t}{\bar{x}_t^*} - \sum_{t=0}^{T-1} \frac{f_t}{n_t^*}$$

At this point, we proceed by lower bounding:

$$\sum_{t=0}^{T-1} \frac{f_t}{n_t^*}.$$

More specifically, consider the following optimization problem:

$$\min_{\boldsymbol{x}} \quad \sum_{i=1}^{K} \frac{f_t}{x_t}$$

$$\text{s.t.} \quad \sum_{t=0}^{T-1} x_t = \Lambda \tag{47}$$

$$x_i \geq x_{i+1}, \quad \forall i \in \{1, \ldots, K-1\}$$
$$x_i \geq 0, \quad \forall i \in \{1, \ldots, K\},$$
$$x_y = x_{y+1} = \cdots = x_{q(y)}, \quad \forall y \in \mathcal{Y},$$

and let $\boldsymbol{x}^*$ be its optimal solution. Then, we have that:

$$\sum_{t=0}^{T-1} \frac{f_t}{n_t^*} \geq \sum_{t=0}^{T-1} \frac{f_t}{x_t^*}. \tag{48}$$

To prove Equation (48), it is sufficient to drop the integer constraints from the (4), then, due to Lemma A.2, we can impose the equality constraints on the resulting optimization problem, and finally, we enlarge the feasible region by setting the constraints $x_i \geq 0$.

At this point, we have:

$$\frac{2}{K} \sum_{t=0}^{T-1} \frac{f_t}{\tilde{x}_t^* + 1} - \sum_{t=0}^{T-1} \frac{f_t}{n_t^*} \leq 2 \sum_{t=0}^{T-1} \frac{f_t}{\bar{x}_t^*} - \sum_{t=0}^{T-1} \frac{f_t}{x_t^*}. \tag{49}$$

By Lemma A.3, we have that:

$$\bar{x}_t^* = \frac{K(b-T)}{\Lambda} x_t^* = \frac{K(b-T)}{Kb} x_t^* = \frac{b-T}{b} x_t^* = \frac{cT - T}{cT} x_t^* = \frac{c-1}{c} x_t^*$$

Plugging this result into Equation (49), we obtain:

$$\frac{2}{K}\sum_{t=0}^{T-1}\frac{f_t}{\tilde{x}_t^*+1} - \sum_{t=0}^{T-1}\frac{f_t}{n_t^*} \leq \left(\frac{2c}{c-1}-1\right)\sum_{t=0}^{T-1}\frac{f_t}{x_t^*}$$

$$= \frac{c+1}{c-1}\sum_{t=0}^{T-1}\frac{f_t}{x_t^*}.$$

□

At this point, we are ready to prove Theorem 4.1.

**Theorem 4.1.** *Let $n^*$ be the optimal solution of problem* (4), *$f_t$ as in Equation* (3), *$b \geq 2T$ and* $\beta = \frac{6(T+T^2)\Lambda K}{\delta}$. *Consider the DCS $\hat{n}$ computed by Algorithm 1. Then, with probability at least* $1 - \delta$ *it holds that:*

$$\mathbb{V}\mathrm{ar}_{\hat{n}}\left[\hat{J}_{\hat{n}}(\pi)\right] \leq 192\left(\frac{b}{\Lambda}\right)^{\frac{3}{2}}\log\left(\beta\right)\left(\sum_{t=0}^{T-1}\gamma^t\right)^2 + 4\mathbb{V}\mathrm{ar}_{n^*}\left[\hat{J}_{n^*}(\pi)\right] + \frac{2b}{\Lambda}\sum_{t:f_t<0}|f_t|. \quad (8)$$

*Proof.* From Lemma A.10, we can upper bound Equation (31) with:

$$\frac{2}{K^2}\sum_{i=1}^{K}\sum_{t=0}^{T-1}f_t\left(\frac{1}{\bar{n}_{t,i}}-\frac{1}{\tilde{x}_t^*+1}\right) + \frac{2}{K^2}\sum_{i=1}^{K}\sum_{t:f_t<0}\frac{|f_t|}{\bar{n}_{t,i}} - \sum_{t:f_t<0}\frac{|f_t|}{\sum_{i=1}^{k}\hat{n}_{t,i}}$$

$$+ \frac{2}{K}\sum_{t=0}^{T-1}\frac{f_t}{\tilde{x}_t^*+1} - \sum_{t=0}^{T-1}\frac{f_t}{n_t^*}. \quad (50)$$

At this point, we notice that:

$$\frac{2}{K^2}\sum_{i=1}^{K}\sum_{t:f_t<0}\frac{|f_t|}{\bar{n}_{t,i}} - \sum_{t:f_t<0}\frac{|f_t|}{\sum_{i=1}^{k}\hat{n}_{t,i}} \leq \frac{2}{K}\sum_{t:f_t<0}f_t$$

Plugging this result into Equation (50), we obtain:

$$\frac{2}{K^2}\sum_{i=1}^{K}\sum_{t=0}^{T-1}f_t\left(\frac{1}{\bar{n}_{t,i}}-\frac{1}{\tilde{x}_t^*+1}\right) + \frac{2}{K}\sum_{t:f_t<0}|f_t| + \frac{2}{K}\sum_{t=0}^{T-1}\frac{f_t}{\tilde{x}_t^*+1} - \sum_{t=0}^{T-1}\frac{f_t}{n_t^*}. \quad (51)$$

Due to Lemma A.11, this can be further upper-bounded with:

$$192\left(\frac{b}{\Lambda}\right)^{\frac{3}{2}}\log\left(\frac{6(T+T^2)\Lambda K}{\delta}\right)\left(\sum_{t=0}^{T-1}\gamma^t\right)^2 + \frac{2}{K}\sum_{t:f_t<0}|f_t| + \frac{2}{K}\sum_{t=0}^{T-1}\frac{f_t}{\tilde{x}_t^*+1} - \sum_{t=0}^{T-1}\frac{f_t}{n_t^*}.$$

Moreover, due to Lemma A.12, we can further bound the previous Equation with:

$$192\left(\frac{b}{\Lambda}\right)^{\frac{3}{2}}\log\left(\frac{6(T+T^2)\Lambda K}{\delta}\right)\left(\sum_{t=0}^{T-1}\gamma^t\right)^2 + \frac{2}{K}\sum_{t:f_t<0}|f_t| + \frac{c+1}{c-1}\sum_{t=0}^{T-1}\frac{f_t}{x_t^*}, \quad (52)$$

where $x_t^*$ is the solution of the following optimization problem:

$$\min_{\boldsymbol{x}} \quad \sum_{i=1}^{K}\frac{f_t}{x_t}$$

$$\text{s.t.} \quad \sum_{t=0}^{T-1}x_t = \Lambda$$

$$x_i \geq x_{i+1}, \quad \forall i \in \{1,\ldots,K-1\}$$

$$x_i \geq 0, \quad \forall i \in \{1,\ldots,K\},$$

$$x_y = x_{y+1} = \cdots = x_{q(y)}, \quad \forall y \in \mathcal{Y}, \quad (53)$$

Moreover, since:

$$\sum_{t=0}^{T-1} \frac{f_t}{n_t^*} \geq \sum_{t=0}^{T_1} \frac{f_t}{x_t^*},$$

Equation 52 reduces to:

$$192 \left(\frac{b}{\Lambda}\right)^{\frac{3}{2}} \log\left(\frac{6(T+T^2)\Lambda K}{\delta}\right) \left(\sum_{t=0}^{T-1} \gamma^t\right)^2 + \frac{2}{K} \sum_{t:f_t<0} |f_t| + \frac{c+1}{c-1} \sum_{t=0}^{T-1} \frac{f_t}{n_t^*},$$

At this point, the results follows by noticing that:

$$\frac{c+1}{c-1} = \frac{b+T}{b-T} \leq 3,$$

and, by isolating $\mathbb{V}\mathrm{ar}_{\hat{n}}\left[\hat{J}_{\hat{n}}(\pi)\right]$ in Equation (31). $\qquad\square$

### A.3 Additional Technical Details

In this section, we provide additional techincal details that have been mentioned in the main text. More specifically, we provide (i) a formal description of the transformation between optimization problems and how we applied this technique in RIDO, (ii) difficulties in deriving closed-form solutions for the optimization problems of interest, (iii) a formal statement and proof of Equation (9), (iv) and theoretical evidence for the sub-optimality of non-adaptive methods whose variance cannot scale with the variance of the optimal DCS).

#### A.3.1 Additional Details on solving the empirical optimization problem

We begin with a more in-depth discussion of the transformation between optimization problems. Let $c_t \in \mathbb{R}$ for each $t \in \{0, \ldots, T-1\}$, and define $\mathcal{Y} = \{i \in \{0, \ldots, T-1\} : c_i < 0\}$. Let $y \in \mathcal{Y}$, and define $q(y)$ as the smallest integer in $\{y+1, \ldots, T-1\}$ such that $\sum_{i=y}^{q(y)} c_i \geq 0$. Due to Lemma A.1 we know that, if $(c_0, \ldots, c_{T-1}) = (f_0, \ldots, f_{T-1})$, then $q(y)$ is always well-defined. At this point, consider the continuous relaxation of the original optimization problem, namely:

$$
\begin{aligned}
\min_{\boldsymbol{n}} \quad & \sum_{t=0}^{T-1} \frac{1}{n_t}\left(\gamma^{2t}\mathbb{V}\mathrm{ar}(R_t) + 2\sum_{t'=t+1}^{T-1} \gamma^{t+t'}\mathbb{C}\mathrm{ov}(R_t, R_{t'})\right) \\
\text{s.t.} \quad & \sum_{t=0}^{T-1} n_t = \Lambda \\
& n_t \geq n_{t+1}, \quad \forall t \in \{0, \ldots, T-2\} \\
& n_t \geq 1, \quad \forall t \in \{0, \ldots, T-1\}.
\end{aligned}
\tag{54}
$$

Due to Lemma A.1 and Lemma A.2, we know that the following optimization problem:

$$
\begin{aligned}
\min_{\boldsymbol{n}} \quad & \sum_{t=0}^{T-1} \frac{1}{n_t}\left(\gamma^{2t}\mathbb{V}\mathrm{ar}(R_t) + 2\sum_{t'=t+1}^{T-1} \gamma^{t+t'}\mathbb{C}\mathrm{ov}(R_t, R_{t'})\right) \\
\text{s.t.} \quad & \sum_{t=0}^{T-1} n_t = \Lambda \\
& n_t \geq n_{t+1}, \quad \forall t \in \{0, \ldots, T-2\} \\
& n_t \geq 1, \quad \forall t \in \{0, \ldots, T-1\} \\
& n_y = n_{y+1} = \cdots = n_{q(y)}, \quad \forall y \in \mathcal{Y},
\end{aligned}
\tag{55}
$$

has the same optimal solution of (55). At this point, to define the transformed problem it is sufficient to introduce additional variables $y_i$ for any contiguous timesteps where $n_i = n_{i+1} = \cdots = n_{i+k}$

holds for some integers $i, k$.[9] The optimization variables $n_i, n_{i+1}, \ldots, n_{i+k+1}$ will be substituted with $y_i$. The objective function will be modified accordingly, namely:

$$\frac{f_i}{n_i} + \cdots + \frac{f_{i+k}}{n_{i+k}},$$

is replaced with:

$$\frac{f_y + \cdots + f_{q(y)}}{y_i}. \tag{56}$$

Consequently, any numerator in the resulting objective function of the transformed problem will be greater or equal than $0$. It is easy to verify that, in this case, the resulting objective function is convex in the considered optimization domain. Finally, as a last remark, we notice that the constraint $\sum_{t=0}^{T-1} n_t = \Lambda$ needs to be modified. More specifically, if $y_i$ substitutes $l_i$ variables, then its contribution within the budget constraint summation will be given by $y_i l_i$.

As discussed in Section 4.1, in RIDO we adopt a procedure that is inspired by the aforementioned theoretical properties of the continuous relaxation of the optimization problem (4). Nevertheless, it has to be noticed that a modification needs to be taken into account when replacing exact quantities (i.e., $f_t$) with their estimation and exploration bonuses (which, in the following, we refer to as $\hat{f}_t$ for brevity). More specifically, in general, contrary to what highlighted in Lemma A.1 for the original objective function, when using $\hat{f}_t$ it might happen that $q(y)$ is not well-defined for every possible $y$. Indeed, due to the noise that is present in the estimation process, there might exists $\bar{t}$ such that $\hat{f}_{\bar{t}} < 0$ and $\sum_{t=\bar{t}}^{t'} \hat{f}_t < 0$ for all $t' > \bar{t}$. Whenever this condition is verified, we adopt the following heuristic to make the computation tractable. If $\bar{t} = 0$, then we just set the DCS of the current mini-batch to the uniform-in-the-horizon one. When $\bar{t} \neq 0$, instead, we group together $n_{\bar{t}}, \ldots, n_{T-1}$ and we introduce a new variable $y$ that will divide, in the objective function, $\hat{f}_{\bar{t}-1}$. As a final remark, however, we notice that these modifications do not impact on the theoretical properties of RIDO. Indeed, Lemma A.9, shows that, with probability at least $1 - \delta$, the aforementioned ill-conditions do not happen. As a consequence, we can study the high-probability behavior of RIDO assuming access to the solution of the transformed optimization problem discussed at the beginning of this section (that preserves the optimal solution of the continuous relaxation of (5)).

### A.3.2   On closed-form solutions

We now continue by discussing the closed-form solutions of the optimization problems of interests. First of all, optimization problems (4) and (5) are integer and non-linear problems. Even neglecting the non-linear dependency on $n$, we remark that solving integer and linear problem is NP-hard. At this point, one might resort to study their continuous relaxations. In the following, we focus on the continuous relaxation of (4) (indeed, as noticed at the end of the previous section, the continuous relaxation of (5) requires additional effort). As mentioned above, whenever $f_t < 0$ holds for some $t \in \{0, \ldots, T - 1\}$, the continuous relaxation of $f_t$ is non-convex. Nevertheless, from Lemma A.2, we know that we can always derive an equivalent convex problem (where the numerator in the objective function is always greater or equal than $0$) that preserves the optimal solution. For

---

[9]More precisely, we notice that $n_i = n_{i+1} = \cdots = n_{i+k}$ might involve multiple constraints in the formulation of (55). In this sense, we need to refer to the largest intervals in which these constraints are enforced, otherwise we might introduce multiple variables that refer to the same original optimization variable.

this reason, we now report the KKT conditions under the assumption that $f_t \geq 0$ holds for all $t \in \{0, \ldots, T-1\}$.[10]

$$
\begin{cases}
-\frac{f_t}{n_t^2} + \eta - \xi_t - \mu_t \mathbf{1}\{t < T-1\} + \mu_{t-1}\mathbf{1}\{t > 0\} = 0 & \forall t \in \{0, \ldots, T-1\} \\
\xi_t(1 - n_t) = 0 & \forall t \in \{0, \ldots, T-1\} \\
\mu_t(n_{t+1} - n_t) = 0 & \forall t \in \{0, \ldots, T-2\} \\
\eta(\sum_{t=0}^{T-1} n_t - \Lambda) = 0 & \\
\sum_{t=0}^{T-1} n_t - \Lambda = 0 & \\
\mu_t \geq 0 & \forall t \in \{0, \ldots, T-2\} \\
\xi_t \geq 0 & \forall t \in \{0, \ldots, T-1\}
\end{cases}
\tag{57}
$$

At this point, we notice that a similar problem has been solved in Poiani et al. [2023] for deriving a closed-form solutions that minimizes confidence intervals around the return estimator. In that situation, however, the constraints $n_t \geq n_{t+1}$ were not present since they were automatically satisfied by any optimal solution (and, consequently, they were removed from the optimization problem of interest). The main challenge in our setting is, indeed, the presence of $\mu_t(n_{t+1} - n_t)$, together with the terms related to $\mu_t$ in the first Equation of (57). These additional components within (57) prevented us to derive a closed-form solutions of the continuous relaxation (4) (and, (5)).

### A.3.3 Proof of Equation (9)

We now continue with providing a formal statement and proof of Equation (9).

**Corollary A.13.** *Suppose that $f_t \geq 0$ for all $t \in \{0, \ldots, T-1\}$, and $\sum_{t=0}^{T-1} f_t > 0$. Let:*

$$
\Lambda \geq \Lambda_0 := \left( 55296 \frac{b^{\frac{3}{2}}}{\delta} \frac{\left(\sum_{t=0}^{T-1} \gamma^t\right)^2}{\sum_{t=0}^{T-1} f_t} \right)^3.
\tag{58}
$$

*Let $\beta = \frac{6(T+T^2)\Lambda K}{\delta}$. Then, with probability at least $1 - \delta$, it holds that:*

$$
\mathbb{V}\mathrm{ar}_{\hat{\boldsymbol{n}}}\left[\hat{J}_{\hat{\boldsymbol{n}}}(\pi)\right] \leq 5\mathbb{V}\mathrm{ar}_{\boldsymbol{n}^*}\left[\hat{J}_{\boldsymbol{n}^*}(\pi)\right].
\tag{59}
$$

*Proof.* The proof follows by analyzing, under the condition provided by Equation (58), the upper bound provided in Theorem 4.1. More specifically, since $f_t \geq 0$ holds, Theorem 4.1 reduces to:

$$
\mathbb{V}\mathrm{ar}_{\hat{\boldsymbol{n}}}\left[\hat{J}_{\hat{\boldsymbol{n}}}(\pi)\right] \leq 192 \left(\frac{b}{\Lambda}\right)^{\frac{3}{2}} \log(\beta) \left(\sum_{t=0}^{T-1} \gamma^t\right)^2 + 4\mathbb{V}\mathrm{ar}_{\boldsymbol{n}^*}\left[\hat{J}_{\boldsymbol{n}^*}(\pi)\right].
$$

To prove Equation (59) it is thus sufficient to show that, under $\Lambda \geq \Lambda_0$, the following holds:

$$
192 \left(\frac{b}{\Lambda}\right)^{\frac{3}{2}} \log(\beta) \left(\sum_{t=0}^{T-1} \gamma^t\right)^2 \leq \mathbb{V}\mathrm{ar}_{\boldsymbol{n}^*}\left[\hat{J}_{\boldsymbol{n}^*}(\pi)\right].
\tag{60}
$$

We proceed by lower bounding the right hand side of Equation (60).

$$
\mathbb{V}\mathrm{ar}_{\boldsymbol{n}^*}\left[\hat{J}_{\boldsymbol{n}^*}(\pi)\right] = \sum_{t=0}^{T-1} \frac{f_t}{n_t^*} \geq \sum_{t=0}^{T-1} \frac{f_t}{\Lambda} = \frac{1}{\Lambda} \sum_{t=0}^{T-1} f_t,
$$

where, the inequality follows from the fact that $n_t^* \leq \Lambda$ and $f_t \geq 0$. Given this result, Equation (60) holds whenever the following holds:

$$
192 \left(\frac{b}{\Lambda}\right)^{\frac{3}{2}} \log(\beta) \left(\sum_{t=0}^{T-1} \gamma^t\right)^2 \leq \frac{1}{\Lambda} \sum_{t=0}^{T-1} f_t.
\tag{61}
$$

---

[10]Under the assumption that $f_t \geq 0$ holds, the problem is convex, and the KKT conditions provides necessary and sufficient conditions for optimality.

Therefore, we now focus on Equation (61), and proceed by upper-bounding its left hand side. More specifically, we have that:

$$192 \left(\frac{b}{\Lambda}\right)^{\frac{3}{2}} \log\left(\frac{6(T+T^2)\Lambda K}{\delta}\right) \left(\sum_{t=0}^{T-1} \gamma^t\right)^2 \leq 192 \left(\frac{b}{\Lambda}\right)^{\frac{3}{2}} \log\left(\frac{12\Lambda^4}{\delta}\right) \left(\sum_{t=0}^{T-1} \gamma^t\right)^2$$

$$\leq 768 \left(\frac{b}{\Lambda}\right)^{\frac{3}{2}} \log\left(\frac{12\Lambda}{\delta}\right) \left(\sum_{t=0}^{T-1} \gamma^t\right)^2$$

$$\leq 55296 \frac{b^{\frac{3}{2}}}{\delta} \Lambda^{-\frac{4}{3}} \left(\sum_{t=0}^{T-1} \gamma^t\right)^2,$$

where, in the first step we have used $T \leq \Lambda$ and $K \leq \Lambda$, in the second one we have used logarithm properties and in the last one we have used $\log x \leq \frac{x^{\frac{1}{6}}}{\frac{1}{6}}$. At this point, Equation (61) holds whenever the following holds:

$$55296 \frac{b^{\frac{3}{2}}}{\delta} \Lambda^{-\frac{4}{3}} \left(\sum_{t=0}^{T-1} \gamma^t\right)^2 \leq \frac{1}{\Lambda} \sum_{t=0}^{T-1} f_t,$$

which can be rewritten as:

$$\Lambda^{\frac{1}{3}} \geq 55296 \frac{\frac{b^{\frac{3}{2}}}{\delta} \left(\sum_{t=0}^{T-1} \gamma^t\right)^2}{\sum_{t=0}^{T-1} f_t} \tag{62}$$

Equation (62) is clearly satisfied for $\Lambda \geq \Lambda_0$, thus concluding the proof. $\square$

### A.3.4 Theoretical Sub-optimality of pre-determined schedules

Finally, we conclude by providing theoretical evidence on the reasons why claims similar to the one of Corollary (A.13) does not hold for pre-determined schedules (i.e, the uniform-in-the-horizon one and the robust DCS of Poiani et al. [2023]).

**Proposition A.14** (Sub-optimality of the Uniform Strategy). *Let $f_0 \neq 0$ and $f_i = 0$ for all $i \geq 1$. Let $T > 2$. Let $\boldsymbol{n}_u = \left(\frac{T}{\Lambda}, \ldots, \frac{T}{\Lambda}\right)$. For any value of budget $\Lambda$, it does not exist a universal constant $c > 0$ for which the following holds:*

$$\mathbb{V}\mathrm{ar}_{\boldsymbol{n}_u}\left[\hat{J}_{\boldsymbol{n}_u}(\pi)\right] \leq c\mathbb{V}\mathrm{ar}_{\boldsymbol{n}^*}\left[\hat{J}_{\boldsymbol{n}^*}(\pi)\right]. \tag{63}$$

*Proof.* Under the assumption that $f_0 \neq 0$ and $f_i = 0$ for all $i \geq 1$, we have that:

$$\mathbb{V}\mathrm{ar}_{\boldsymbol{n}_u}\left[\hat{J}_{\boldsymbol{n}_u}(\pi)\right] = \frac{T}{\Lambda} f_0, \tag{64}$$

and, from Theorem 3.1:

$$\mathbb{V}\mathrm{ar}_{\boldsymbol{n}^*}\left[\hat{J}_{\boldsymbol{n}^*}(\pi)\right] = \frac{1}{\Lambda - (T-1)} f_0. \tag{65}$$

Furthermore, if $T > 2$, the variance of the optimal DCS can be upper bounded by:

$$\mathbb{V}\mathrm{ar}_{\boldsymbol{n}^*}\left[\hat{J}_{\boldsymbol{n}^*}(\pi)\right] = \frac{1}{\Lambda - (T-1)} f_0 \leq \frac{2}{\Lambda} f_0. \tag{66}$$

At this point, proceed by contradiction and suppose that Equation (63) holds. Then, it follows that the following equation should holds as well for some universal constant $c$:

$$\frac{T}{\Lambda} f_0 \leq c \frac{2}{\Lambda} f_0. \tag{67}$$

Equation (67) reduces to:

$$c \geq \frac{T}{2}, \tag{68}$$

which contradicts the claim, thus concluding the proof. $\square$

**Proposition A.15** (Sub-optimality of the Robust Strategy of Poiani et al. [2023]). *Let $f_0 \neq 0$ and $f_i = 0$ for all $i \geq 1$. Let $\tilde{\boldsymbol{n}}$ be the robust DCS of Poiani et al. [2023]. Let $T > 2$ and $d_t = \frac{\gamma^t(\gamma^t + \gamma^{t+1} - 2\gamma^T)}{1-\gamma}$ and suppose that $\Lambda \geq \Lambda_0 := \frac{\sum_{t=0}^{T-1} \sqrt{d_t}}{\sqrt{d_{T-1}}}$. For any value of budget $\Lambda \geq \Lambda_0$, it does not exist a universal constant $c > 0$ for which the following holds*

$$\mathbb{V}\mathrm{ar}_{\tilde{\boldsymbol{n}}}\left[\hat{J}_{\tilde{\boldsymbol{n}}}(\pi)\right] \leq c\mathbb{V}\mathrm{ar}_{\boldsymbol{n}^*}\left[\hat{J}_{\boldsymbol{n}^*}(\pi)\right]. \tag{69}$$

*Proof.* Under the assumption that $f_0 \neq 0$, $f_i = 0$ for all $i \geq 1$, and $\Lambda \geq \Lambda_0$ we have that:[11]

$$\mathbb{V}\mathrm{ar}_{\tilde{\boldsymbol{n}}}\left[\hat{J}_{\tilde{\boldsymbol{n}}}(\pi)\right] \geq \frac{f_0}{2\Lambda}\frac{\sum_{t=0}^{T-1} \sqrt{d_t}}{\sqrt{d_0}}. \tag{70}$$

From Theorem 3.1:

$$\mathbb{V}\mathrm{ar}_{\boldsymbol{n}^*}\left[\hat{J}_{\boldsymbol{n}^*}(\pi)\right] = \frac{1}{\Lambda - (T-1)}f_0. \tag{71}$$

Furthermore, if $T > 2$, the variance of the optimal DCS can be upper bounded by:

$$\mathbb{V}\mathrm{ar}_{\boldsymbol{n}^*}\left[\hat{J}_{\boldsymbol{n}^*}(\pi)\right] = \frac{1}{\Lambda - (T-1)}f_0 \leq \frac{2}{\Lambda}f_0. \tag{72}$$

At this point, proceed by contradiction and suppose that Equation (69) holds. Then, it follows that the following equation should hold as well for some universal constant $c$:

$$\frac{f_0}{\Lambda}\frac{\sum_{t=0}^{T-1} \sqrt{d_t}}{\sqrt{d_0}} \leq c\frac{2}{\Lambda}f_0. \tag{73}$$

Equation (73) can be rewritten as:

$$c \geq \frac{1}{2}\frac{\sum_{t=0}^{T-1} \sqrt{d_t}}{\sqrt{d_0}}. \tag{74}$$

However, if Equation (74) holds, then, also the following holds:

$$c \geq \frac{1}{4}\sum_{t=0}^{T-1} \sqrt{\gamma^t\left(\gamma^t + \gamma^{t+1} - \gamma^{2T}\right)}, \tag{75}$$

which, however, contradicts the claim [12], thus concluding the proof. $\square$

Proposition A.14 and A.15 shows that the variance of both schedules cannot attain the minimum variance up to multiplicative constant factors as RIDO, instead, does (notice, indeed, that the assumptions on $f_t$ fits the ones of Corollary A.13). These results complements, in this sense, what has been presented in the main text, and highlights the theoretical benefits of adaptive DCSs.

# B Experiment Details and Additional Results

In this section, we provide further details on our experimental settings and additional results. Section B.1 contains descriptions on the environments, Section B.2 contains details regarding hyperparameters, and Section B.3 contains additional results.

Our results have been produced using 100 Intel(R) Xeon(R) Gold 6238R CPU @ 2.20GHz cpus and 256GB of RAM. The total time taken to have all the results is around 2 weeks of computation.

## B.1 Environment Details

In this section, we provide additional details on the environments that we used in our experiments.

---

[11]Notice that the requirement $\Lambda \geq \Lambda_0$ provides a simple closed-form expression for the robust DCS of Poiani et al. [2023]. The reader can refer to Theorem 3.3 and Appendix B of Poiani et al. [2023].

[12]Indeed, it is sufficient to take $T \to +\infty$, and $\gamma \to 1$, to show that Equation (75) tends to $+\infty$.

**Ablation Domains**    In Setion B.3, the reader can find results and ablations that involve the scenarios described as examples in Section 1, namely Examples 1 and 2. We now provide a precise description of these domains. We start with Example 1, where the reward is gathered only at the end of the estimation horizon $T$. The state space is described by a 1-dimensional vector that contains only the interaction timestep $t$; the action space is a discrete set $\{0, 1\}$. The agent receives reward 0 in the first 9 timesteps. In the last step, instead, it receives $r \sim \mathcal{N}(3, 10)$ for action 0, and $r \sim \mathcal{N}(2, 10)$ for action 1. Concerning Example 2, instead, the setup is identical to the one of Example 1, with the only different that the non-zero reward is receives in the first interaction step. The policy that we evaluate is the uniform random.

**Continuous Navigation**    Here, we describe in more details the 2D continuous navigation environment that we used in our experiments. The state space $\mathcal{S}$ is 2-dimensional vector $\boldsymbol{s} = (s_0, s_1) \in \mathbb{R}^2$ such that $s_i \in [0, 92]$ for all $i$. Similarly, the action space $\mathcal{A}$ is a 2-dimensional vector $\boldsymbol{a} = (a_0, a_1)$ such that $a_i \in [-1, 1]$ for all $i$. When the agent takes action $\boldsymbol{a}$ in state $\boldsymbol{s}$, it transitions to a new state $\boldsymbol{s'}$ such that:

$$s'_0 = \max\{0, \min\{s_0 + q_0, 92\}\}, \qquad s'_1 = \max\{0, \min\{s_1 + q_1, 92\}\},$$

where $q_0 \sim \mathcal{N}(a_0, 0.1)$, $q_1 \sim \mathcal{N}(a_1, 0.1)$, and the max-min operations simply guarantees that the resulting state lies within the desired state space $\mathcal{S}$. The agent receives rewards egual to 0 at every time step, except when the resulting state $\boldsymbol{s'}$ falls within a goal region. More specifically, the goal is defined as a 2-dimensional vector $\boldsymbol{g} = (91, 91)$. Whenever $||\boldsymbol{s'} - \boldsymbol{g}||_2 \leq 1$ the reward received by the agent is sampled from the following Gaussian distribution: $\mathcal{N}(1, 1)$. The agent starts in a random position that is sampled from a uniform distribution in the area $[0, 5] \times [0, 5]$. The agent policy that we evaluate in our experiments is an hand-coded expert policy that minimizes the distance between the agent's position and the center of the goal area. More specifically, given the agent position $\boldsymbol{s}$, $\boldsymbol{a}$ is computed in the following way.

$$a_0 = \max\{-1, \min\{g_0 - s_0, 1\}\}, \qquad a_1 = \max\{-1, \min\{g_1 - s_1, 1\}\},$$

where the max-min operation guarantees that $\boldsymbol{a}$ belongs to $\mathcal{A}$.

**LQG**    Concerning the LQG, we consider the following 1-dimensional case (i.e., the dimension of the state and action spaces is 1). The initial state is drawn from a uniform distribution in $[-80, +80]$. Upon taking action $a \in \mathcal{A}$, the agent transitions to a new state $s' = s + (a + \xi) + \eta$, where $\eta \sim \mathcal{N}(0, 0.1)$ models the noise in the system, and $\xi \sim \mathcal{N}(0, 0.1)$ denotes the controller's noise. The reward for taking action $a$ in state $s$ is computed as $s^2 + (a + \xi)^2$. The policy that we evaluate is the optimal one and it is computed by solving the Riccati equations.

**MuJoCo suite**    In the main text, we presented results on the Ant environment of the MuJoCo suite. In the appendix, we present additional experiments on the HalfCheetah and Swimmer domains [Todorov et al., 2012]. In all cases, we adopted trained deep RL agents made publicly available by Raffin [2020] (MIT License).

## B.2    Hyper-parameters

Table 1 reports the hyper-parameters that we used in our experiments. To select the robustness level $\beta$ we tried different values in $[1, 3]$, while for the batch-size we tried different values in $[2T, 10T]$. We then report the results using the best hyper-parameters configuration.

## B.3    Additional Results

### B.3.1    Ablations

In this section, we present ablations on RIDO on the two environments (described in Section B.1) that models Examples 1 and 2. More specifically, we conduct the following two ablations to understand the behavior of RIDO according to changes in its hyper-parameters, i.e., the robustness level $\beta$ and the mini-batch size. To properly assess the effect of these designer's choices, we report and discuss both the average variance and the resulting DCSs. We test our method using $\gamma = 1$, but similar results can be obtained varying the discount factor.

Table 1: Hyper-Parameters

| Environment | $\beta$ | Mini-batch size |
|---|---|---|
| Pendulum | 1.01 | 500 |
| LQG | 2.0 | 400 |
| 2D Continuous Navigation | 1.0 | 1000 |
| Ant | 1.5 | 3000 |
| HalfCheetah | 2 | 3000 |
| Swimmer | 1.0 | 1000 |

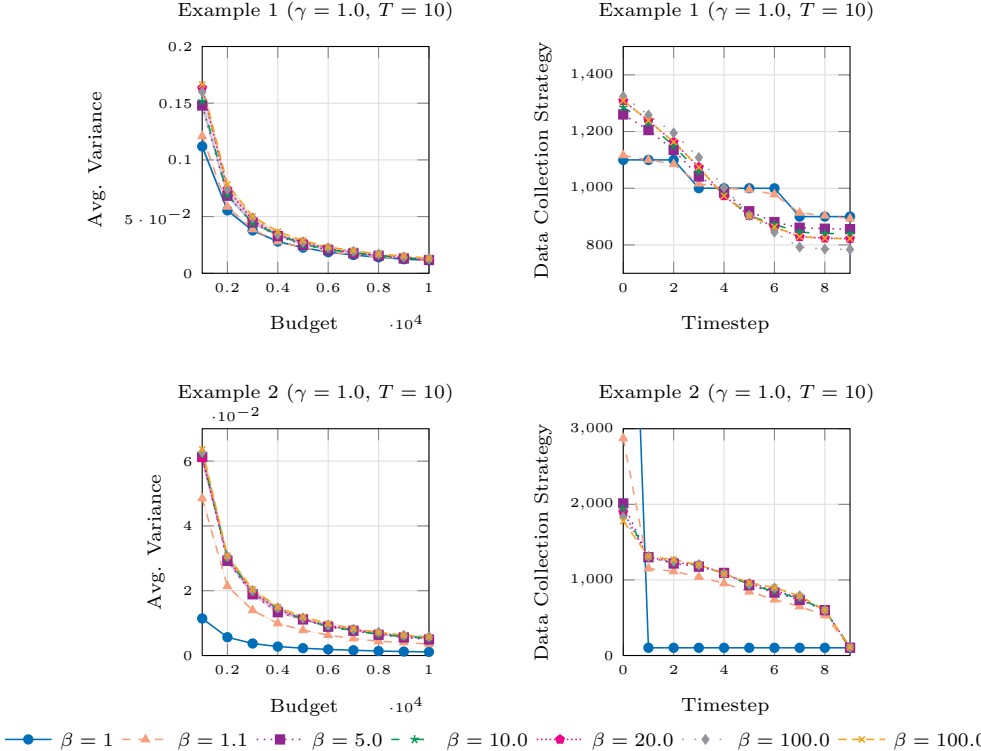

Figure 3: Ablations on different values of $\beta$ on Examples 1 (*top*) and 2 (*bottom*). Empirical variance (mean and 95% confidence intervals over 100 runs) (*left*). DCS visualiaztion (mean and 95% confidence intervals over 100 runs) using $\Lambda = 10000$ (*right*).

**Ablations on $\beta$** We begin by performing an ablation on the robustness parameter $\beta$. More specifically, we analyze the behavior of RIDO for the following values of $\beta$: $\{1, 1.1, 5.0, 10.0, 20.0, 100.0, 1000.0\}$ (the value of the mini-batch size here is fixed to $b = 100$). Figure 3 reports the results. Let us first focus on Example 1 (i.e., the top row). In this case, the reward is gathered at the end of the estimation horizon. As we can see, increasing the value of $\beta$, leads to a larger amount of data spent in the first interaction steps (i.e., top-right in Figure 3). Indeed, when higher values of $\beta$ are used, the cumulative sum of exploration bonuses in the early steps is larger w.r.t. the late ones. For this reason, RIDO spends a larger portion of its budget to decrease these exploration bonuses. As a consequence, given that the reward process of the underlying environment, this results in a higher empirical variance (i.e., top-left in Figure 3). Furthermore, given that the reward is 0 everywhere except at $t = T - 1$, even using the smallest value of $\beta$ (i.e., $\beta = 1$) allows the algorithm to quickly adapt its DCS toward the most relevant timestep (i.e., $t = T - 1$). Similar comments hold for Example 2 as well (i.e., bottom row in Figure 3). Finally, we notice that the behavior changes almost unsgnificantly for values of $\beta$ larger than 5.0.

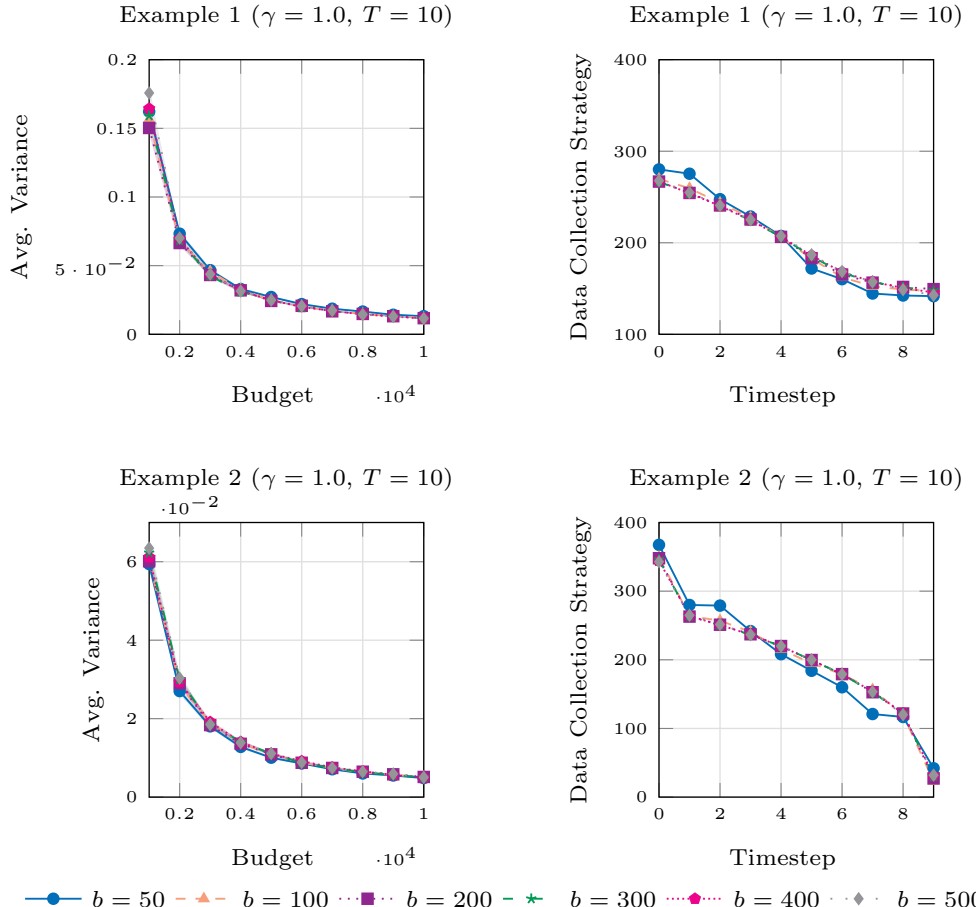

Figure 4: Ablations on different mini-batch sizes on Examples 1 (*top*) and 2 (*bottom*). Empirical variance (mean and 95% confidence intervals over 100 runs) (*left*). DCS visualiaztion (mean and 95% confidence intervals over 100 runs) using $\Lambda = 1000$ (*right*).

**Ablations on mini-batch size** $b$    We now continue by presenting an ablation on the batch size. More specifically, we analyze the behavior of RIDO for the following values of $b$: $\{50, 100, 200, 300, 400, 500\}$ (the value of $\beta$ here is fixed to 5.0). Figure 4 reports the results. First of all, as we can notice, in both Examples 1 and 2 the mini-batch size impacts the performance in a less significant way w.r.t. to the value of $\beta$ (compare the left column of Figure 4 and 3). Secondly, let us focus on the the top-row (i.e., Example 1, where the reward is gathered at the end of the episode). For the smallest value of $\Lambda$ of Figure 4 (i.e., $\Lambda = 1000$, that is the only for which there is some difference in performance), we notice that the best configuration is not $b = 50$ (i.e., the smallest batch-size among the presented ones). This is confirmed also by its corresponding DCS, which is not the one that allocates the highest number of data at $T-1$. We conjecture that the reason behind this phenomena are numerical instabilities that might arise while solving the empirical problem with the use of convex solvers.[13] Concerning Example 2 (where the reward is gathered only at $t = 0$), we notice that smaller values of $b$ performs better (this is confirmed by the corresponding DCS, that allocates more data to the first interaction step). In this case, the aforementioned problem is not present. We conjecture that the reason is that, even in the case of numerical instabilities, errors that arise from converting the continuous DCS to its integer version provably minimizes the variance, since the remaining budget is allocated uniformly starting from $t = 0$ (i.e., the most relevant timestep from

---

[13]We notice that even small imprecisions can result in DCSs that differ by 1 when converting the continuous relaxation to its integer version. For smaller values of $b$, this behavior might happen multiple times w.r.t. larger values of $b$.

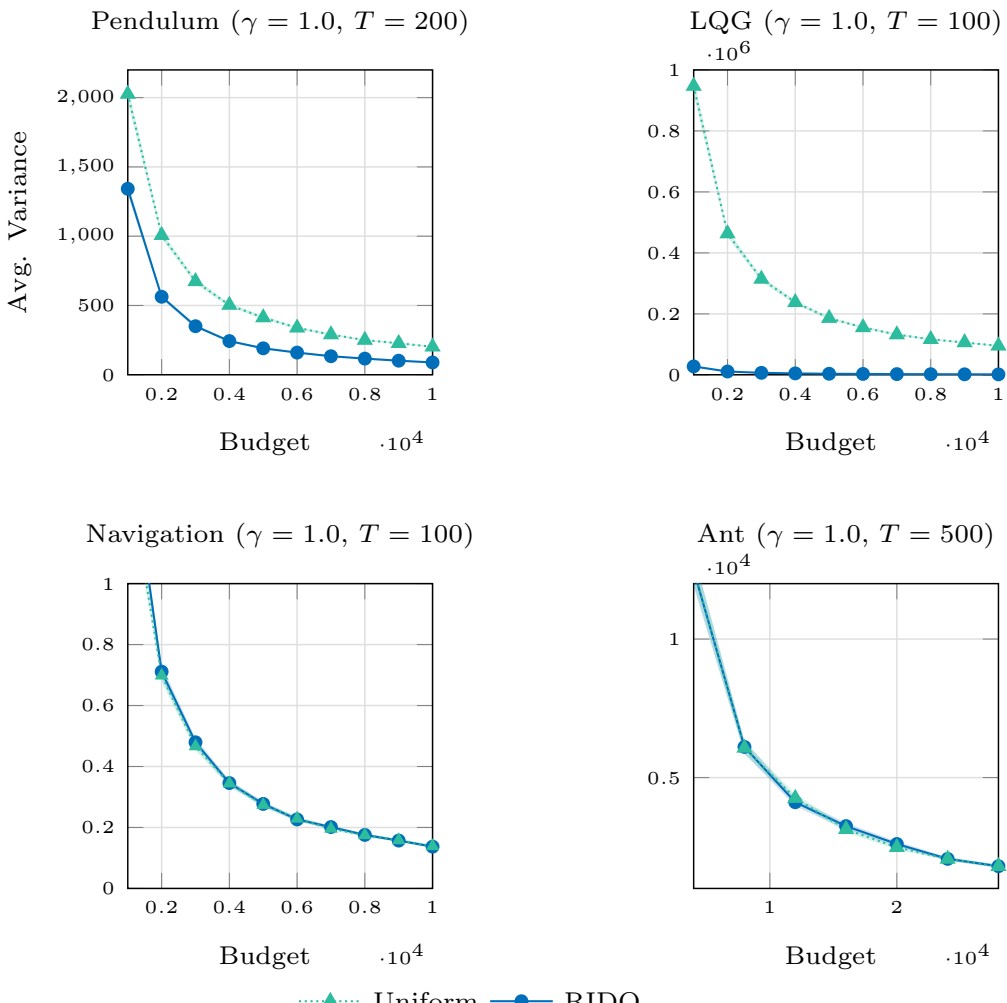

Figure 5: Empirical variance (mean and 95% confidence intervals over 100 runs) on the considered domains and baselines using $\gamma = 1$.

the point of view of the estimation quality). Finally, we notice that, whatever value of $b$ we use, the behavior of RIDO is stable under reasonable variations of the mini-batch size.

### B.3.2 Experiments with $\gamma = 1$

In this section, we present results under the experimental setting of Figure 2 but using $\gamma = 1$. Figure 5 reports the comparison between RIDO and the uniform-in-the-horizon strategy. Notice that these experiments highlight a particular beneficial feature of RIDO w.r.t. the schedule of Poiani et al. [2023]. Indeed, when $\gamma = 1$ their robust DCS does not formally exists (i.e., the method requires $\gamma < 1$, and when $\gamma \to 1$, their strategy tends to the uniform one). RIDO, on the other hand, does not heavily rely on the property of discounted sum and can be applied as-is also when $\gamma = 1$. Furthermore, Figure 5 confirms the adaptivity of RIDO that has already been highlighted in the main text. Namely, it does not underperform the uniform-in-the-horizon strategy when long trajectories are required, while it reduces the return estimator's variance when truncated trajectories are convenient.

### B.3.3 DCS Visualizations for Figure 2

In this section, we present visualizations of the DCSs for the experiments presented in Figure 2 and 5. Figure 6 and 7 reports our results (mean and 95% confidence intervals over 100 runs). For $\gamma = 1$, the robust DCS is missing since it coincides with the uniform-in-the-horizon one (further details on

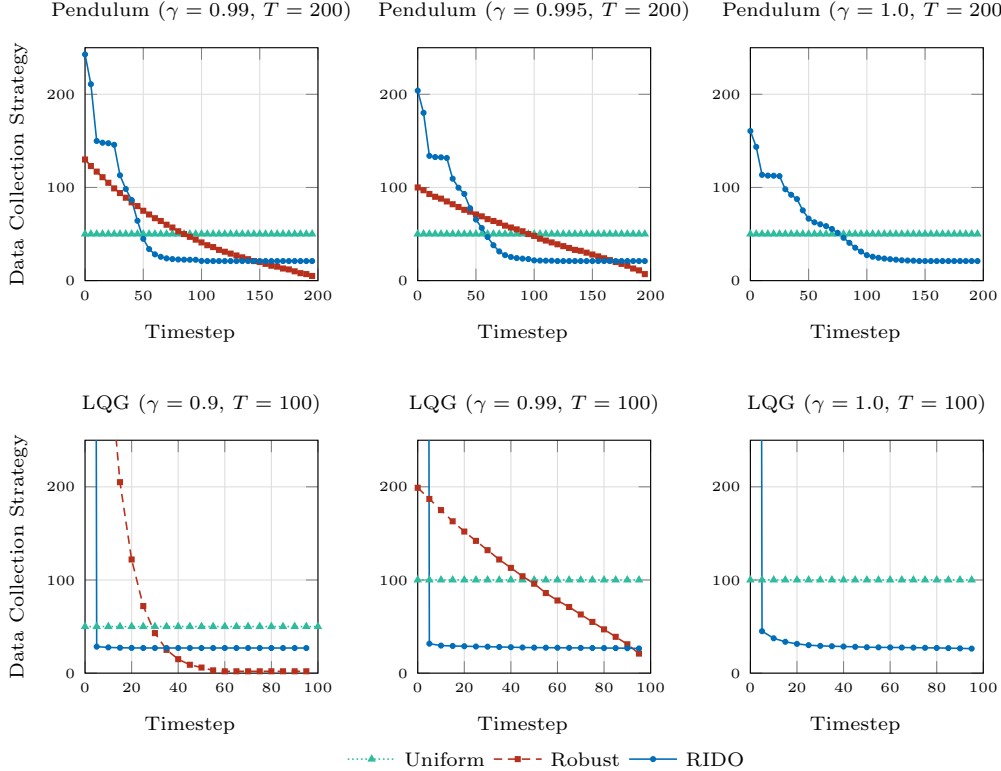

Figure 6: DCS visualiaztion for Pendulum and LQG (mean and 95% confidence intervals over 100 runs). The $x$ axis reports the timestep $t$, while the $y$ axis $n_t$. We consider $\Lambda = 10000$ both for the Pendulum and the LQG.

this point are available in Appendix B.3.2). The resulting visualizations reinforce the adaptivity of RIDO. Indeed, depending on the domain, the behavior of RIDO changes significantly, resulting in behaviors that are similar to the uniform strategy (i.e., Navigation), to the robust strategy (i.e., Ant), or significantly different from both pre-computed schedules (i.e., Pendulum and LQG).

### B.3.4 Results on Additional Environments

In this section, we present results on additional MuJoCo environments, namely Swimmer and Half-Cheetah. Figure 8 reports our results, and Figure 9 the DCSs visualization in the considered domain. In these cases, RIDO confirms its adaptivity achieving a satisfying performance level. For HalfCheetah similar comments w.r.t. made for the Ant in Figure 2. In the Swimmer domain, on the other hand, the robust DCS shows sub-optimal performance w.r.t. the uniform schedule and RIDO.

### B.4 Experiments on the Suite of Experiments of Poiani et al. [2023]

In this section, we provide empirical results on the domains that were analyzed in Poiani et al. [2023]. Figure 10 reports the results.

As one can verify, all the comments made in the main text also directly extend to these situations. In other words, our results confirm (i) the importance of building DCSs that can adapt to the underlying structure of the estimation process and (ii) the ability of RIDO to achieve low variance estimates thanks to its adaptivity.

### B.5 Experiments with Sub-optimal Policies

In this section, we present empirical results on sub-optimal policies. First of all, we would like to remark that all experiments in the main text have proposed an empirical analysis on the performance

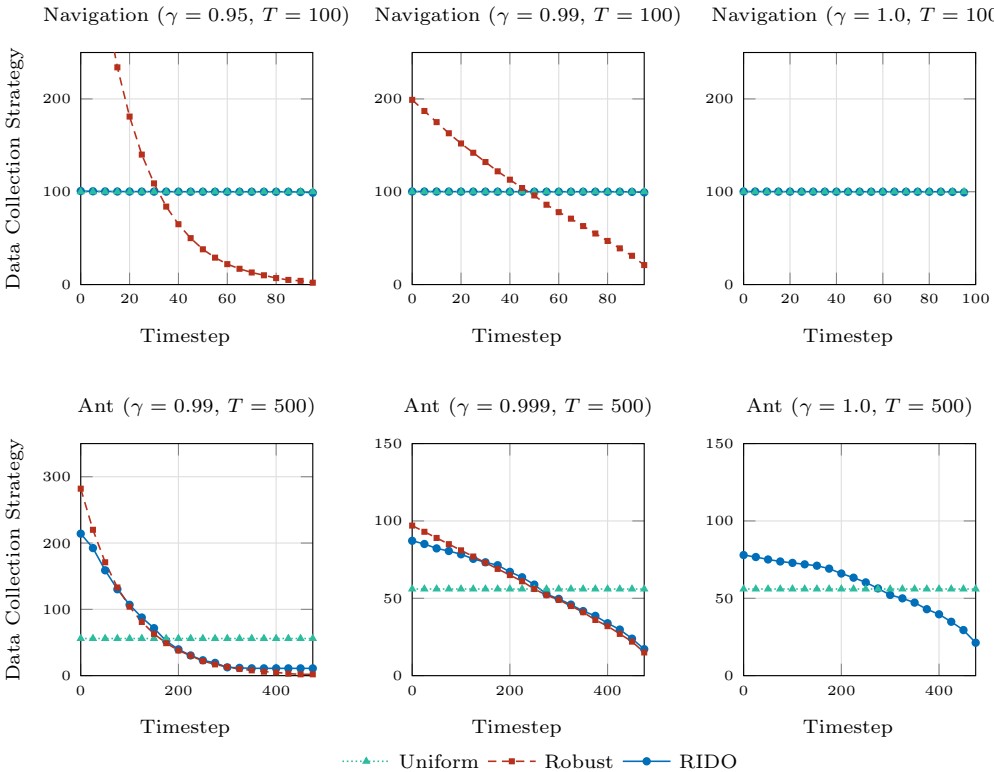

Figure 7: DCS visualization for 2D Continuous Navigation and Ant (mean and 95% confidence intervals over 100 runs). The $x$ axis reports the timestep $t$, while the $y$ axis $n_t$. For 2D Continuous Navigation we consider $\Lambda = 10000$, while for the Ant we consider experiments using $\Lambda = 28000$.

of our method for optimal/pre-trained policies. Nevertheless, in our work, our choice is mainly motivated by the fact that, for these policies, at least for some environments (e.g., LGQ, Navigation), it is possible to provide a straightforward interpretation of the obtained result. Indeed, we remark that, in the theory we developed, the performance index of the evaluated policy $\pi$ does not play any role. What matters, instead, are variances and covariances of the Markov reward process induced by the policy $\pi$ on the MDP at hand. From a theoretical perspective, it is indeed easy to construct MDPs and policies that suffer from identical variances for any possible DCSs but whose performance indexes $J$ completely differ. Furthermore, to empirically demonstrate this point, Figure 11 reports additional results on the LQG domain where we propose the evaluation of 3 different sub-optimal policies, under the following experimental setting: $\gamma = 0.99$, $T = 100$, and $\Lambda = [1000, 2000, 3000, 4000, 5000]$. The policies that we run are the following ones:

- $\pi_1$: the optimal action is perturbed by an additive Gaussian noise with mean 100 and standard deviation 0.01.
- $\pi_2$: a random policy that samples actions uniformly in [0,1]
- $\pi_3$: the optimal action is perturbed by a 0 mean Gaussian additive noise with standard deviation 100

The three considered policies all lead to sub-optimal behaviors. In all cases, however, RIDO is always competitive against the other baselines, thus confirming the importance of building adaptive DCSs that adapt to the underlying Markov reward process.

### B.6  Ablations varying $\gamma$ and $T$

In this section, we provide additional ablations in which we vary the value of $\gamma$ and $T$. We selected one domain (LQG), and kept the budget fixed. In these experiments, we have used the same policy as for the experiments in the main text. Figure 12 reports the results.

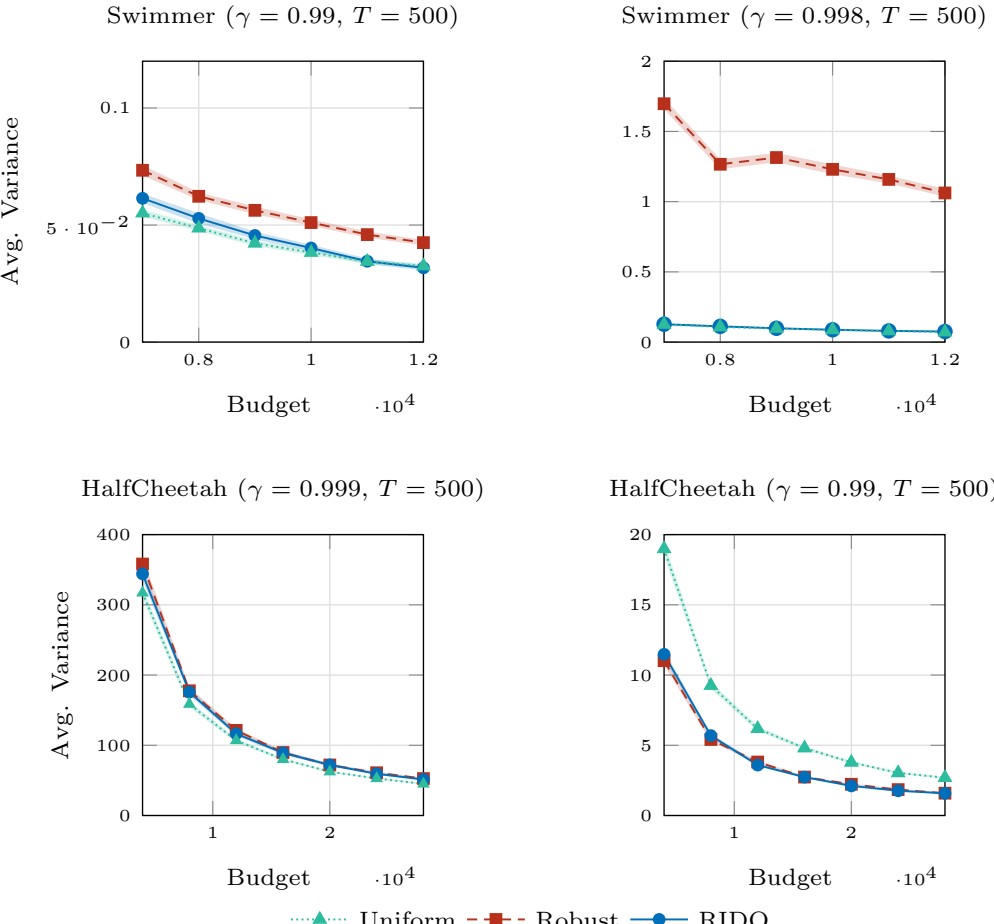

Figure 8: Empirical variance (mean and 95% confidence intervals over 100 runs) on the Swimmer and HalfCheetah for the considered baselines.

Concerning the ablation on $\gamma$, we have run the algorithms with $\Lambda = 3000$, $T = 100$, and $\gamma = [0.7, 0.8, 0.9, 0.95, 0.99, 0.995, 0.999, 1.0]$. As one can verify, RIDO outperforms both baselines for any chosen value of the discount factor (and, especially, for values that are significant for the horizon $T = 100$). Interestingly, we notice that as the discount factor increases, the sub-optimality of the robust strategy of Poiani et al. [2023] increases as well. Indeed, as $\gamma \to 1$, this strategy tends to the uniform one.

Concerning the ablation on $T$, we considered $\Lambda = 1500$, $\gamma = 0.99$, and $T = [25, 50, 75, 100, 125, 150]$. In this case, we notice that for small values of $T$, all methods perform similarly. Indeed, in this case, they all allocate a significant amount of data in estimating the first transition steps (as discussed in the main text, these are the most relevant ones to decrease the variance in the LQG domain using the optimal policy). Interestingly, as $T$ increases, the sub-optimality of Poiani et al. [2023] and the uniform-in-the-horizon DCS increases. Focus for a moment on the uniform DCS; in this case, since $\Lambda$ is fixed, the number of samples allocated to the first interaction steps decreases as $T$ increases. Similar comments also hold for Poiani et al. [2023]. However, in this case, the performance decreases more slowly since their truncating trajectories mechanism implicitly puts more focus on the first interaction steps.

As a summary, both experiments strengthen the importance of adapting to the underlying estimation process and highlight the ability of RIDO to reduce the variance of the return estimator.

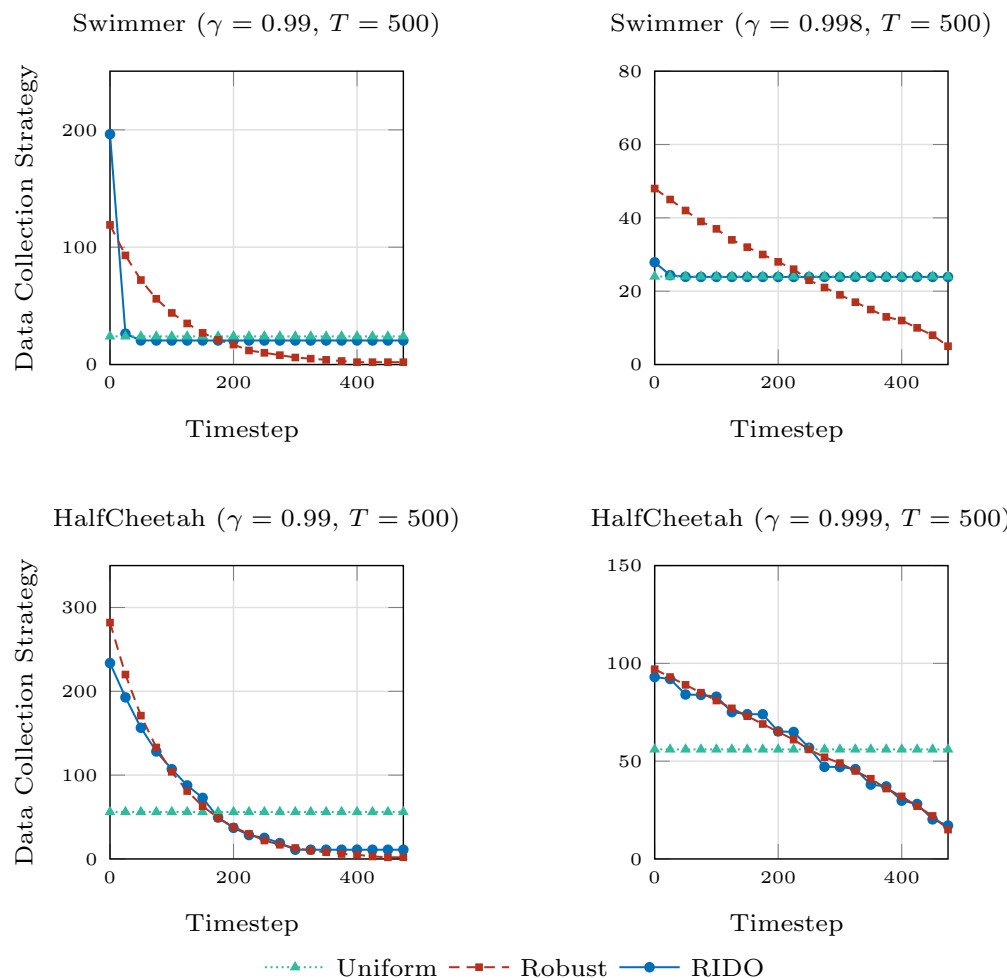

Figure 9: DCS visualization for Swimmer and HalfCheetah (mean and 95% confidence intervals over 100 runs). The $x$ axis reports the timestep $t$, while the $y$ axis $n_t$. For Swimmer we consider $\Lambda = 12000$, while for the HalfCheetah we consider experiments using $\Lambda = 28000$.

### B.7 Additional Details on the Running Time

In this section, we provide additional details on the running time of the algorithms.

More specifically, we have run all algorithms on Ant and HalfCheetah domain with $\Lambda = 4000$ and $T = 500$. For our method, we have used a batch size of 1000. For the Ant, our method took roughly 27 seconds, while for uniform and Poiani et al. [2023], the run took roughly 20 seconds. In the HalfCheetah, instead, RIDO took $23s$, while the baselines $15s$. As soon as we increase $\Lambda = 8000$ (keeping the batch size 1000), we obtain $47s$ for RIDO, and $29s$ for the baselines (Ant environment). For HalfCheetah, instead, $37s$ for RIDO, and $20s$ for the baselines. At this point, increasing the batch size to 2000, RIDO obtains $41s$ in the Ant, and $31s$ in the HalfCheetah.

Overall, RIDO requires some additional computational overheads, nevertheless, the running time is still comparable. Furthermore, we also notice that the code of our algorithm has not been specifically optimized for time efficiency. Finally, we also notice that in our experiments, we rely on open-source solvers. Relying on commercial solvers might increase the computational efficiency of our method.

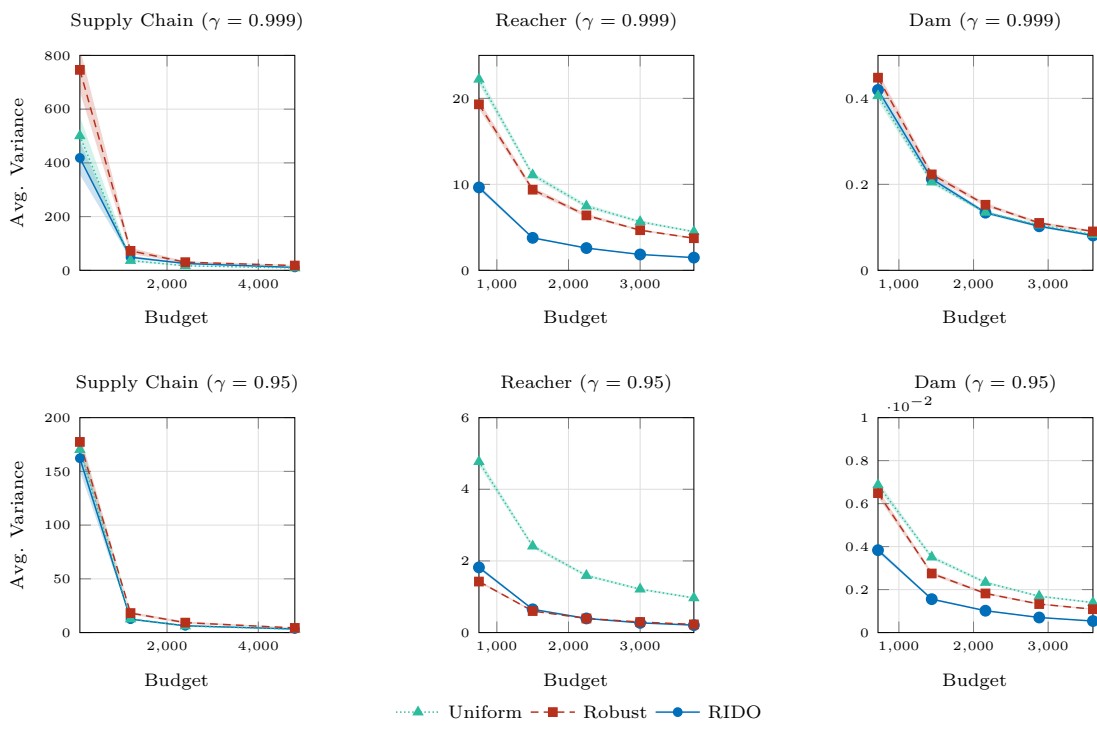

Figure 10: Empirical variance (mean and 95% confidence intervals over 100 runs) on the domains of Poiani et al. [2023].

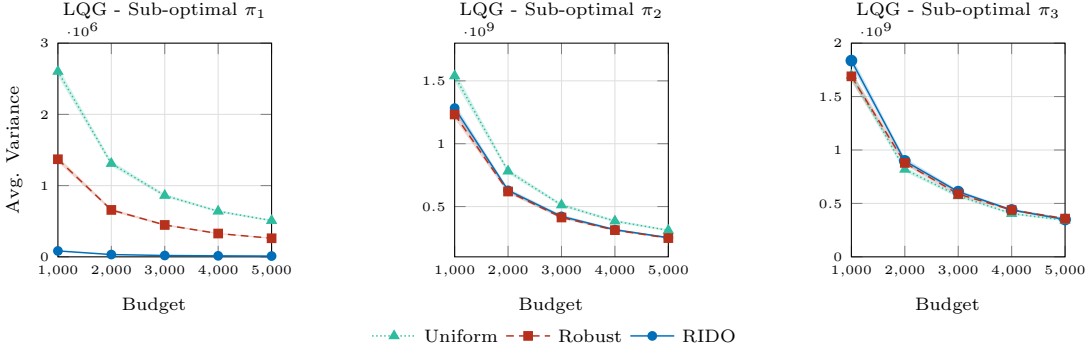

Figure 11: Empirical variance (mean and 95% confidence intervals over 100 runs) on the LQG domain varying the evaluation policy $\pi$.

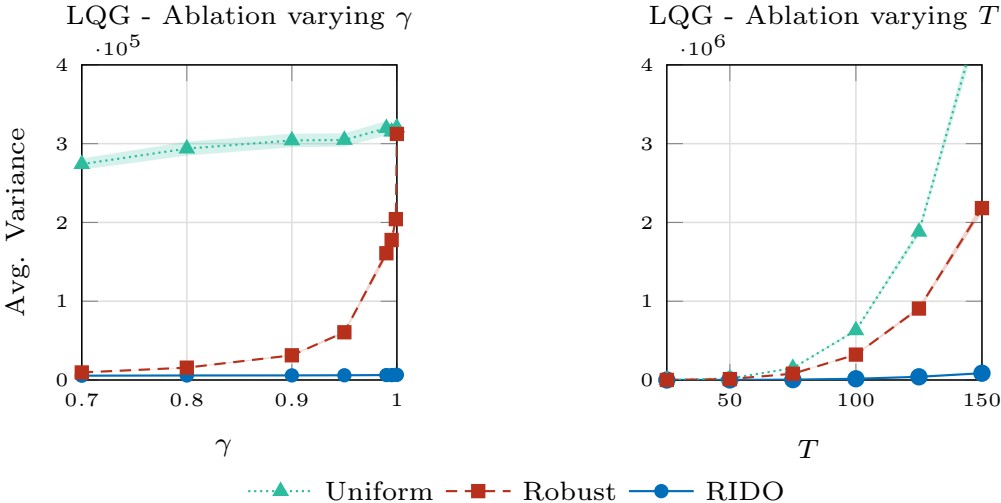

Figure 12: Empirical variance (mean and 95% confidence intervals over 100 runs) on the LQG domain varying the discount factor $\gamma$ and the estimation horizon $T$.