# OpenReview forum: "Truncating Trajectories in Monte Carlo Policy Evaluation: an Adaptive Approach"
_NeurIPS.cc/2023/Conference — NeurIPS 2023 poster_

### Official Review · Reviewer_Acm2 · 2023-07-06

**Soundness:** 3 good
**Presentation:** 2 fair
**Contribution:** 2 fair
**Rating:** 5
**Confidence:** 3

**Summary:**

This paper looks at the problem of designing truncated trajectory lengths to be used in policy evaluation in a reinforcement learning context. It builds on the work [R. Poiani, A. M. Metelli, & M. Restelli, _Truncating Trajectories in Monte Carlo Reinforcement Learning_, ICML 2023], which, instead of using the fixed trajectory lengths that are common in practice, provides a trajectory length selection scheme to minimize confidence interval width around empirical estimates of the expected return of a policy. The paper under review proposes an alternative scheme for selecting trajectory lengths, called RIDO, that focuses on minimizing the variance of the reward estimate. Unlike [Poiani et al., 2023], the proposed scheme is adaptive in the sense that it divides the total interaction budget into mini-batches and iteratively selects intra-batch trajectory length schemes to minimize running estimates of the reward estimate variance. Challenging proxy optimization problems for performing trajectory length sequences are formulated, a transformation to a tractable problem is given, and theoretical results are provided assuring that the resulting reward estimate variance is upper bounded by the best-case reward estimate variance plus error terms that decay as $O(1/\Lambda)$, where $\Lambda$ is the interaction budget. Experimental results are provided indicating that the proposed method achieves lower variance than fixed-length trajectory schemes and the selection scheme proposed in [Poiani et al., 2023].

**Strengths:**

The general topic of the paper is timely and well-motivated, as truncating trajectories to make more efficient use of a limited interaction budget is an important problem with many applications. The proposed adaptive scheme, RIDO, provides another tool for deciding trajectory lengths in addition to that from [Poiani et al., 2023], and the new adaptive scheme appears to provide advantages in certain types of problems (e.g., with delayed rewards). The objectives of the problem formulations (see equations (4) and (5)) provide interesting alternatives to the objective considered in [Poiani et al., 2023], and the convex reformulation of problem (5) and the corresponding analysis in the appendix (which I skimmed, but did not check thoroughly) behind Theorem 4.1 are fairly involved. Finally, the experiments indicate that RIDO enjoys some advantages over that of [Poiani et al., 2023] in terms of return estimate variance minimization.

**Weaknesses:**

I have some concerns about the clarity and significance of this work. Specific concerns include the following:
* Though the proposed method provides an alternative to the selection scheme of [Poiani et al., 2023], aside from the intuitive examples provided on lines 155-158 and the experiments in Figure 2, it remains unclear from the main paper why RIDO is either convincingly superior or on precisely what classes of problems it should be preferred. It seems likely that the ambiguity is due to the fact that the objectives considered in both are closely related: in [Poinani et al., 2023], the objective is to minimize confidence interval width (defined in terms of variance via standard deviation) for the estimated return; in the present work, the objective is to minimize variance of the return estimate. Both objectives seek lower variance.
* In light of the foregoing, a theoretical comparison of the two methods would be very helpful in determining their fundamental differences, but this is absent in the main paper (though Proposition B.15 in the appendix does provide some insight). In addition, the experimental results do suggest that RIDO enjoys lower variance on some problems, but much of the time RIDO and [Poinani et al., 2023]'s method appear to experience roughly the same variance. A more in-depth analysis of the differences between the two methods could reveal the issue here. Inclusion of Proposition B.15 and an expand discussion in the main paper may be helpful in this regard.

It is possible that with additional clarification/expansion the above issues can be resolved. In it's current form, however, its precise contribution, significance, and potential impact remain unclear.

**Questions:**

* Why is your estimator (1) unbiased? In general, such a finite-horizon approximator will be biased due to the infinite-horizon setting under consideration (this is acknowledged in footnote 3 on page 3). The unbiasedness statement appears to be inherited from [Poiani et al., 2023], but it would be helpful to see this made explicit in the current paper, since it is critical for the rest of the analysis.
* What is the effect of policy selection on performance of RIDO versus [Poiani et al., 2023]'s method? You appear to have considered optimal policies when performing policy evaluation. What differences might we expect to see when evaluating policies encountered during training, for example?
* What are the main challenges to using RIDO to perform policy evaluation in a policy search scheme?

**Limitations:**

The authors have adequately addressed the limitations of the present work.

---

> ### Author Rebuttal · Authors · 2023-08-09
>
> 1. **I have some concerns about the clarity and significance of this work**
>
> In the following, we discuss the concerns raised by the reviewer. We hope that the following clarification can solve the raised doubts.
>
> It is of primary importance to remark that a crucial difference between our work and Poiani et al., 2023 lies in the metric used while building alternative DCSs. Indeed, while Poiani et al., 2023 seek strategies that minimize confidence intervals, we directly tackle the more involved problem of variance minimization in our work. In this sense, **minimizing confidence intervals around the return estimator is not equivalent to minimizing the variance**. Indeed, while minimizing confidence intervals can be interpreted as minimizing a worst-case variance (i.e., an upper bound on the actual variance), in our work, we aim to build DCSs that minimize the actual variance. In other words, **we aim to minimize the estimator's error, which is the usual goal in policy evaluation problems.**
>
>
> Furthermore, we notice that the difference in the chosen metric has a crucial implication in the context of truncating trajectories. Indeed, as a direct consequence of minimizing confidence intervals, **the resulting DCS of Poiani et al., 2023 is fixed prior to interacting with the environment.** Nevertheless, as we show in Theorem 3.1, **any schedule that is fixed apriori fails at minimizing the variance of the return estimator**. This formally shows the sub-optimality of prior work. In other words, **the schedule of Poiani et al., 2023 does not seek for lower variance** (i.e., it does not minimize the estimator's error). RIDO, on the other hand, takes the results of Theorem 3.1 directly into account, and it aims at building an adaptive schedule that minimizes the estimators' error.
>
>
> Concerning the experiments, the fact that Poiani et al., 2023 fails at minimizing the error of the estimator is evident. Indeed, we notice **Poiani et al., 2023 underperforms significantly RIDO** in all the following cases:  LQG ($\gamma=0.99$), in the Navigation (both $\gamma=0.99$ and $\gamma=0.95$), in the Swimmer (Appendix, both $\gamma=0.998$ and $\gamma=0.99$ and $\gamma=0.998$), in the Reacher (see attached PDF, $\gamma=0.999$), in the Supply Chain for small values of $\Lambda$ (see attached PDF, $\gamma=0.999$), in the Dam (see attached PDF, $\gamma=0.95$), which are a significant amount of domains. **This sub-optimality is directly dependent on the fact that Poiani et al., 2023 do not seek lower variance by adapting to the feature of the underlying Markov reward process but naively deploys a pre-determined DCS.**
>
> 2. **Why is your estimator (1) unbiased?**
>
> We notice that what we state in Section 2 is that Estimator (1) is an unbiased estimate of $J(\pi) = \mathbb{E} [ \sum_{t=0}^{T-1} \gamma^t R_t]$, i.e., the discounted sum of rewards up to time $T$. As the reviewer suggests, and as we state in our paper, this still introduces a bias w.r.t. the infinite horizon MDP model. Nevertheless, for sufficiently large $T$, this bias is at most $\epsilon$. This procedure, which is commonly adopted in the literature (e.g., "Truncating Trajectories in Monte Carlo Reinforcement Learning, Poiani et al., ICML, 2023, "Smoothing Policies and Safe Policy Gradients" Papini et al., Machine Learning, 2022), serves to justify the approximation of the infinite horizon MDP model with a finite horizon one.
>
> To conclude, we take the chance to remark that our study directly extends to the finite horizon setting, and particularly the case $\gamma=1$. Poiani et al., 2023 did not consider the $\gamma=1$ case.
>
> 3. **What is the effect of policy selection on performance of RIDO versus [Poiani et al., 2023]'s method? What differences might we expect to see when evaluating policies encountered during training, for example?**
>
> We thank the reviewer for raising this interesting point. Reviewer ZDY9 has asked to experiment with sub-optimal as well. Therefore, we defer the reviewer to our in-depth response to Reviewer ZDY9 (see point 3). In short, **in the theory we developed, the performance index of the evaluated policy does not play any role**. Indeed, what matters are variances and covariances of the Markov reward process induced by the policy on the MDP at hand. We have also verified this point empirically in a new set of experiments.
>
> Given the discussion we presented (see Reviewer ZDY9) and the additional experiments, what happens in policies encountered during training only depends on the variance and covariances of the underlying Markov reward process, which is entirely problem dependent. RIDO, for all policies and MDPs, will always seek to truncate trajectories to minimize the estimation error. Other pre-determined DCSs, instead (e.g., uniform and Poiani et al., 2023), will blindly allocate trajectories according to some fixed schedule that ignores the underlying structure of the problem.
>
>
> 4. **What are the main challenges to using RIDO to perform policy evaluation in a policy search scheme?**
>
> First of all, we notice that, in principle, any data collection strategy can be employed in any policy search scheme. Nevertheless, we notice that extending an algorithm to properly incorporate a new DCS is algorithmic dependent. Indeed, consider the case of POIS ("Policy Optimization via Importance Sampling", NeurIPS 2018) that has been extended in Poiani et al., 2023 for their robust DCS. In that case, the extension required proving a theorem to properly incorporate the new DCS within the loss function. For some simple algorithms, such as Cross-entropy methods, any DCS can be incorporated directly to evaluate each candidate in the population using the Estimator of Equation (3). For more complex ones (e.g., POIS), the problem can involve some effort, that, however, depend on the algorithm at hand.

---

> > ### Comment · Reviewer_Acm2 · 2023-08-15
> >
> > Thanks for your thorough response.
> >
> > Part 1 of your response clarified the merits of RIDO over existing methods, particularly that of [Poiani et al., 2023], which was my main concern. Parts 2 through 4 also provided useful additional context/clarification, and I have increased my score accordingly.
> >
> > Regarding part 1, I agree variance is technically distinct from previous objectives, the theoretical results provide support for why we might prefer RIDO, and the experiments illustrate that RIDO often outperforms previous methods w.r.t. minimizing variance. That said, I still highlight that RIDO and [Poiani et al., 2023] experience about the same variance on several domains -- I continue to think that this is due to the fact that minimizing confidence interval width is closely related to minimizing variance, and it again raises the question: on what classes of problems should RIDO be preferred over previous methods? I hope this can be addressed in the revision or future work.

---

> > > ### Author Response · Authors · 2023-08-16
> > > **Additional Clarification**
> > >
> > > We thank the reviewer for the response. We are glad that the main concern has been addressed and that the reviewer increased the score. We now follow up the reviewer's question to clarify the remaining doubt. We hope that the following can show further lights on the contributions of our work.
> > >
> > > In policy evaluation problems, where the **goal is obtaining a low error estimate** of the policy return, Theorem 3.1 clearly shows that the optimal DCS directly depends on the underlying Markov reward process induced by policy $\pi$ on the MDP at hand. As a direct consequence, to minimize the estimation error, **an algorithm with adaptivity properties is required in order to fulfill this goal**. At this point, we remark that RIDO shows the aforementioned adaptivity property both theoretically and empirically. The other available DCSs **(Poiani et al., 2023 and the uniform DCS) fail in being adaptive by definition.**
> > >
> > > In other words, since the agent is not aware of the underlying Markov reward process before interacting with the environment, **RIDO should be preferred to minimize the estimation error for all classes of problems.**

---

### Official Review · Reviewer_Fu4w · 2023-07-06

**Soundness:** 2 fair
**Presentation:** 2 fair
**Contribution:** 2 fair
**Rating:** 5
**Confidence:** 2

**Summary:**

This work proposed a new Monte Carlo (MC) policy evaluation scheme that can adaptively decide the length of the trajectories being sampled.
The authors claim this approach can reduce the variance of the final estimate given a fixed interaction budget.
This work is primarily theoretical but they also have some empirical validation on a set of continuous control tasks.

**Strengths:**

* This work focuses on a rather fundamental problem of reinforcement learning, that is, MC-based policy evaluation. While such methods have been deprecated for quite a while in most RL algorithms, recently, they've been applied again in some sequence modelling methods. So if this approach indeed works well, it can bring benefits to a wide spectrum of downstream algorithms.
* The write-up of the paper is clear.


**Weaknesses:**

* This paper lacks a related work section. MC estimation for a fixed policy is, for example, relevant to the Markov Chain literature. It's hard to believe there is no related work at all.
* To apply this approach, some additional assumptions should be made. So the major one is we can actually terminate the episode and use the "budget" to restart a new episode. Another one is the initial states should be similar if not exactly the same. These assumptions can undermine the practical value of the proposed method.


**Questions:**

In Figure 2, top right corner, the Ant experiment with $\gamma$=0.999 RIDO has a similar performance with the baselines. This is somewhat alarming to me. Because one can interpret this as RIDO only bring benefits for easier tasks. Do you have alternative interpretations or further experiments?

**Limitations:**

Yes.

---

> ### Author Rebuttal · Authors · 2023-08-09
>
> 1. **This paper lacks a related work section. MC estimation for a fixed policy is, for example, relevant to the Markov Chain literature. It's hard to believe there is no related work at all.**
>
> We remark that Appendix A presents and discusses several works that are related to ours. For the sake of presentation, in the main text, we have chosen to delve more deeply into the details of the paper (i.e., Poiani et al., 2023) that bears the closest resemblance to the work we presented.
>
> Concerning Monte Carlo Chain, they are discussed within the work "Monte Carlo Theory, methods and examples" Owen 2013, which we cite in our work. We will include additional book references on Markov Chains (e.g., "Markov Chain and Mixing Times", Levin and Peres, 2017) in the related work section. Nevertheless, to the best of our knowledge, we are not aware of adaptive approaches that are based on truncating trajectories.
>
>
>
> 2. **To apply this approach, some additional assumptions should be made. So the major one is we can actually terminate the episode and use the "budget" to restart a new episode. Another one is the initial states should be similar if not exactly the same. These assumptions can undermine the practical value of the proposed method.**
>
> We thank the reviewer for touching upon these points. In the following, we address each of the limitations raised by the reviewer. We hope that the following discussion can solve the points raised by the reviewer.
>
> - *"The major one is we can actually terminate the episode and use the "budget" to restart a new episode"*
> As mentioned in the introduction, the availability of the simulator that can be exploited to reset the environment, is at grounding core of the entire strand of Reinforcement Learning algorithms that rely on Monte Carlo (MC) simulation to estimate performance/gradients on the task being solved. Despite the availability of alternative methods, like Temporal Difference (TD) ones, which do not necessitate a reset option, a significant number of successful RL approaches continue to depend on MC evaluation (e.g., policy search methods). Indeed, MC-based approaches can be easily applied to non-Markovian environments (which are frequently encountered in real-world domains). This peculiar and favorable property of MC-based methods is directly inherited by our method as well; indeed, although, for simplicity, we formalized the setting using the MDP tool, **our method and all the derived theoretical results do not require a Markovian transition model**.
> - *"Another one is the initial states should be similar if not exactly the same"*
> **Our algorithm, the whole theoretical analysis, and the experiments do not require this assumption**. Indeed, all the expectations (and, consequently, also variances and covariances) are taken w.r.t. the stochasticity of the policy, the transition kernel, and the initial-state distribution. We notice, furthermore, that in all the standard benchmarks (e.g., Pendulum, MuJoCo), the reset method that is available within the classical environments has not been modified (i.e., each time a new trajectory begins, the initial state is sampled according to the random initial distribution of the corresponding domain).
>
> 3. **In Figure 2, top right corner, the Ant experiment with =0.999 RIDO has a similar performance with the baselines. This is somewhat alarming to me. Because one can interpret this as RIDO only bring benefits for easier tasks. Do you have alternative interpretations or further experiments?**
>
> Yes, we do have alternative interpretations and, also, further experiments.
>
> We first proceed with the interpretation. We notice that the performance metric of each algorithm is the variance of the return estimator (i.e., Equation (3)). Equation (3) only depends on the underlying Markov reward process induced by policy $\pi$ in the MDP at hand. Consequently, the performance metric is **not affected** by the complexity of the domain and by how difficult it is to learn a good policy within the environment.
>
> For the sake of clarity, consider the 2D Navigation domain we presented in the main text. The same environment could be replaced with one whose state-space are images taken from a camera. Nevertheless, if the reward function remains unchanged and the agent still deploys the same policy, the underlying Markov reward process would be the same, and the performance of all algorithms would be left unchanged.
>
> Concerning the experiments, instead, in the attached PDF, the reviewer can see that RIDO outperforms all methods in the Reacher domain with $\gamma=0.999$, and significantly outperforms the robust schedule of Poiani et al., 2023 in the Swimmer domain for $\gamma=0.998$ (Figure 8 in the Appendix), and also in the Supply Chain for small values of $\Lambda$ (whose state space dimension is $33$, more than the Ant, that is $27$-dimensional)

---

> > ### Comment · Reviewer_Fu4w · 2023-08-15
> > **Response to the rebuttal**
> >
> > I want to thank the authors for their rebuttal. Some of my questions have been addressed.
> >
> > I don't really buy the argument against the additional assumptions though.
> >
> > > As mentioned in the introduction, the availability of the simulator that can be exploited to reset the environment, is at grounding core of the entire strand of Reinforcement Learning algorithms that rely on Monte Carlo (MC) simulation to estimate performance/gradients on the task being solved
> >
> > I agree the idea of exploiting reset is an interesting one. But the need for a reset function can also be a huge burden if we want to apply RL in the physics world, or digital world with enough complexity.
> >
> > > We notice, furthermore, that in all the standard benchmarks (e.g., Pendulum, MuJoCo), the reset method that is available within the classical environments has not been modified (i.e., each time a new trajectory begins, the initial state is sampled according to the random initial distribution of the corresponding domain).
> >
> > Sure, but these environments only have a Gaussian noise added to the initial state. How about Procedurally generated environments or, say, RLHF settings where the initial states are drastically different?
> >
> > I don't have enough expertise to tell the author's rebuttal on the theoretical results so I'll take their words for now.
> >
> > In general, I think this is an interesting work but unsure about its practical value. I'll keep the scores.

---

> > > ### Author Response · Authors · 2023-08-16
> > > **Additional Clarifications**
> > >
> > > We thank the reviewer for the answer, and we are glad that some concerns have been addressed. In the following, we present additional clarifications that we believe solve the remaining doubts.
> > >
> > > 1. **I agree the idea of exploiting reset is an interesting one. But the need for a reset function can also be a huge burden if we want to apply RL in the physics world, or digital world with enough complexity.**
> > >
> > >
> > > Although we agree with the reviewer that our study exploits a reset function, we remark that **this assumption is necessary and employed by a large number of works published at top-tier venues**. Here, we list a few (but many more could be cited)
> > > - "Simple statistical gradient-following algorithms for connectionist reinforcement learning", Williams, Machine Learning, 1992
> > > - "Infinite-horizon policy-gradient estimation", Baxter and Barlett, Journal of Artificial Intelligence, 2001
> > > - "Policy Optimization via Importance Sampling", Metelli et al., NeurIPS 2018
> > > - "Importance Sampling Techniques for Policy Optimization", Metelli et al., JMLR, 2020
> > > - "Reinforcement Learning in Linear MDPs: Constant Regret and Representation Selection", Papini et al., NeurIPS 2021
> > > - "Kernel-Based Reinforcement Learning: A Finite-Time Analysis", Domingues et al., ICML, 2021
> > > - "Confident Approximate Policy Iteration for Efficient Local Planning in $q^\pi$-realizable MDPs", Weisz et al., NeurIPS 2022
> > > - "Smoothing policies and safe policy gradients", Papini et al., Machine Learning 2022
> > > - "Truncating Trajectories in Monte Carlo RL", Poiani et al., ICML 2023
> > >
> > > More generally, we remark that **both the policy search literature and the finite-horizon RL setting largely rely on the reset function**, (please, notice that our work, both theory and algorithms, extends as-is to the finite horizon setting as well; we have also experiments in the appendix that show the empirical success of RIDO in the setting in which $\gamma=1$).
> > >
> > > Finally, we mention that **many RL success stories relied on a simulator to achieve significant results across a broad spectrum of applications** (e.g., "Mastering the game of Go with deep neural networks and tree search", Silver et al., 2016, "A cooperative multi-agent reinforcement learning framework for resource balancing in complex logistics network", Li et al., AAMAS 2019, "Cooperative Policy Learning with Pre-trained Heterogeneous Observation Representations", AAMAS 2021, "Hindsight Learning for MDPs with Exogenous Inputs", Sinclair et al., ICML 2023; but many mores could be cited).
> > >
> > > **For all these reasons, we believe that the setting is of interest to the community.**
> > >
> > >
> > > 2. **Sure, but these environments only have a Gaussian noise added to the initial state. How about Procedurally generated environments or, say, RLHF settings where the initial states are drastically different?**
> > >
> > > **Our LQG experiment provides an example of an environment where initial states drastically differ.** More precisely, as the reviewer can verify in Appendix C.1, the initial state is sampled uniformly between $[-80, 80]$. Notice that, under the optimal policy considered in the main text, **this leads (for each distinct trajectory) to significantly different states, actions, and rewards in the first interaction steps.** This, in turn, generates significant stochasticity (i.e., variances) in the rewards of the initial interaction steps, which **are directly taken into account by RIDO**. Indeed, we notice that **RIDO is, significantly, the algorithm with the highest performance in this scenario.**  This behavior is a direct consequence of the fact that all expectations are taken (also) w.r.t. the stochasticity of the initial state distribution. Indeed, since RIDO discovers the high variance of the rewards in the initial interaction steps, it successfully allocates more portion of the budget to estimate their rewards.
> > >
> > > To conclude, **both theoretical and empirical evidence support our claim: our work does not rely on this assumption**.

---

### Official Review · Reviewer_ccYJ · 2023-07-08

**Soundness:** 3 good
**Presentation:** 2 fair
**Contribution:** 2 fair
**Rating:** 6
**Confidence:** 3

**Summary:**

This paper tackles the problem of optimally allocating data in Monte Carlo policy evaluation.
It proposes an adaptive approach for setting trajectory lengths.
Building on the Data Collection Strategy (DCS) [Poiani et al. 2023], the authors derive a closed-form variance of an unbiased estimator of the on-policy return. It reveals that a uniform horizon is sub-optimal, and that the optimal method should allocate data adaptively taking into account the variance of the reward, and the covariance of the rewards across time steps.

The authors develop an algorithm, RIDO, which optimizes the DCS based on the data.
The original optimization problem,  a nonlinear integer problem, is transformed into a linear continuous optimization problem that preserves the optimal solution.
Further, an upper bound is established for the variance estimated by RIDO.
The experimental results validate the efficacy of RIDO, demonstrating that it outperforms the non-adaptive approach proposed by Poiani et al. 2023.


**Strengths:**

- The adaptive Data Collection Strategy is well-motivated
- The paper presents solid theoretical analysis, which aligns well with the synthetic examples.
- The experimental results, including comprehensive ablations, validate the efficacy and robustness of the approach (though there is a concern as noted in the weakness section).


**Weaknesses:**

My primary concern lies in the comparison made with the non-adaptive approach from Poiani et al. (2023). The presentation of the comparison is such that it obscures the contributions of the adaptive strategy proposed in this paper. It would be beneficial if the authors could clarify the following points.

1. Theoretical Analysis: Theoretical analysis of the non-adaptive approach was performed for both on-policy and off-policy settings in the referenced work. However, the current paper only provides an analysis in the on-policy setting. Are there specific challenges that preclude analysis in the off-policy setting for the proposed method?

2. Application Scope: In addition to policy evaluation, the non-adaptive approach was applied to policy optimization settings in Poiani et al. (2023). They even developed an algorithm and performed experiments for the latter. However, the current paper does not address the policy optimization setting at all, which reduces its comprehensiveness.

3. Evaluation Environments: Poiani et al. (2023) evaluated their non-adaptive approach in three primary environments (Dam, Reacher, and Supply Chain) and included two additional environments in their appendix specifically for policy evaluation. For an effective comparison of the two approaches, it would be logical to test the proposed method in these same environments. However, the authors evaluated their approach in four different environments that do not overlap with those used in the previous work, making it challenging to accurately assess the contribution of their work.

Another concern is regarding the performance of the adaptive approach, on the discrepancy between the theory and experiment result as seen in example 1 (Fig. 4).
As noted in the paper, the DCS returned by the proposed approach incorrectly allocates more data towards the beginning of episodes and also does not exhibit a preference for a smaller batch size, contradicting the theory. Although the paper attributes this discrepancy to potential numerical instabilities, it is concerning, particularly as it occurs in one of the two illustrative examples. Also, in the limiting case where gamma approaches 1, the non-adaptive approach would return a uniform DCS as suggested by Poiani et al. (2023), thereby outperforming the adaptive approach.


**Questions:**

Please refer to the discussion in the weakness section.

**Limitations:**

The authors discussed the limitations in Section 6.
However, as outlined in the weaknesses section, there are additional limitations that are not addressed in the paper.

---

> ### Author Rebuttal · Authors · 2023-08-09
>
> 1. **Theoretical Analysis: Are there specific challenges that preclude analysis in the off-policy setting for the proposed method?**
>
> We thank the reviewer for touching upon this point. Extending our result to the off-policy setting requires significant and non-trivial effort, and represents an exciting line for future research. One of the main issues lies in deriving an equivalent of Theorem 3.1 (i.e., deriving a closed-form expression of the variance as a function of $\boldsymbol{n}$ that holds with equality). Indeed, the importance sampling weights that would appear within the equations, significantly complicate dealing analytically with variances of the truncated returns. Indeed, they would become variances of importance-weighted truncated returns rather than variances of truncated returns.
>
> We notice that, in Poiani et al., 2023, dealing with confidence intervals for the off-policy case required simple modifications w.r.t. the proofs of the on-policy setting. Analytically, this is simpler since the confidence intervals that they developed rely on *upper bounds* on the variance of the truncated returns (the reviewer can refer to the proof of Theorem B.15 in Poiani et al., 2023). These reasonings, however, cannot be applied to the more complex problem of variance minimization. Indeed, as soon as we upper-bound such quantities, we implicitly lose the adaptivity w.r.t. the underlying Markov reward process (i.e., we are no longer aiming at minimizing the variance). This is, indeed, one significant technical challenge behind our analysis.
>
> 2. **Application Scope: the current paper does not address the policy optimization setting.**
>
> The reviewer is correct in stating that we do not tackle the policy optimization setting. Our work's focus lies in the policy evaluation problem; i.e., **the optimization aspect is outside the scope of the current work**. We remark that policy evaluation represents, in itself, a fundamental problem of Reinforcement Learning. Furthermore, it is also a key component in several RL algorithms that uses policy evaluation as a building block. Therefore, since our theoretical effort directly tackles this relevant problem, we preferred to disentangle the contribution by presenting experiments that tackle the evaluation setting. We do believe that theoretical and empirical advancements in the performance of policy evaluation algorithms are of interest to the community.
>
> 3. **Evaluation Environments: Poiani et al. (2023) evaluated their non-adaptive approach in three primary environments (Dam, Reacher, and Supply Chain).**
>
> We thank the reviewer for proposing this experimental setting. **The reviewer can now find in the attached PDF the performance of RIDO, Poiani et al., 2023, and the uniform DCS across the Dam, Reacher and Supply Chain** (experimental setting 3). As the reviewer can verify, all the comments made in the main text also directly extend to these situations. In other words, **our results confirm (i) the importance of building DCSs that can adapt to the underlying structure of the estimation process and (ii) the ability of RIDO to achieve low variance estimates thanks to its adaptivity**.
>
> To conclude, we take the chance to remark that further experiments and domains are already presented in the appendix (i.e., HalfCheetah and Swimmer). To summarize, we have experimented on 1) several MuJoCo environments (i.e., Ant, HalfCheetah, Swimmer, Reacher), 2) Operation Research environments (Dam and Supply Chain), 3) Control-theory benchmarks (LQG), 4) Navigation environments.
>
> We will include these results in the final version of the manuscript.
>
>
> 4. **Another concern is regarding the performance of the adaptive approach in example 1 (Fig. 4).**
>
> We hope that the following discussion can solve the doubts raised by the reviewer.
>
> **On the numerical instabilities**
> We have verified that that are indeed numerical instabilities within the convex solver. To verify this point, we have run the optimization procedure with $T=5$ and the various $f_t$ equal to $(0, 0, 0, 0, 1)$. In this context, with a budget $\Lambda = 200$, the optimal allocation is clearly $(40, 40, 40, 40, 40)$ (i.e., the uniform strategy). Nevertheless, the solution returned by the optimizer sometimes contains values of $(40.00000192, 40.0000002, 39.99999935, 39.99999878, 39.99999836)$. When flooring, the values shink down to $(40, 40, 39, 39, 39)$, and, allocating the remaining budget uniformly, we obtain $(41, 41, 40, 39, 39)$. Therefore, by increasing the number of iterations, this effect will be reduced since this rounding effect appears less frequently. Nevertheless, we notice that with our rounding strategy, the resulting DCS changes at most by a factor of $1$. As a consequence, for instance, by keeping the ratio $\Lambda/b$ constant, and increasing the budget $\Lambda$ the difference in performance nullifies. Finally, we remark that this is not a weakness of our method, but it only depends on the implementation of the open-source convex solver we used.
>
> **On $\gamma = 1$ in Example 1**
> Finally, concerning the fact that when $\gamma \rightarrow 1$, the reviewer is correct in stating that Poiani et al., 2023 would be uniform and thus optimal in Example 1. However, the main benefit of building adaptive strategies is that they can indeed adapt to the underlying Markov reward process. Indeed, if we consider the opposite case in which the reward is gathered only in the first step (Example 2), the uniform strategy is the DCS with the worst possible variance. RIDO, on the other hand, as shown across several environments, can adapt to the underlying domain.
> Thus, we believe that this phenomenon is an acceptable price to be paid for obtaining the adaptivity of the approach.

---

> > ### Comment · Reviewer_ccYJ · 2023-08-17
> >
> > I appreciate the comprehensive rebuttal from the authors, which has addressed many of my concerns.
> >
> > However, I still find the breadth of the applicability of the work somewhat limiting. Specifically, after  considering your response here and to Acm2, I’m not sure how well the proposed technique can be applied in a policy optimization regime.
> >
> > Also, there seems to be a lack of clarity as to the specific scenarios where the proposed method offers advantages over the non-adaptive approach proposed in Poiani et al., 2023. While both the intuition and the theoretical analysis lean towards the superiority of the adaptive approach, the empirical results from environments like the supply chain, and settings in Reacher, Dam, Ant, Swimmer and HalfCheetah, indicate comparable performance between the two methods.
> >
> > Given the above considerations, I have increased the score accordingly.

---

> > > ### Author Response · Authors · 2023-08-18
> > > **Additional Clarification**
> > >
> > > We thank the reviewer for the response. We are glad that our in-depth rebuttal solved many concerns and that the reviewer increased the score accordingly. In the following, we follow up on the reviewer's comment to clarify one crucial aspect regarding empirical performance.
> > >
> > > First of all, we would like to summarize precisely the performance of RIDO and Poiani et al., 2023. **RIDO and Poiani perform similarly in the following domains**: Supply Chain ($\gamma=0.95$), Reacher ($\gamma=0.95$), Dam ($\gamma=0.999$), Ant and HalfCheetah ($\gamma=0.95$ and $\gamma=0.999$), Pendulum ($\gamma=0.99$), and LQG ($\gamma=0.9$).
> > > **RIDO performs statistically significantly better than Poiani et al., 2023 in the following cases**: Reacher ($\gamma=0.999$),  Dam ($\gamma=0.95$), Supply Chain ($\gamma=0.999$; notice that for small values of $\Lambda$ RIDO obtains half the variance of Poiani et al., 2023), Swimmer ($\gamma=0.99$ and $\gamma=0.998$ ), Navigation ($\gamma=0.99$ and $\gamma=0.95$), and LQG ($\gamma=0.99$).
> > >
> > > At this point, we agree with the reviewer that, in some experiments, the performance of RIDO and Poiani et al., 2023 are comparable. Nevertheless, as highlighted above, there are also cases in which we observe significant performance differences between RIDO and Poiani et al., 2023. We notice that **this is a direct consequence of the fact that RIDO is adaptive w.r.t. the underlying Markov reward process induced by the policy $\pi$ on the MDP hand**. On the other hand, **Poiani et al., 2023 is a fixed schedule blindly deployed before interacting with the environment.** In this sense, we acknowledge that there exist couples of environments and policies to evaluate in which the DCS of Poiani et al., 2023 obtains satisfying performance (usually significantly better than the uniform DCS). Nevertheless, there are also a significant number of cases in which this is not the case.
> > >
> > > The main point is that **RIDO can automatically discover when and how to truncate trajectories according to the underlying scenario, thus achieving competitive performance in all scenarios. On the other hand, the schedule of Poiani et al., 2023, does not enjoy this adaptivity property and can incur (as shown by our experiments) significant sub-optimal performance levels.**

---

> > > > ### Comment · Reviewer_ccYJ · 2023-08-21
> > > >
> > > > I appreciate the authors' follow-up response. I have decided to keep the existing score.

---

### Official Review · Reviewer_scHF · 2023-07-25

**Soundness:** 2 fair
**Presentation:** 2 fair
**Contribution:** 2 fair
**Rating:** 5
**Confidence:** 3

**Summary:**

This paper studies minimizing the variance of an unbiased policy return estimator that uses trajectories of different lengths, i.e., truncated trajectories. They consider the finite budget setting  of budget $\Lambda$ transitions. They only consider an unbiased return estimator, and therefore were able to derive a closed-form expression and analyze it for every possible schedule of trajectories. Then, they define the optimal trajectories schedule as the one that attains the minimum variance subject to the available budget constraint. Then they present their algorithm, Robust and Iterative Data collection strategy Optimization (RIDO), which splits its available budget $\Lambda$ into mini-batches of interactions that are allocated sequentially to minimize an empirical and robust estimate of the variance of the estimator. They derive upper bounds on the variance of the policy return estimator which is of the same order as oracle baseline (under sufficient condition). Finally they empirically validate the performance of RIDO in pendulum, LQG, continuous navigation, and ANT environment.

**Strengths:**

1) The problem is well motivated as the problem of data collection adaptively and non-uniformly by truncating trajectories seems relevant.
2) The proposed solution is analyzed theoretically showing that the upper bounds on the variance of the policy return estimator which is of the same order as oracle baseline .
3) They provide a computationally feasible solution. First, they show that directly solving the original optimization in (5) id computationally difficult as it is an integer, non-linear optimization problem and use a continuous relaxation to solve it.
4) They empirically validate their algorithm in several well-known environments.

**Weaknesses:**

1) The writing needs significant improvement. For example in Figure 2 put the legends in plots. I had to search for the legends. Also the significance of this work compared to [Poiani 2023](https://arxiv.org/pdf/2305.04361.pdf) is not clear. The [Poiani 2023](https://arxiv.org/pdf/2305.04361.pdf)  looks into the same setting but under a non-adaptive data collection strategy. Can the authors should clarify more clearly how this works improve over  [Poiani 2023](https://arxiv.org/pdf/2305.04361.pdf)?

2) Some definitions are missing or confusing. I could not find a formal definition of $\operatorname{Var}\left(R_t\right)$ and $\operatorname{Cov}\left(R_t, R_{t^{\prime}}\right)$. It will be good to define these quantities formally. Also some definitions are very notation heavy, for example in $\hat{J}\_{\boldsymbol{m}}(\pi)=\sum\_{h=1}^T \sum_{i=1}^{m_h} \sum\_{t=0}^{T-1} \frac{\gamma^t}{n_t} R_t^{(i)}$ I am confused between the outer $T$ and inner $T-1$. Can the authors clarify this in detail?

3) After eq (2) it is said that $d_t=\frac{\gamma^t\left(\gamma^t+\gamma^{t+1}-2 \gamma^T\right)}{1-\gamma}$ controls the relative importance of samples gathered at round $t$. No further explanation given regarding this. Is there an easy way to understand the implication of this statement?

4) The integer non-linear optimization in (5) and makes sense to me. However, I am concerned with the approach of continuous relaxation of flooring each $\bar{n}_t^*$ and allocating the remaining budget uniformly. I think this might lead to creeping approximation error. Can the authors justify this approach? Is this standard in literature?

5) One of the main claims by the paper is that their approach adapts to the underlying problem setting.  However, it is not clear from the theorem 4.1 how this shows up in the bound. I could not find and problem dependent complexity parameter in the upper bound of $\operatorname{Var}_{\hat{\boldsymbol{n}}}\left[\hat{J}_{\hat{\boldsymbol{n}}}(\pi)\right]$. Can the authors clarify this?

6) Why no oracle baseline in the experiments? It's a easy way to show how fast RIDO decreases average variance. Also can you describe the key aspects of Robust [Poiani 2023](https://arxiv.org/pdf/2305.04361.pdf) (robust to what?)?

**Questions:**

See weakness section.

**Limitations:**

See weakness section.

---

> ### Author Rebuttal · Authors · 2023-08-09
>
> We thank the reviewer for all the interesting questions and directions for improvements (e.g., clarity).
>
> 1. **Can the authors should clarify more clearly how this works improve over Poiani 2023?**
>
> We will make the legend easily visible in the final version. We now comment on the significance of our work compared to Poiani 2023.
>
> **On Poiani 2023**
> The authors propose a **fixed** DCS that is obtained by minimizing confidence intervals around the policy return estimator. Although this DCS has nice statistical properties, it is **computed prior to interacting with the environment**. Consequently, as the usual uniform DCS, it fails to adapt to the peculiarities of the problem at hand and, ultimately, might not produce a low error estimate **We remark that achieving low-error estimates is the usual goal in policy evaluation problems.**
>
> **On the sub-optimality of previous works (contribution 1)**
> In our work, we **directly tackle the problem of variance minimization** for policy evaluation. To this end, we first derive, in closed form, the expression of the variance of the return estimator for any DCS (Theorem 3.1). This expression makes a major theoretical contribution to the study of truncating trajectories. Indeed, since it depends on quantities that are unknown to the agent (e.g., variance and covariance), it implies that any pre-determined DCS will fail at minimizing this quantity. Consequently, **it formally shows the sub-optimality of the approach presented in Poiani 2023**.
>
> **On adaptive methods (contribution 2)**
> We then exploit Theorem 3.1 to derive an algorithm (RIDO) that **adapts** to the peculiarity of the underlying process. The algorithm enjoys favorable theoretical properties, such as Theorem 4.1, which guarantees that, under favorable conditions, the upper bound on the variance of the return estimator is of the same order as the oracle baseline. As shown by Proposition B.14 and B.15, this result (which required an involved theoretical study) is impossible for the uniform DCS and that of Poiani 2023.
>
> **Experiments (contribution 3)**
> The failure of existing and fixed DCSs to adapt to the underlying process has been verified empirically across several domains, values of $\gamma$ and budget $\Lambda$. This shows light on the importance of building strategies that directly minimizes the error of the return estimator.
>
> We believe that contributions 1, 2, and 3 represent significant advancements (both theoretical and empirical). We will include this in-depth comparison in the final version.
>
> 2. **Some definitions are missing or confusing**
>
> We will add a table in the appendix with the definition of all the symbols used in the paper.
>
> Concerning variances and covariances, we note that $R_t$ is the r.v. we defined in Sec 2 and their symbols are, as usual, w.r.t. to the stochasticity induced by the policy, the transition kernel, and the initial state distribution. Concerning $\hat{J}(\pi)$ there is a typo we have now fixed. The correct expression is $\hat{J}(\pi) = \sum_{h=1}^{T} \sum_{i=1}^{m_h} \sum_{t=0}^{{\color{red}\mathbf{h-1}}} \frac{\gamma^t}{n_t} R_t^{(i)}$. As explained in Section 2, the two external summations iterate over the collected trajectories of different lengths the rescaled empirical trajectory return.
>
> 3. **Explanation of $d_t$**
>
> In Eq. 2, $n_t$ always appears for a fixed $t$ in combination with $d_t$, ie $\frac{d_t}{n_t}$. Thus, for any DCS, $d_t$ directly weighs the reciprocal of the number of samples collected at $t$ (i.e., their relative importance). Finally, since $d_t$ is a decreasing function of $t$, this implies that to minimize Eq. (2), more samples should be allocated to the initial steps.
>
> 4. **I am concerned with the approach flooring each and allocating the remaining budget uniformly.**
>
> Lemma B.5 shows that this rounding strategy introduces only a constant multiplicative factor of 2 in Theorem 4.1. A similar argument is also used in Poiani 2023. However, this kind of reasoning (Lemma B.5) exploits the structure of the specific objective function. Thus, using this reasoning within the analysis is problem-dependent and has to be studied case by case.
>
> 5. **I could not find and problem dependent complexity parameter in the upper bound**
>
> Within our context, the problem lies in learning a DCS that can successfully minimize the variance of the return estimator. In this sense, the most relevant problem-dependent quantity for our setting is the **variance of the return estimator under the optimal DCS** defined as in Eq. (4), ie $\mathrm{Var}_{n^*}[\hat{J}]$.
>
> This quantity captures the problem-dependent features of the underlying process induced by the policy $\pi$ on the MDP. For these reasons, the effort behind Theorem 4.1 is in deriving an upper-bound that directly depends on $\mathrm{Var}_{n^*}[\hat{J}]$. Since the optimal DCS does not admit a closed-form expression retrieving this sort of result requires significant technical effort.
>
> 6. **Why no oracle baseline? Can you describe the key aspects of Poiani 2023?**
>
> We note from Eq. 4 that computing the oracle baseline requires (i) exact knowledge of Var and Cov at each $t$ and (ii) solving an NP-hard problem. Concerning (i), in the settings we consider, this information is not available. One might object that it is possible to estimate Var and Cov using a large amount of data and then use such estimates to build the oracle. However, (a) this still requires solving an NP-hard problem, and (b) to avoid introducing correlation between different runs, the estimation of Var and Cov and the computation of the NP-hard problem should be repeated for each run, environment, and value of the budget. To conclude, this baseline is not present because it is not available / cannot be computed efficiently.
>
> Key aspects of Poiani 2023 were remarked in point 1. "Robust" is the name we have chosen to denote their DCS since their schedule comes with tighter confidence intervals w.r.t. the uniform DCS.

---

> > ### Comment · Reviewer_scHF · 2023-08-16
> > **Further clarification on author's rebuttal**
> >
> > I thank the authors for their response. I have a few further queries which will help me to understand and place the paper in context:
> > - "Consequently, as the usual uniform DCS, it fails to adapt to the peculiarities of the problem at hand and, ultimately, might not produce a low error estimate" : Can the authors be more specific about this comment? Can you give a particular example where this happens and you outperform DCS (not just experiments, deeper insights/intuition)?
> > - Thank you for clarifying the theoretical contribution w.r.t Poiani 2023
> > - Regarding the typo in $\hat{J}(\pi)$, does it affect any theoretical result? It is not exactly possible for me to dig deep, but would like clarification from the authors.
> > - "Since the optimal DCS does not admit a closed-form expression retrieving this sort of result requires significant technical effort." Can you expand on this point?
> > - When I was talking about Oracle baseline, I am talking like this, take the RIDO algorithm and replace the variance with true variance (calculated over many samples). I am not sure why this cannot be done. Plot your avg variance reduction against such an oracle baseline. If you are reducing close to it, it basically means that you are close to optimal. Can you clarify why this can't be done?

---

> > > ### Author Response · Authors · 2023-08-16
> > > **Additional Clarifications**
> > >
> > > We thank the reviewer for the response, and we are glad that some concerns, such as the theoretical contribution w.r.t. Poiani et al., 2023 have been clarified. In the following, we answer the additional questions.
> > >
> > > 1. **"Consequently, as the usual uniform DCS, it fails to adapt to the peculiarities of the problem at hand and, ultimately, might not produce a low error estimate" : Can the authors be more specific about this comment? Can you give a particular example where this happens and you outperform DCS (not just experiments, deeper insights/intuition)?**
> > >
> > > We thank the reviewer for the question. We hope that the following discussion can clarify the doubt.
> > >
> > > First of all, **Example 1 provides an intuitive example of why this is the case**. We recall that in Example 1, the underlying Markov reward process induced by the policy $\pi$ gathers rewards **only** at the end of the estimation horizon $T$. In this scenario, to minimize the variance of the return estimator, we would expect an intelligent agent to spend the entire budget $\Lambda$ in collecting trajectories of length $T$ (i.e., the uniform DCS). However, since the schedule of DCS Poiani et al., 2023 blindly truncates trajectories prior to interacting with the environment (i.e., **the DCS is fixed and equal for all the possibles underlying Markov reward processes**), it fails to adapt to the peculiarities of the problem at hand (i.e., the fact that, in this case, rewards are gathered **only** at the end of the estimation horizon $T$), and consequently, it fails to minimize the variance of the return estimator (i.e., producing a low error estimate). Similar comments holds for Example 2, but replacing Poiani et al., 2023 with the uniform DCS.
> > >
> > > **Theorem 3.1, on the other hand, also provides a formal justification for the statement quoted by the reviewer**. Indeed, Theorem 3.1 shows that the **DCS that minimizes the variance of the return estimator directly depends on the underlying Markov reward process that is induced by policy $\pi$ for the MDP at hand**. For this reason,
> > > **all predetermined schedule** (and, consequently, also the one of Poiani et al., 2023) fails to minimize the variance of the return estimator (i.e., producing a low error estimate).
> > >
> > >
> > >
> > > 2. **Regarding the typo in $\hat{J}(\pi)$, does it affect any theoretical result? It is not exactly possible for me to dig deep, but would like clarification from the authors.**
> > >
> > > We thank the reviewer for stressing on this point. **The theoretical contribution are completely left unchanged. The typo was only in Equation (2).**
> > >
> > >
> > >
> > >
> > >
> > > 3. **Since the optimal DCS does not admit a closed-form expression retrieving this sort of result requires significant technical effort." Can you expand on this point?**
> > >
> > > We thank the reviewer for asking this question, as this highlights the analytical challenges behind our analysis. As mentioned, the problem is that Equation (4) does not admit a closed-form expression. We notice that this problem is also present in the empirical version of Equation (4) that RIDO exploits during each mini-batch.
> > >
> > > As a consequence, all the analysis to upper-bound the variance of RIDO with the one of Equation (4) deals with **unknown quantities** (i.e., solutions of non-linear optimization problems for which we do not have a closed-form expression), which requires careful manipulations. More precisely, our analysis relies on several transformations of optimization problems, together with sensitivity analysis, that formally describes how the solution of the empirical optimization problem relates to the oracle one provided in Equation (4).
> > >
> > >
> > >
> > > 4. **Can you clarify why this can't be done?**
> > >
> > > We try to clarify this point more precisely w.r.t. our previous answer.
> > >
> > > From the definition of the oracle baseline (Equation 4), we notice that computing the oracle baseline requires (i) exact knowledge of the underlying Markov reward process (i.e., variance and covariances at each interaction step, and (ii) the ability to solve an integer and non-linear optimization problem.
> > >
> > > Concerning (i), we remark that in the environments we considered, this information is not available. At this point, as suggested by the reviewer, one might be tempted to estimate $\mathbb{V}\mathrm{ar}(R_t)$ and  $\mathbb{C}\mathrm{ov}(R_t, R_t')$ using a large amount of data, and then using such estimates to build the oracle baseline (i.e., solving the optimization problem defined in Equation 4).
> > >
> > > Nevertheless, (a) this still requires solving an **NP-hard problem**, and (b) to **avoid introducing correlation between different runs**, the estimation of $\mathbb{V}\mathrm{ar}(R_t)$ and  $\mathbb{C}\mathrm{ov}(R_t, R_t')$, and the computation of the NP-hard **should be repeated for each run, for each environment and for each value of the budget** $\Lambda$, which, however, is computationally infeasible. We remark that this is the correct way of computing the performance of an **estimated** oracle baseline.

---

> > > > ### Author Response · Authors · 2023-08-20
> > > >
> > > > Dear Reviewer,
> > > >
> > > > Since the discussion period is coming close to its end, we would kindly ask you if the previous set of answers has helped in placing our paper into context. Thank you very much in advance for your time in revising our paper and engaging in the discussion. We remain available for further clarification.
> > > >
> > > > The Authors.

---

> > > > > ### Comment · Reviewer_scHF · 2023-08-21
> > > > > **Response to authors**
> > > > >
> > > > > Thanks for your response and I have noted the additional clarifications as well. We will discuss these responses in the reviewer-area chair discussion period.

---

### Official Review · Reviewer_ZDY9 · 2023-07-29

**Soundness:** 3 good
**Presentation:** 3 good
**Contribution:** 3 good
**Rating:** 7
**Confidence:** 2

**Summary:**

* This paper studies online policy evaluation in RL using monte carlo methods.
* A standard evaluation approach when there is access to a simulator is running rollouts of a given policy with some time horizon, and computing the average return over these.
* However, this approach for evaluation isn't necessarily optimal, since it may make sense to collect variable length trajectories to evaluate a policy (given that the discount factor downweights future rewards).
* Closely related prior work to this paper identified that collecting truncated rollouts/trajectories is a policy eval strategy with attractive theoretical properties.
* However, this prior work proposes a non-adaptive collection strategy that is computed prior to evaluation, based only on the discount factor. This non-adaptive strategy does not work as well in practice (as shown in expts) because it does not adapt to specifics of the problem at hand (for example, the specific nature of the rewards induced by the policy being followed, based on the underlying MDP).
* The present work proposes a collection strategy for how to adaptively sample trajectories such that one can evaluate the policy with the lowest estimation error.
* The proposed algorithm proceeds by allocating the total evaluation budget into acquiring minibatches of trajectories with a some starting horizon. Then at each iteration, the algorithm uses the already collected trajectories to compute the number of trajectories to gather at the next horizon length, where this number is deduced based on how to obtain the lowest variance policy evaluation estimate.
* The paper provides theoretical justification for the proposed algorithm, including that it provably is low variance.
* In experiments, the proposed algorithm is used to evaluate effective policies in different benchmarks: inverted pendulum, linear quadratic regulator, 2D continuous navigation, and Ant (from Mujoco). When compared to the algorithm from prior work (that precomputes the collection strategy based on the discount factor) and a uniform monte carlo scheme, the proposed method does significantly better.

**Strengths:**

* Problem is very well framed and articulated. Examples as to where existing methods fail was very clear
* Overall notation is easy to understand and well presented, helping to make the contributions clearer
* Problem tackled is original -- it's clear what gap in the literature that it tackles.
* To the best of my knowledge (did not go in depth into the proofs), the proposed algorithm is (1) theoretically sound (in terms of variance guarantees) and (2) empirically effective compared to baselines.



**Weaknesses:**

* From a writing perspective, it might help to break up big paragraphs of text (particularly in the expts section) -- these are hard to parse. Using bullet points/subsectioning would help a lot to make it easier to read.
* My main comments on areas to improve are related to experiments:
   * The evaluation focuses on policies that are optimal, or near-optimal. Can you demonstrate how it performs on more suboptimal policies?
   * What is the motivation for the specific discount factors chosen in the experiments? Since the discount factor is an important property of the method, can you demonstrate (maybe just for one domain) how the estimation quality varies as a more fine-grained function of the discount factor and evaluation budget? (or maybe at a fixed eval budget, to save computation time).
   * Similar to above, but related to the horizon length T?
   * Can you comment on computational cost of running your algorithm instead of the Robust strategy and the Uniform strategy?
* Small point, but the appendix has a lot of interesting experiments -- would be nice to summarize what these are in the main text (particularly related to ablation on $\beta$, and additional environment experiments from Mujoco.

**Questions:**

See above.

**Limitations:**

Limitations  are alluded to but having a subsection/paragraph would be good.

---

> ### Author Rebuttal · Authors · 2023-08-09
>
> 1. **Can you demonstrate (for one domain) how the estimation quality varies as a more fine-grained function of the discount factor**.
> 2. **Similar to above, but related to the horizon length T?**
>
> We thank the reviewer for highlighting these interesting experimental setups. **In the attached PDF, the reviewer can find results for each setting** (experimental setting 1). As suggested by the reviewer, we selected one domain (LQG) and kept the budget fixed. Here, we have used the same policy as for the experiments in the main text.
>
> Concerning (1), we have run the algorithms with $\Lambda = 3000$ and $T=100$, and different $\gamma$'s. RIDO outperforms both baselines for any chosen value of the discount factor (and, especially, for values that are significant for the horizon $T=100$). Interestingly, we notice that as the discount factor increases, the sub-optimality of the robust strategy of Poiani et al., 2023 increases as well. Indeed, as $\gamma \rightarrow 1$, this DCS tends to the uniform one.
>
> Concerning (2), we considered $\Lambda = 1500$, $\gamma = 0.99$, and different $T$'s. In this case, we notice that for small values of $T$, all methods perform similarly. Indeed, in this case, they all allocate a significant amount of data in the first transition steps (i.e., the most relevant for the LQG and the given policy). Interestingly, as $T$ increases, the sub-optimality of Poiani et al., 2023 and the uniform DCS increases. Focus for a moment on the uniform DCS; in this case, since $\Lambda$ is fixed, the number of samples allocated to the first interaction steps decreases as $T$ increases. Similar comments also hold for Poiani et al., 2023. However, in this case, the performance decreases more slowly since their DCS implicitly puts more focus on the first interaction steps.
>
> As a summary, **both experiments strengthen the importance of adapting to the underlying estimation process and highlight the ability of RIDO to reduce the variance of the return estimator**.
>
> To conclude, similarly to Poiani et al., 2023, the chosen value of the $\gamma$ we used in the main text where chosen to benchmark the methods against different values of $\gamma$ that highlight different properties of the methods.
>
>
> 3. **Can you demonstrate how it performs on more suboptimal policies?**
>
> We thank the reviewer for raising this interesting point. The reviewer is correct in mentioning that our evaluation focuses on policies that perform optimal/near-optimal. Nevertheless, in our work, our choice is mainly motivated by the fact that, for these policies, at least for some environments (e.g., LGQ, Navigation), it is possible to provide a straightforward interpretation of the obtained result.
>
> Indeed, we remark that **in the theory we developed, the performance index of the evaluated policy $\pi$ does not play any role**. Indeed, what matters are variances and covariances of the Markov reward process induced by the policy $\pi$ on the MDP at hand. From a theoretical perspective, it is indeed easy to construct MDPs and policies that suffer from identical variances for any possible DCSs but whose performance indexes $J$ completely differ. Furthermore, to empirically demonstrate this point, **we included a new set of experiments** (experimental setting 2) on the LGQ domain that the reviewer can find in the attached PDF. More specifically, we propose the evaluation of $3$ different sub-optimal policies:
> - $\pi_1$: the optimal action is perturbed by an additive Gaussian noise with mean $100$ and std $0.01$.
> - $\pi_2$: a random policy that samples actions uniformly in [0,1]
> - $\pi_3$: the optimal action is perturbed by a $0$ mean Gaussian additive noise with std $100$
>
> The three considered policies all lead to sub-optimal behaviors. In all cases, however, **RIDO is always competitive against the other baselines, thus confirming the importance of building adaptive DCSs that adapt to the underlying Markov reward process.** To conclude, we notice that these experiments confirm that what matters is not the level of optimality of a certain policy, but only the underlying Markov reward process that is induced on the MDP.
>
> 4. **Can you comment on computational cost of running your algorithm?**
>
> To answer the question, we have made some experiments in which we timed the execution.
>
> More specifically, we have run all algorithms on Ant and HalfCheetah with $\Lambda = 4000$ and $T = 500$.  For RIDO we have used a batch size of $b=1000$. For the Ant, RIDO took roughly $27$ seconds, while the baselines $20s$. In the HalfCheetah, RIDO took $23s$, while the baselines $15s$. As soon as we increase $\Lambda = 8000$ (keeping $b=1000$), we obtain $47s$ for RIDO, and $29s$ for the baselines (Ant). For HalfCheetah, instead, $37s$ for RIDO, and $20s$ for the baselines. At this point, increasing $b=2000$, RIDO obtains $41s$ in the Ant, and $31s$ in the HalfCheetah.
>
> Overall, RIDO requires some additional computational overheads, however, the running time is still comparable. Furthermore, we also notice that the code of our algorithm has not been specifically optimized for time efficiency. Finally, in our experiments, we rely on open-source solvers available within the cvxpy library. Relying on commercial solvers might increase the computational efficiency.
>
> All these additional results will be included in the final version of this paper.
>
> 5. **It might help to break up big paragraphs of text (particularly in the expts section)**
> 6. **The appendix has a lot of interesting experiments -- would be nice to summarize what these are in the main text**
>
> We thank the reviewer for these suggestions, and we are happy that the reviewer appreciated our additional experiments. Unfortunately, due to space constraints, we had to make some choices in the presentation of our results. Nevertheless, we are happy to take these modifications into account for the final version of our work using the additional page that is available for camera-ready.

---

> > ### Comment · Reviewer_ZDY9 · 2023-08-12
> > **Thank you for your response!**
> >
> > I thank the authors for their detailed response -- These answer my questions!

---

> > > ### Author Response · Authors · 2023-08-14
> > > **Thank you!**
> > >
> > > We thank the reviewer for the acknowledgement. We are happy that all questions have been answered.
> > >
> > > The Authors.

---

### Author Rebuttal · Authors · 2023-08-09

We thank the reviewers for their effort in reviewing our paper. We are happy that the reviewers have recognized that "the problem tacked is original" (Reviewer ZDY9) and "well-motivated" (Reviewer scHF, ccYJ, Acm2), that the "limited interaction budget is an important problem with many applications" (Reviewer Acm2), that the paper presents "solid theoretical analysis" (Reviewer ccYJ), and the method is "empirically effective against the baselines" (Reviewer ZDY9). In the following, we address the concerns raised by the reviewers.

We attach a PDF containing additional experiments that were requested by Reviewer ZDY9 and ccYJ.

---

### Decision · Program_Chairs · 2023-09-21

**Decision:**

Accept (poster)

**Comment:**

Monte Carlo policy evaluation is a fundamental problem in RL and this paper studies how it can be made more data efficient by adaptively truncating trajectories. The key novelty is to estimate the variance of the reward at different time-steps and adaptively set the sample budget per time-step to focus on acquiting more samples at time-steps where variance is high. Theoretical and numerical studies show the benefit of this approach compared to a fixed horizon truncation scheme or not truncating.

Main strengths: Given the fundamental importance of the Monte Carlo estimator to RL, this paper makes a contribution that could have widespread impact in different RL methods. The work proposes a novel method (adaptive trajectory truncation) and provides theoretical analysis of its benefits and policy evaluation experiments showing a reduction in mean squared error. Overall the combination of theoretical and empirical analysis is strong.

Main concerns: Main concerns about the paper were 1) how novel it was relative to a paper from earlier this year on non-adaptive trajectory truncation and 2) the fact that the method relies on a "reset" function which is an unrealistic assumption in many real world settings. A minor comment is that in the camera ready I would encourage the authors to discuss related work in the main paper body as I had questions about novelty initially that the section that is currently in the appendix helped clarify. Another reviewer also had this concern.

Overall the paper presents a novel method with intuitive justification, theoretical analysis of potential benefit, and empirical analysis complementing theory. While a couple of limitations are noted, I believe the paper is sufficiently novel relative to the older work and, even if many RL environments lack reset functions, there are also many environments where resets are practical. The widespread use of Monte Carlo sampling in RL means the work has potential broad impact and so I recommend acceptance.